# DiffDock: Diffusion Steps, Twists, and Turns for Molecular Docking

**Gabriele Corso**\*, **Hannes Stärk**\*, **Bowen Jing**\*, **Regina Barzilay** & **Tommi Jaakkola**
CSAIL, Massachusetts Institute of Technology

## Abstract

Predicting the binding structure of a small molecule ligand to a protein—a task known as *molecular docking*—is critical to drug design. Recent deep learning methods that treat docking as a regression problem have decreased runtime compared to traditional search-based methods but have yet to offer substantial improvements in accuracy. We instead frame molecular docking as a *generative* modeling problem and develop DiffDock, a diffusion generative model over the non-Euclidean manifold of ligand poses. To do so, we map this manifold to the product space of the degrees of freedom (translational, rotational, and torsional) involved in docking and develop an efficient diffusion process on this space. Empirically, DiffDock obtains a 38% top-1 success rate (RMSD<2Å) on PDB-Bind, significantly outperforming the previous state-of-the-art of traditional docking (23%) and deep learning (20%) methods. Moreover, while previous methods are not able to dock on computationally folded structures (maximum accuracy 10.4%), DiffDock maintains significantly higher precision (21.7%). Finally, DiffDock has fast inference times and provides confidence estimates with high selective accuracy.

## 1 Introduction

The biological functions of proteins can be modulated by small molecule ligands (such as drugs) binding to them. Thus, a crucial task in computational drug design is *molecular docking*—predicting the position, orientation, and conformation of a ligand when bound to a target protein—from which the effect of the ligand (if any) might be inferred. Traditional approaches for docking [Trott & Olson, 2010; Halgren et al., 2004] rely on scoring-functions that estimate the correctness of a proposed structure or pose, and an optimization algorithm that searches for the global maximum of the scoring function. However, since the search space is vast and the landscape of the scoring functions rugged, these methods tend to be too slow and inaccurate, especially for high-throughput workflows.

Recent works [Stärk et al., 2022; Lu et al., 2022] have developed deep learning models to predict the binding pose in one shot, treating docking as a regression problem. While these methods are much faster than traditional search-based methods, they have yet to demonstrate significant improvements in accuracy. We argue that this may be because the regression-based paradigm corresponds imperfectly with the objectives of molecular docking, which is reflected in the fact that standard accuracy metrics resemble the *likelihood* of the data under the predictive model rather than a regression loss. We thus frame molecular docking as a *generative modeling problem*—given a ligand and target protein structure, we learn a distribution over ligand poses.

To this end, we develop DiffDock, a diffusion generative model (DGM) over the space of ligand poses for molecular docking. We define a diffusion process over the degrees of freedom involved in docking: the position of the ligand relative to the protein (locating the binding pocket), its orientation in the pocket, and the torsion angles describing its conformation. DiffDock samples poses by running the learned (reverse) diffusion process, which iteratively transforms an uninformed, noisy prior distribution over ligand poses into the learned model distribution (Figure 1). Intuitively, this process can be viewed as the progressive refinement of random poses via updates of their translations, rotations, and torsion angles.

---

\*Equal contribution. Correspondence to {`gcorso, hstark, bjing`}`@mit.edu`.

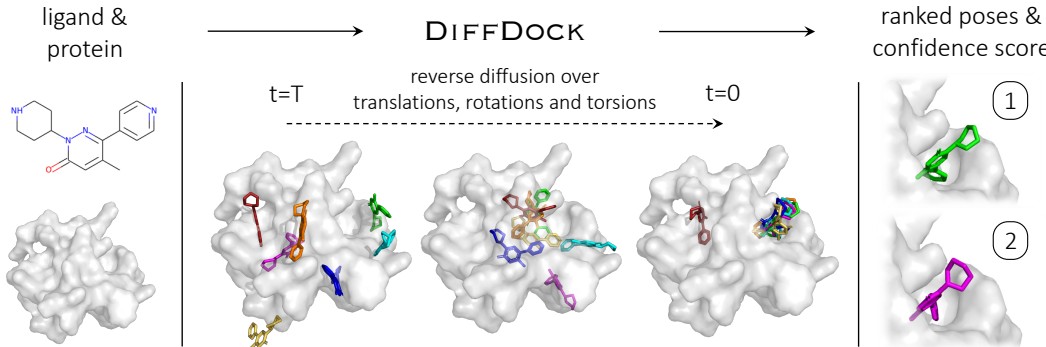

Figure 1: Overview of DIFFDOCK. *Left*: The model takes as input the separate ligand and protein structures. *Center*: Randomly sampled initial poses are denoised via a reverse diffusion over translational, rotational, and torsional degrees of freedom. *Right*:. The sampled poses are ranked by the confidence model to produce a final prediction and confidence score.

While DGMs have been applied to other problems in molecular machine learning [Xu et al., 2021; Jing et al., 2022; Hoogeboom et al., 2022], existing approaches are ill-suited for molecular docking, where the space of ligand poses is an $(m + 6)$-dimensional submanifold $\mathcal{M} \subset \mathbb{R}^{3n}$, where $n$ and $m$ are, respectively, the number of atoms and torsion angles. To develop DIFFDOCK, we recognize that the docking degrees of freedom define $\mathcal{M}$ as the space of poses accessible via a set of allowed *ligand pose transformations*. We use this idea to map elements in $\mathcal{M}$ to the product space of the groups corresponding to those transformations, where a DGM can be developed and trained efficiently.

As applications of docking models often require only a fixed number of predictions and a confidence score over these, we train a *confidence model* to provide confidence estimates for the poses sampled from the DGM and to pick out the most likely sample. This two-step process can be viewed as an intermediate approach between brute-force search and one-shot prediction: we retain the ability to consider and compare multiple poses without incurring the difficulties of high-dimensional search.

Empirically, on the standard blind docking benchmark PDBBind, DIFFDOCK achieves 38% of top-1 predictions with ligand root mean square distance (RMSD) below 2Å, nearly doubling the performance of the previous state-of-the-art deep learning model (20%). DIFFDOCK significantly outperforms even state-of-the-art search-based methods (23%), while still being 3 to 12 times faster on GPU. Moreover, it provides an accurate confidence score of its predictions, obtaining 83% RMSD<2Å on its most confident third of the previously unseen complexes.

We further evaluate the methods on structures generated by ESMFold [Lin et al., 2022]. Our results confirm previous analyses [Wong et al., 2022] that showed that existing methods are not capable of docking against these approximate apo-structures (RMSD<2Å equal or below 10%). Instead, without further training, DIFFDOCK places 22% of its top-1 predictions within 2Å opening the way for the revolution brought by accurate protein folding methods in the modeling of protein-ligand interactions.

To summarize, the main contributions of this work are:

1. We frame the molecular docking task as a generative problem and highlight the issues with previous deep learning approaches.

2. We formulate a novel diffusion process over ligand poses corresponding to the degrees of freedom involved in molecular docking.

3. We achieve a new state-of-the-art 38% top-1 prediction with RMSD<2Å on PDBBind blind docking benchmark, considerably surpassing the previous best search-based (23%) and deep learning methods (20%).

4. Using ESMFold to generate approximate protein apo-structures, we show that our method places its top-1 prediction with RMSD<2Å on 28% of the complexes, nearly tripling the accuracy of the most accurate baseline.

## 2 BACKGROUND AND RELATED WORK

**Molecular docking.** The molecular docking task is usually divided between known-pocket and blind docking. Known-pocket docking algorithms receive as input the position on the protein where the molecule will bind (the *binding pocket*) and only have to find the correct orientation and conformation. Blind docking instead does not assume any prior knowledge about the binding pocket; in this work, we will focus on this general setting. Docking methods typically assume the knowledge of the protein holo-structure (bound), this assumption is in many real-world applications unrealistic, therefore, we evaluate methods both with holo and computationally generated apo-structures (unbound). Methods are normally evaluated by the percentage of hits, or approximately correct predictions, commonly considered to be those where the ligand RMSD error is below 2Å [Alhossary et al., 2015; Hassan et al., 2017; McNutt et al., 2021].

**Search-based docking methods.** Traditional docking methods [Trott & Olson, 2010; Halgren et al., 2004; Thomsen & Christensen, 2006] consist of a parameterized physics-based scoring function and a search algorithm. The scoring-function takes in 3D structures and returns an estimate of the quality/likelihood of the given pose, while the search stochastically modifies the ligand pose (position, orientation, and torsion angles) with the goal of finding the scoring function's global optimum. Recently, machine learning has been applied to parameterize the scoring-function [McNutt et al., 2021; Méndez-Lucio et al., 2021]. However, these search-based methods remain computationally expensive to run, are often inaccurate when faced with the vast search space characterizing blind docking [Stärk et al., 2022], and significantly suffer when presented with apo-structures [Wong et al., 2022].

**Machine learning for blind docking.** Recently, EquiBind [Stärk et al., 2022] has tried to tackle the blind docking task by directly predicting pocket keypoints on both ligand and protein and aligning them. TANKBind [Lu et al., 2022] improved over this by independently predicting a docking pose (in the form of an interatomic distance matrix) for each possible pocket and then ranking them. Although these one-shot or few-shot regression-based prediction methods are orders of magnitude faster, their performance has not yet reached that of traditional search-based methods.

**Diffusion generative models.** Let the data distribution be the initial distribution $p_0(\mathbf{x})$ of a continuous diffusion process described by $d\mathbf{x} = \mathbf{f}(\mathbf{x}, t)\, dt + g(t)\, d\mathbf{w}$, where $\mathbf{w}$ is the Wiener process. *Diffusion generative models* (DGMs)[1] model the *score*[2] $\nabla_{\mathbf{x}} \log p_t(\mathbf{x})$ of the diffusing data distribution in order to generate data via the reverse diffusion $d\mathbf{x} = [\mathbf{f}(\mathbf{x}, t) - g(t)^2 \nabla_{\mathbf{x}} \log p_t(\mathbf{x})] + g(t)\, d\mathbf{w}$ [Song et al., 2021]. In this work, we always take $\mathbf{f}(\mathbf{x}, t) = 0$. Several DGMs have been developed for molecular ML tasks, including molecule generation [Hoogeboom et al., 2022], conformer generation [Xu et al., 2021], and protein design [Trippe et al., 2022]. However, these approaches learn distributions over the full Euclidean space $\mathbb{R}^{3n}$ with 3 coordinates per atom, making them ill-suited for molecular docking where the degrees of freedom are much more restricted (see Appendix E.2).

## 3 DOCKING AS GENERATIVE MODELING

Although EquiBind and other ML methods have provided strong runtime improvements by avoiding an expensive search process, their performance has not reached that of search-based methods. As our analysis below argues, this may be caused by the models' uncertainty and the optimization of an objective that does not correspond to how molecular docking is used and evaluated in practice.

**Molecular docking objective.** Molecular docking plays a critical role in drug discovery because the prediction of the 3D structure of a bound protein-ligand complex enables further computational and human expert analyses on the strength and properties of the binding interaction. Therefore, a docked prediction is only useful if its deviation from the true structure does not significantly affect the output of such analyses. Concretely, a prediction is considered acceptable when the distance between the structures (measured in terms of ligand RMSD) is below some small tolerance on the order of the length scale of atomic interactions (a few Ångström). Consequently, the standard evaluation metric used in the field has been the percentage of predictions with a ligand RMSD (to the crystal ligand pose) below some value $\epsilon$.

---

[1] Also known as diffusion probabilistic models (DPMs), denoising diffusion probabilistic models (DDPMs), diffusion models, score-based models (SBMs), or score-based generative models (SGMs).

[2] Not to be confused with the *scoring function* of traditional docking methods.

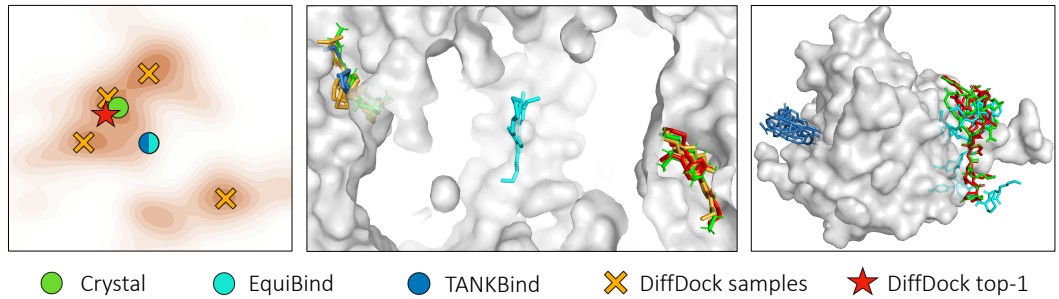

○ Crystal    ○ EquiBind    ○ TANKBind    ✗ DiffDock samples    ★ DiffDock top-1

Figure 2: "DIFFDOCK top-1" refers to the sample with the highest confidence. "DIFFDOCK samples" to the other diffusion model samples. *Left:* Visual diagram of the advantage of generative models over regression models. Given uncertainty in the correct pose (represented by the orange distribution), regression models tend to predict the mean of the distribution, which may lie in a region of low density. *Center:* when there is a global symmetry in the protein (aleatoric uncertainty), EquiBind places the molecule in the center while DIFFDOCK is able to sample all the true poses. *Right:* even in the absence of strong aleatoric uncertainty, the epistemic uncertainty causes EquiBind's prediction to have steric clashes and TANKBind's to have many self-intersections.

However, the objective of maximizing the proportion of predictions with RMSD within some tolerance $\epsilon$ is not differentiable and cannot be used for training with stochastic gradient descent. Instead, maximizing the expected proportion of predictions with RMSD $< \epsilon$ corresponds to maximizing the likelihood of the true structure under the model's output distribution, in the limit as $\epsilon$ goes to 0. This observation motivates training a generative model to minimize an upper bound on the negative log-likelihood of the observed structures under the model's distribution. Thus, we view molecular docking as the problem of learning a distribution over ligand poses conditioned on the protein structure and develop a diffusion generative model over this space (Section 4).

**Confidence model.** With a trained diffusion model, it is possible to sample an arbitrary number of ligand poses from the posterior distribution according to the model. However, researchers are often interested in seeing only one or a small number of predicted poses and an associated confidence measure[3] for downstream analysis. Thus, we train a confidence model over the poses sampled by the diffusion model and rank them based on its confidence that they are within the error tolerance. The top-ranked ligand pose and the associated confidence are then taken as DIFFDOCK's top-1 prediction and confidence score.

**Problem with regression-based methods.** The difficulty with the development of deep learning models for molecular docking lies in the aleatoric (which is the data inherent uncertainty, e.g., the ligand might bind with multiple poses to the protein) and epistemic uncertainty (which arises from the complexity of the task compared with the limited model capacity and data available) on the pose. Therefore, given the available co-variate information (only protein structure and ligand identity), any method will exhibit uncertainty about the correct binding pose among many viable alternatives. Any regression-style method that is forced to select a single configuration that minimizes the expected square error would learn to predict the (weighted) mean of such alternatives. In contrast, a generative model with the same co-variate information would instead aim to capture the distribution over the alternatives, populating all/most of the significant modes even if similarly unable to distinguish the correct target. This behavior, illustrated in Figure 2, causes the regression-based models to produce significantly more physically implausible poses than our method. In particular, we observe frequent steric clashes (e.g., 26% of EquiBind's predictions) and self-intersections in EquiBind's and TANKBind's predictions (Figures 4 and 12). We found no intersections in DIFFDOCK's predictions. Visualizations and quantitative evidence of these phenomena are in Appendix F.1.

---

[3]For example, the pLDDT confidence score of AlphaFold2 [Jumper et al., 2021] has had a very significant impact in many applications [Necci et al., 2021; Bennett et al., 2022].

## 4 METHOD

### 4.1 OVERVIEW

A ligand pose is an assignment of atomic positions in $\mathbb{R}^3$, so in principle, we can regard a pose $\mathbf{x}$ as an element in $\mathbb{R}^{3n}$, where $n$ is the number of atoms. However, this encompasses far more degrees of freedom than are relevant in molecular docking. In particular, bond lengths, angles, and small rings in the ligand are essentially rigid, such that the ligand flexibility lies almost entirely in the torsion angles at rotatable bonds (see Appendix E.2 for further discussion). Traditional docking methods, as well as most ML ones, take as input a seed conformation $\mathbf{c} \in \mathbb{R}^{3n}$ of the ligand in isolation and change only the relative position and the torsion degrees of freedom in the final bound conformation.[4] The space of ligand poses consistent with $\mathbf{c}$ is, therefore, an $(m + 6)$-dimensional submanifold $\mathcal{M}_{\mathbf{c}} \subset \mathbb{R}^{3n}$, where $m$ is the number of rotatable bonds, and the six additional degrees of freedom come from rototranslations relative to the fixed protein. We follow this paradigm of taking as input a seed conformation $\mathbf{c}$, and formulate molecular docking as learning a probability distribution $p_{\mathbf{c}}(\mathbf{x} \mid \mathbf{y})$ over the manifold $\mathcal{M}_{\mathbf{c}}$, conditioned on a protein structure $\mathbf{y}$.

DGMs on submanifolds have been formulated by De Bortoli et al. [2022] in terms of projecting a diffusion in ambient space onto the submanifold. However, the kernel $p(\mathbf{x}_t \mid \mathbf{x}_0)$ of such a diffusion is not available in closed form and must be sampled numerically with a geodesic random walk, making training very inefficient. We instead define a one-to-one mapping to another, "nicer" manifold where the diffusion kernel can be sampled directly and develop a DGM in that manifold. To start, we restate the discussion in the last paragraph as follows:

> *Any ligand pose consistent with a seed conformation can be reached by a combination of (1) ligand translations, (2) ligand rotations, and (3) changes to torsion angles.*

This can be viewed as an informal definition of the manifold $\mathcal{M}_{\mathbf{c}}$. Simultaneously, it suggests that given a continuous family of ligand pose transformations corresponding to the $m + 6$ degrees of freedom, a distribution on $\mathcal{M}_{\mathbf{c}}$ can be lifted to a distribution on the product space of the corresponding groups—which is itself a manifold. We will then show how to sample the diffusion kernel on this product space and train a DGM over it.

### 4.2 LIGAND POSE TRANSFORMATIONS

We associate translations of ligand position with the 3D translation group $\mathbb{T}(3)$, rigid rotations of the ligand with the 3D rotation group $SO(3)$, and changes in torsion angles at each rotatable bond with a copy of the 2D rotation group $SO(2)$. More formally, we define operations of each of these groups on a ligand pose $\mathbf{c} \in \mathbb{R}^{3n}$. The translation $A_{\text{tr}} : \mathbb{T}(3) \times \mathbb{R}^{3n} \rightarrow \mathbb{R}^{3n}$ is defined straightforwardly as $A_{\text{tr}}(\mathbf{r}, \mathbf{x})_i = \mathbf{x}_i + \mathbf{r}$ using the isomorphism $\mathbb{T}(3) \cong \mathbb{R}^3$ where $\mathbf{x}_i \in \mathbb{R}^3$ is the position of the $i$th atom. Similarly, the rotation $A_{\text{rot}} : SO(3) \times \mathbb{R}^{3n} \rightarrow \mathbb{R}^{3n}$ is defined by $A_{\text{rot}}(R, \mathbf{x})_i = R(\mathbf{x}_i - \bar{\mathbf{x}}) + \bar{\mathbf{x}}$ where $\bar{\mathbf{x}} = \frac{1}{n} \sum \mathbf{x}_i$, corresponding to rotations around the (unweighted) center of mass of the ligand.

Many valid definitions of a change in torsion angles are possible, as the torsion angle around any bond $(a_i, b_i)$ can be updated by rotating the $a_i$ side, the $b_i$ side, or both. However, we can specify changes of torsion angles to be *disentangled* from rotations or translations. To this end, we define the operation of elements of $SO(2)^m$ such that it causes a minimal perturbation (in an RMSD sense) to the structure:[5]

**Definition.** *Let $B_{k,\theta_k}(\mathbf{x}) \in \mathbb{R}^{3n}$ be any valid torsion update by $\theta_k$ around the $k$th rotatable bond $(a_k, b_k)$. We define $A_{tor} : SO(2)^m \times \mathbb{R}^{3n} \rightarrow \mathbb{R}^{3n}$ such that*

$$A_{tor}(\boldsymbol{\theta}, \mathbf{x}) = \text{RMSDAlign}(\mathbf{x}, (B_{1,\theta_1} \circ \cdots B_{m,\theta_m})(\mathbf{x}))$$

*where $\boldsymbol{\theta} = (\theta_1, \ldots \theta_m)$ and*

$$\text{RMSDAlign}(\mathbf{x}, \mathbf{x}') = \underset{\mathbf{x}^\dagger \in \{g\mathbf{x}' | g \in SE(3)\}}{\arg\min} \text{RMSD}(\mathbf{x}, \mathbf{x}^\dagger) \tag{1}$$

---

[4]RDKit ETKDG is a popular method for predicting the seed conformation. Although the structures may not be predicted perfectly, the errors lie largely in the torsion angles, which are resampled anyways.

[5]Since we do not define or use the composition of elements of $SO(2)^m$, strictly speaking, it is a product *space* but not a *group* and can be alternatively thought of as the torus $\mathbb{T}^m$ with an origin element.

This means that we apply all the $m$ torsion updates in any order and then perform a global RMSD alignment with the unmodified pose. The definition is motivated by ensuring that the infinitesimal effect of a torsion is orthogonal to any rototranslation, i.e., it induces no *linear or angular momentum*. These properties can be stated more formally as follows (proof in Appendix A):

**Proposition 1.** *Let* $\mathbf{y}(t) := A_{tor}(t\boldsymbol{\theta}, \mathbf{x})$ *for some* $\boldsymbol{\theta}$ *and where* $t\boldsymbol{\theta} = (t\theta_1, \ldots t\theta_m)$. *Then the linear and angular momentum are zero:* $\frac{d}{dt}\bar{\mathbf{y}}|_{t=0} = 0$ *and* $\sum_i (\mathbf{x} - \bar{\mathbf{x}}) \times \frac{d}{dt}\mathbf{y}_i|_{t=0} = 0$ *where* $\bar{\mathbf{x}} = \frac{1}{n}\sum_i \mathbf{x}_i$.

Now consider the product space[6] $\mathbb{P} = \mathbb{T}^3 \times SO(3) \times SO(2)^m$ and define $A : \mathbb{P} \times \mathbb{R}^{3n} \to \mathbb{R}^{3n}$ as

$$A((\mathbf{r}, R, \boldsymbol{\theta}), \mathbf{x}) = A_{\text{tr}}(\mathbf{r}, A_{\text{rot}}(R, A_{\text{tor}}(\boldsymbol{\theta}, \mathbf{x}))) \tag{2}$$

These definitions collectively provide the sought-after product space corresponding to the docking degrees of freedom. Indeed, for a seed ligand conformation $\mathbf{c}$, we can formally define the space of ligand poses $\mathcal{M}_{\mathbf{c}} = \{A(g, \mathbf{c}) \mid g \in \mathbb{P}\}$. This corresponds precisely to the intuitive notion of the space of ligand poses that can be reached by rigid-body motion plus torsion angle flexibility.

## 4.3 Diffusion on the Product Space

We now proceed to show how the product space can be used to learn a DGM over ligand poses in $\mathcal{M}_{\mathbf{c}}$. First, we need a theoretical result (proof in Appendix A):

**Proposition 2.** *For a given seed conformation* $\mathbf{c}$*, the map* $A(\cdot, \mathbf{c}) : \mathbb{P} \to \mathcal{M}_{\mathbf{c}}$ *is a bijection.*

which means that the inverse $A_{\mathbf{c}}^{-1} : \mathcal{M}_{\mathbf{c}} \to \mathbb{P}$ given by $A(g, \mathbf{c}) \mapsto g$ maps ligand poses $\mathbf{x} \in \mathcal{M}_{\mathbf{c}}$ to points on the product space $\mathbb{P}$. We are now ready to develop a diffusion process on $\mathbb{P}$.

De Bortoli et al. [2022] established that the DGM framework transfers straightforwardly to Riemannian manifolds with the score and score model as elements of the tangent space and with the geodesic random walk as the reverse SDE solver. Further, the score model can be trained in the standard manner with denoising score matching [Song & Ermon, 2019]. Thus, to implement a diffusion model on $\mathbb{P}$, it suffices to develop a method for sampling from and computing the score of the diffusion kernel on $\mathbb{P}$. Furthermore, since $\mathbb{P}$ is a product manifold, the forward diffusion proceeds independently in each manifold [Rodolà et al., 2019], and the tangent space is a direct sum: $T_g\mathbb{P} = T_{\mathbf{r}}\mathbb{T}_3 \oplus T_R SO(3) \oplus T_{\boldsymbol{\theta}}SO(2)^m \cong \mathbb{R}^3 \oplus \mathbb{R}^3 \oplus \mathbb{R}^m$ where $g = (\mathbf{r}, R, \boldsymbol{\theta})$. Thus, it suffices to sample from the diffusion kernel and regress against its score in each group independently.

In all three groups, we define the forward SDE as $d\mathbf{x} = \sqrt{d\sigma^2(t)/dt}\, d\mathbf{w}$ where $\sigma^2 = \sigma_{\text{tr}}^2, \sigma_{\text{rot}}^2$, or $\sigma_{\text{tor}}^2$ for $\mathbb{T}(3)$, $SO(3)$, and $SO(2)^m$ respectively and where $\mathbf{w}$ is the corresponding Brownian motion. Since $\mathbb{T}(3) \cong \mathbb{R}^3$, the translational case is trivial and involves sampling and computing the score of a standard Gaussian with variance $\sigma^2(t)$. The diffusion kernel on $SO(3)$ is given by the $IGSO(3)$ distribution [Nikolayev & Savyolov, 1970; Leach et al., 2022], which can be sampled in the axis-angle parameterization by sampling a unit vector $\hat{\boldsymbol{\omega}} \in \mathfrak{so}(3)$ uniformly[7] and random angle $\omega \in [0, \pi]$ according to

$$p(\omega) = \frac{1 - \cos\omega}{\pi}f(\omega) \quad \text{where} \quad f(\omega) = \sum_{l=0}^{\infty}(2l + 1)\exp(-l(l + 1)\sigma^2/2)\frac{\sin((l + 1/2)\omega)}{\sin(\omega/2)} \tag{3}$$

Further, the score of the diffusion kernel is $\nabla \ln p_t(R' \mid R) = (\frac{d}{d\omega}\log f(\omega))\hat{\omega} \in T_{R'}SO(3)$, where $R' = \mathbf{R}(\omega\hat{\omega})R$ is the result of applying Euler vector $\omega\hat{\omega}$ to $R$ [Yim et al., 2023]. The score computation and sampling can be accomplished efficiently by precomputing the truncated infinite series and interpolating the CDF of $p(\omega)$, respectively. Finally, the $SO(2)^m$ group is diffeomorphic to the torus $\mathbb{T}^m$, on which the diffusion kernel is a *wrapped normal distribution* with variance $\sigma^2(t)$. This can be sampled directly, and the score can be precomputed as a truncated infinite series [Jing et al., 2022].

## 4.4 Training and Inference

**Diffusion model.** Although we have defined the diffusion kernel and score matching objectives on $\mathbb{P}$, we nevertheless develop the training and inference procedures to operate on ligand poses in 3D

---

[6]Since we never compose elements of $\mathbb{P}$, we do not need to define a group structure.

[7]$\mathfrak{so}(3)$ is the tangent space of $SO(3)$ at the identity and is the space of Euler (or rotation) vectors, which are equivalent to the axis-angle parameterization.

coordinates directly. Providing the full 3D structure, rather than abstract elements of the product space, to the score model allows it to reason about physical interactions using $SE(3)$ equivariant models, not be dependent on arbitrary definitions of torsion angles [Jing et al., 2022], and better generalize to unseen complexes. In Appendix B, we present the training and inference procedures and discuss how we resolve their dependence on the choice of seed conformation $\mathbf{c}$ used to define the mapping between $\mathcal{M}_{\mathbf{c}}$ and the product space.

**Confidence model.** In order to collect training data for the confidence model $\mathbf{d}(\mathbf{x}, \mathbf{y})$, we run the trained diffusion model to obtain a set of candidate poses for every training example and generate labels by testing whether or not each pose has RMSD below 2Å. The confidence model is then trained with cross-entropy loss to correctly predict the binary label for each pose. During inference, the diffusion model is run to generate $N$ poses in parallel, which are passed to the confidence model that ranks them based on its confidence that they have RMSD below 2Å.

### 4.5 Model Architecture

We construct the score model $\mathbf{s}(\mathbf{x}, \mathbf{y}, t)$ and the confidence model $\mathbf{d}(\mathbf{x}, \mathbf{y})$ to take as input the current ligand pose $\mathbf{x}$ and protein structure $\mathbf{y}$ in 3D space. The output of the confidence model is a single scalar that is $SE(3)$-invariant (with respect to joint rototranslations of $\mathbf{x}, \mathbf{y}$) as lig- and pose distributions are defined relative to the protein structure, which can have arbitrary location and orientation. On the other hand, the output of the score model must be in the tangent space $T_{\mathbf{r}}\mathbb{T}_3 \oplus T_R SO(3) \oplus T_{\boldsymbol{\theta}} SO(2)^m$. The space $T_{\mathbf{r}}\mathbb{T}_3 \cong \mathbb{R}^3$ corresponds to translation vectors and $T_R SO(3) \cong \mathbb{R}^3$ to rotation (Euler) vectors, both of which are $SE(3)$-equivariant. Finally, $T_{\boldsymbol{\theta}} SO(2)^m$ corresponds to scores on $SE(3)$-invariant quantities (torsion angles). Thus, the score model must predict two $SE(3)$-equivariant vectors for the ligand as a whole and an $SE(3)$-invariant scalar at each of the $m$ freely rotatable bonds.

The score model and confidence model have similar architectures based on $SE(3)$-equivariant convolutional networks over point clouds [Thomas et al., 2018; Geiger et al., 2020]. However, the score model operates on a coarse-grained representation of the protein with $\alpha$-carbon atoms, while the confidence model has access to the all-atom structure. This multiscale setup yields improved performance and a significant speed-up w.r.t. doing the whole process at the atomic scale. The architectural components are summarized below and detailed in Appendix C.

Structures are represented as heterogeneous geometric graphs formed by ligand atoms, protein residues, and (for the confidence model) protein atoms. Residue nodes receive as initial features language model embeddings trained on protein sequences [Lin et al., 2022]. Nodes are sparsely connected based on distance cutoffs that depend on the types of nodes being linked and on the diffusion time. The ligand atom representations after the final interaction layer are then used to produce the different outputs. To produce the two $\mathbb{R}^3$ vectors representing the translational and rotational scores, we convolve the node representations with a tensor product filter placed at the center of mass. For the torsional score, we use a pseudotorque convolution to obtain a scalar at each rotatable bond of the ligand analogously to Jing et al. [2022], with the distinction that, since the score model operates on coarse-grained representations, the output is not a pseudoscalar (its parity is neither odd nor even). For the confidence model, the single scalar output is produced by mean-pooling the ligand atoms' scalar representations followed by a fully connected layer.

## 5 Experiments

**Experimental setup.** We evaluate our method on the complexes from PDBBind [Liu et al., 2017], a large collection of protein-ligand structures collected from PDB [Berman et al., 2003], which was used with time-based splits to benchmark many previous works [Stärk et al., 2022; Volkov et al., 2022; Lu et al., 2022]. We compare DiffDock with state-of-the-art search-based methods SMINA [Koes et al., 2013], QuickVina-W [Hassan et al., 2017], GLIDE [Halgren et al., 2004], and GNINA [McNutt et al., 2021] as well as the older Autodock Vina [Trott & Olson, 2010], and the recent deep learning methods EquiBind and TANKBind presented above. Extensive details about the experimental setup, data, baselines, and implementation are in Appendix D.4 and all code is available at `https://github.com/gcorso/DiffDock`. The repository also contains videos of the reverse diffusion process (images of the same are in Figure 14).

Table 1: **PDBBind blind docking.** All methods receive a small molecule and are tasked to find its binding location, orientation, and conformation. Shown is the percentage of predictions with RMSD < 2Å and the median RMSD (see Appendix D.3). The top half contains methods that directly find the pose; the bottom half those that use a pocket prediction method. The last two lines show our method's performance. In parenthesis, we specify the number of poses sampled from the generative model. * indicates that the method runs exclusively on CPU, "-" means not applicable; some cells are empty due to infrastructure constraints. For TANKBind, the runtimes for the top-1 and top-5 predictions are different. No method received further training or tuning on ESMFold structures. Further evaluation details are in Appendix D.4 and further metrics are reported in Appendix F.

| | Holo crystal proteins | | | | Apo ESMFold proteins | | | | |
| | Top-1 RMSD | | Top-5 RMSD | | Top-1 RMSD | | Top-5 RMSD | | Average |
| Method | %<2 | Med. | %<2 | Med. | %<2 | Med. | %<2 | Med. | Runtime (s) |
|---|---|---|---|---|---|---|---|---|---|
| GNINA | 22.9 | 7.7 | 32.9 | 4.5 | 2.0 | 22.3 | 4.0 | 14.22 | 127 |
| SMINA | 18.7 | 7.1 | 29.3 | 4.6 | 3.4 | 15.4 | 6.9 | 10.0 | 126* |
| GLIDE | 21.8 | 9.3 | | | | | | | 1405* |
| EQUIBIND | 5.5 | 6.2 | - | - | 1.7 | 7.1 | - | - | 0.04 |
| TANKBIND | 20.4 | 4.0 | 24.5 | 3.4 | 10.4 | 5.4 | 14.7 | 4.3 | 0.7/2.5 |
| P2RANK+SMINA | 20.4 | 6.9 | 33.2 | 4.4 | 4.6 | 10.0 | 10.3 | 7.0 | 126* |
| P2RANK+GNINA | 28.8 | 5.5 | 38.3 | 3.4 | 8.6 | 11.2 | 12.8 | 7.2 | 127 |
| EQUIBIND+SMINA | 23.2 | 6.5 | 38.6 | 3.4 | 4.3 | 8.3 | 11.7 | 5.8 | 126* |
| EQUIBIND+GNINA | 28.8 | 4.9 | 39.1 | 3.1 | 10.2 | 8.8 | 18.6 | 5.6 | 127 |
| **DIFFDOCK (10)** | 35.0 | 3.6 | 40.7 | 2.65 | **21.7** | **5.0** | **31.9** | **3.3** | 10 |
| **DIFFDOCK (40)** | **38.2** | **3.3** | **44.7** | **2.40** | 20.3 | 5.1 | 31.3 | **3.3** | 40 |

As we are evaluating blind docking, the methods receive two inputs: the ligand with a predicted seed conformation (e.g., from RDKit) and the crystal structure of the protein. Since search-based methods work best when given a starting binding pocket to restrict the search space, we also test the combination of using an ML-based method, such as P2Rank [Krivák & Hoksza, 2018] (also used by TANKBind) or EquiBind to find an initial binding pocket, followed by a search-based method to predict the exact pose in the pocket. To evaluate the generated poses, we compute the heavy-atom RMSD (permutation symmetry corrected) between the predicted and the ground-truth ligand when the protein structures are aligned. Since all methods except for EquiBind generate multiple ranked structures, we report the metrics for the highest-ranked prediction as the top-1 prediction and top-5 refers to selecting the most accurate pose out of the 5 highest ranked predictions; a useful metric when multiple predictions are used for downstream tasks.

**Apo-structure docking setup.** In order to test the performance of the models against computationally generated apo-structures, we run ESMFold [Lin et al., 2022] on the proteins of the test-set of PDBBind and align its predictions with the crystal holo-structure. In order to align structure changing as little as possible to the conformation of the binding site, we use in the alignment of the residues a weight exponential with the distance to the ligand. More details on this alignment are provided in Appendix D.2. The docking methods are then run against the generated structures and compared with the ligand pose inferred by the alignment.

**Docking accuracy.** DIFFDOCK significantly outperforms all previous methods (Table 1). In particular, DIFFDOCK obtains an impressive 38.2% top-1 success rate (i.e., percentage of predictions with RMSD <2Å[8]) when sampling 40 poses and 35.0% when sampling just 10. This performance vastly surpasses that of state-of-the-art commercial software such as GLIDE (21.8%, $p=2.7 \times 10^{-7}$) and the previous state-of-the-art deep learning method TANKBind (20.4%, $p=1.0 \times 10^{-12}$). The use of ML-based pocket prediction in combination with search-based docking methods improves over the baseline performances, but even the best of these (EquiBind+GNINA) reaches a success rate of only 28.8% ($p=0.0003$). Unlike regression methods like EquiBind, DIFFDOCK is able to provide multiple diverse predictions of different likely poses, as highlighted in the top-5 performances.

**Apo-structure docking accuracy.** Previous work Wong et al. [2022], that highlighted that existing docking methods are not suited to dock to (computational) apo-structures, are validated by the results

---

[8]Most commonly used evaluation metric [Alhossary et al., 2015; Hassan et al., 2017; McNutt et al., 2021]

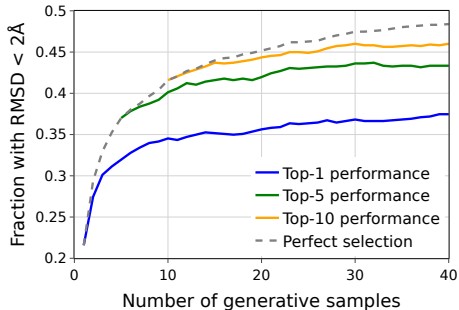 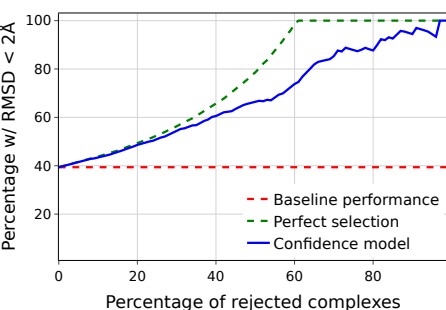

Figure 3: **Left:** DIFFDOCK's performance as a function of the number of samples from the generative model. "Perfect selection" refers to choosing the sample with the lowest RMSD. **Right:** Selective accuracy: percentage of predictions with RMSD below 2Å when only making predictions for the portion of the dataset where DIFFDOCK is most confident.

in Table 1 where previous methods obtain top-1 accuracies of only 10% or below. This is most likely due to their reliance on trying to find key-lock matches, which makes them inflexible to imperfect protein structures. Even when built-in options allowing side-chain flexibility are activated, the results do not improve (see Appendix 8). Meanwhile, DIFFDOCK is able to retain a larger proportion of its accuracy, placing the top-ranked ligand below 2Å away on 22% of the complexes. This ability to generalize to imperfect structures, even without retraining, can be attributed to a combination of (1) the robustness of the diffusion model to small perturbations in the backbone atoms, and (2) the fact that DIFFDOCK does not use the exact position of side chains in the score model and is therefore forced to implicitly model their flexibility.

**Inference runtime.** DIFFDOCK holds its superior accuracy while being (on GPU) 3 to 12 times faster than the best search-based method, GNINA (Table 1). This high speed is critical for applications such as high throughput virtual screening for drug candidates or reverse screening for protein targets, where one often searches over a vast number of complexes. As a diffusion model, DIFF-DOCK is inevitably slower than the one-shot deep learning method EQUIBIND, but as shown in Figure 3-*left* and Appendix F.3, it can be significantly sped up without significant loss of accuracy.

**Selective accuracy of confidence score.** As the top-1 results show, DIFFDOCK's confidence model is very accurate in ranking the sampled poses for a given complex and picking the best one. We also investigate the *selective accuracy* of the confidence model across *different* complexes by evaluating how DIFFDOCK's accuracy increases if it only makes predictions when the confidence is above a certain threshold, known as *selective prediction*. In Figure 3-*right*, we plot the success rate as we decrease the percentage of complexes for which we make predictions, i.e., increase the confidence threshold. When only making predictions for the top one-third of complexes in terms of model confidence, the success rate improves from 38% to 83%. Additionally, there is a Spearman correlation of 0.68 between DIFFDOCK's confidence and the negative RMSD. Thus, the confidence score is a good indicator of the quality of DIFFDOCK's top-ranked sampled pose and provides a highly valuable confidence measure for downstream applications.

## 6 CONCLUSION

We presented DIFFDOCK, a diffusion generative model tailored to the task of molecular docking. This represents a paradigm shift from previous deep learning approaches, which use regression-based frameworks, to a generative modeling approach that is better aligned with the objective of molecular docking. To produce a fast and accurate generative model, we designed a diffusion process over the manifold describing the main degrees of freedom of the task via ligand pose transformations spanning the manifold. Empirically, DIFFDOCK outperforms the state-of-the-art by large margins on PDBBind, has fast inference times, and provides confidence estimates with high selective accuracy. Moreover, unlike previous methods, it retains a large proportion of its accuracy when run on computationally folded protein structures. Thus, DIFFDOCK can offer great value for many existing real-world pipelines and opens up new avenues of research on how to best integrate downstream tasks, such as affinity prediction, into the framework and apply similar ideas to protein-protein and protein-nucleic acid docking.

## ACKNOWLEDGMENTS

We pay tribute to Octavian-Eugen Ganea (1987-2022), dear colleague, mentor, and friend without whom this work would have never been possible.

We thank Rachel Wu, Jeremy Wohlwend, Felix Faltings, Jason Yim, Victor Quach, Saro Passaro, Patrick Walters, Michael Heinzinger, Mario Geiger, Michael John Arcidiacono, Noah Getz, and John Yang for valuable feedback and insightful discussions. We thank Wei Lu for his help with running TANKBind. We thank Valentin De Bortoli, Emile Mathieu, Brian Trippe, and Jason Yim for the critical discussions and help to formalize diffusion on $SO(3)$. Noah Getz contributed to the implementation of the apo-structures alignment procedure.

This work was supported by the Machine Learning for Pharmaceutical Discovery and Synthesis (MLPDS) consortium, the Abdul Latif Jameel Clinic for Machine Learning in Health, the DTRA Discovery of Medical Countermeasures Against New and Emerging (DOMANE) threats program, the DARPA Accelerated Molecular Discovery program and the Sanofi Computational Antibody Design grant. Bowen Jing acknowledges the support from the Department of Energy Computational Science Graduate Fellowship.

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

# A PROOFS

## A.1 PROOF OF PROPOSITION 1

**Proposition 1.** *Let* $\mathbf{y}(t) := A_{tor}(t\boldsymbol{\theta}, \mathbf{x})$ *for some* $\boldsymbol{\theta}$ *and where* $t\boldsymbol{\theta} = (t\theta_1, \ldots t\theta_m)$. *Then the linear and angular momentum are zero:* $\frac{d}{dt}\bar{\mathbf{y}}|_{t=0} = 0$ *and* $\sum_i (\mathbf{x} - \bar{\mathbf{x}}) \times \frac{d}{dt}\mathbf{y}_i|_{t=0} = 0$ *where* $\bar{\mathbf{x}} = \frac{1}{n}\sum_i \mathbf{x}_i$.

*Proof.* Let $\mathbf{y}(t) = R(t)(B(t, \boldsymbol{\theta}, \mathbf{x}) - \bar{\mathbf{x}}) + \bar{\mathbf{x}} + \mathbf{p}(t)$ where $B(t, \boldsymbol{\theta}, \cdot) = B_{1,t\theta_1} \circ \cdots B_{m,t\theta_m}$ and $R(t), \mathbf{p}(t)$ are the rotation (around $\bar{\mathbf{x}}$) and translation associated with the optimal RMSD alignment between $B(t, \boldsymbol{\theta}, \mathbf{x})$ and $\mathbf{x}$. By definition of RMSD, for any $t$, $R(t)$ and $\mathbf{p}(t)$ minimize

$$||\mathbf{y}(t) - \mathbf{x}|| = ||R(t)(B(t, \boldsymbol{\theta}, \mathbf{x}) - \bar{\mathbf{x}}) + \bar{\mathbf{x}} + \mathbf{p}(t) - \mathbf{x}|| \quad (4)$$

For infinitesimal $t = dt$, the RHS becomes

$$\text{RHS} = ||R(dt)(B(dt, \boldsymbol{\theta}, \mathbf{x}) - \bar{\mathbf{x}}) + \bar{\mathbf{x}} + \mathbf{p}(dt) - \mathbf{x}||$$
$$= ||(R'(0)\,dt + R(0))\,(B'(0, \boldsymbol{\theta}, \mathbf{x})\,dt + B(0, \boldsymbol{\theta}, \mathbf{x}) - \bar{\mathbf{x}}) + \bar{\mathbf{x}} + \mathbf{p}'(0)\,dt + \mathbf{p}(0) - \mathbf{x}|| \quad (5)$$
$$= ||R'(0)\,(\mathbf{x} - \bar{\mathbf{x}}) + B'(0, \boldsymbol{\theta}, \mathbf{x}) + \mathbf{p}'(0)||\,dt$$

where we have used $R(0) = I$, $B(0, \boldsymbol{\theta}, \mathbf{x}) = \mathbf{x}$, and $\mathbf{p}(0) = 0$. Thus, we see that RMSD alignment implies that the derivatives of $R(t), \mathbf{p}(t)$ minimize the norm of

$$\mathbf{y}'(0) = R'(0)\,(\mathbf{x} - \bar{\mathbf{x}}) + B'(0, \boldsymbol{\theta}, \mathbf{x}) + \mathbf{p}'(0) \quad (6)$$

This expression represents the instantaneous velocity of the points $\mathbf{y}_i$ at $t = 0$. We now show that minimizing the velocity results in zero linear and angular momentum.

We abbreviate $B'(t, \boldsymbol{\theta}, \mathbf{x}(0))_i := \mathbf{b}_i$ and $\mathbf{p}' = \mathbf{v}$. Further, let $\mathbf{r}_i = \mathbf{x}_i - \bar{\mathbf{x}}$, such that the rotational contribution to the velocity can be written in terms of an angular velocity vector $\boldsymbol{\omega}$. With this, at $t = 0$ we have

$$\mathbf{y}_i' = \mathbf{b}_i + \boldsymbol{\omega} \times \mathbf{r}_i + \mathbf{v} \quad (7)$$

We thus obtain the squared norm as

$$\sum_i ||\mathbf{y}_i'||^2 = \sum_i (\mathbf{b}_i + \boldsymbol{\omega} \times \mathbf{r}_i + \mathbf{v}) \cdot (\mathbf{b}_i + \boldsymbol{\omega} \times \mathbf{r}_i + \mathbf{v})$$
$$= \sum_i \left[ ||\mathbf{b}_i||^2 + 2\mathbf{b}_i \cdot (\boldsymbol{\omega} \times \mathbf{r}_i) + 2\mathbf{b}_i \cdot \mathbf{v} + (\boldsymbol{\omega} \times \mathbf{r}_i) \cdot (\boldsymbol{\omega} \times \mathbf{r}_i) + 2(\boldsymbol{\omega} \times \mathbf{r}_i) \cdot \mathbf{v} + ||\mathbf{v}||^2 \right]$$
$$= \sum_i ||\mathbf{b}_i||^2 + 2\boldsymbol{\omega} \cdot \sum_i (\mathbf{r}_i \times \mathbf{b}_i) + 2\left(\sum_i \mathbf{b}_i\right) \cdot \mathbf{v} + n\,||\mathbf{v}||^2 + \boldsymbol{\omega}^T \mathcal{I}(\mathbf{r}) \boldsymbol{\omega}$$
$$(8)$$

where we have used the fact that $\sum_i \mathbf{r}_i = 0$ and where $\mathcal{I}(\mathbf{r}) = \left(\sum_i \mathbf{r}_i \cdot \mathbf{r}_i\right) I - \sum_i \mathbf{r}_i \mathbf{r}_i^T$ is the $3 \times 3$ *inertia tensor*. To minimize the squared norm (and thus the norm itself), we set gradients with respect to $\mathbf{v}, \boldsymbol{\omega}$ to zero. This gives

$$\mathbf{v} = -\frac{1}{n}\sum_i \mathbf{b}_i \quad \text{and} \quad \boldsymbol{\omega} = -\mathcal{I}(\mathbf{r})^{-1}\left(\sum_i \mathbf{r}_i \times \mathbf{b}_i\right) \quad (9)$$

Now with $\mathbf{y}_i' = \mathbf{b}_i + \boldsymbol{\omega} \times \mathbf{r}_i + \mathbf{v}$ we evaluate the linear momentum

$$\frac{1}{n}\sum_i \mathbf{y}_i' = \frac{1}{n}\left(\sum_i \mathbf{b}_i + \boldsymbol{\omega} \times \sum_i \mathbf{r}_i + n\mathbf{v}\right) = 0 \quad (10)$$

which is zero by direct substitution of $\mathbf{v}$. Similarly, we evaluate the angular momentum

$$\sum_i \mathbf{r}_i \times \mathbf{y}_i' = \sum_i \mathbf{r}_i \times \mathbf{b}_i + \sum_i \mathbf{r}_i \times (\boldsymbol{\omega} \times \mathbf{r}_i) + \sum_i \mathbf{r}_i \times \mathbf{v}$$
$$= \sum_i \mathbf{r}_i \times \mathbf{b}_i + \mathcal{I}(\mathbf{r})\boldsymbol{\omega} = 0$$
$$(11)$$

which is zero by direct substitution of $\boldsymbol{\omega}$. Thus, the linear and angular momentum are zero at $t = 0$ for arbitrary $\mathbf{x}$. $\qquad\square$

Note that since we did not use the particular form of $B(t\boldsymbol{\theta}, \mathbf{x})$ in the above proof, we have shown that RMSD alignment can be used to disentangle rotations and translations from the infinitesimal action of any arbitrary function.

## A.2 PROOF OF PROPOSITION 2

**Proposition 2.** *For a given seed conformation* $\mathbf{c}$*, the map* $A(\cdot, \mathbf{c}) : \mathbb{P} \to \mathcal{M}_{\mathbf{c}}$ *is a bijection.*

*Proof.* Since we defined $\mathcal{M}_{\mathbf{c}} = \{A(g, \mathbf{c}) \mid g \in \mathbb{P}\}$, $A(\cdot, \mathbf{c})$ is automatically surjective. We now show that it is injective. Assume for the sake of contradiction that $A(\cdot, \mathbf{c})$ is not injective, so that there exist elements of the product space $g_1, g_2 \in \mathbb{P}$ with $g_1 \neq g_2$ but with $A(g_1, \mathbf{c}) = A(g_2, \mathbf{c}) = \mathbf{c}'$. That is,

$$A_{\text{tr}}(\mathbf{r}_1, A_{\text{rot}}(R_1, A_{\text{tor}}(\boldsymbol{\theta}_1, \mathbf{c}))) = A_{\text{tr}}(\mathbf{r}_2, A_{\text{rot}}(R_2, A_{\text{tor}}(\boldsymbol{\theta}_2, \mathbf{c}))) \tag{12}$$

which we abbreviate as $\mathbf{c}^{(1)} = \mathbf{c}^{(2)}$. Since only $A_{\text{tr}}$ changes the center of mass $\sum_i \mathbf{c}_i/n$, we have $\sum_i \mathbf{c}_i^{(1)}/n = \sum_i \mathbf{c}_i/n + \mathbf{r}_1$ and $\sum_i \mathbf{c}_i^{(2)}/n = \sum_i \mathbf{c}_i/n + \mathbf{r}_2$. However, since $\mathbf{c}^{(1)} = \mathbf{c}^{(2)}$, this implies $\mathbf{r}_1 = \mathbf{r}_2$. Next, consider the torsion angles $\boldsymbol{\tau}_1 = (\tau_1^{(1)}, \dots \tau_m^{(1)})$ of $\mathbf{c}^{(1)}$ corresponding to some choice of dihedral angles at each rotatable bond. Because $A_{\text{tr}}$ and $A_{\text{rot}}$ are rigid-body motions, only $A_{\text{tor}}$ changes the dihedral angles; in particular, by definition we have $\tau_i^{(1)} \cong \tau_i + \theta_i^{(1)}$ mod $2\pi$ and $\tau_i^{(2)} \cong \tau_i + \theta_i^{(2)}$ mod $2\pi$ for all $i = 1, \dots m$. However, because $\tau_i^{(1)} = \tau_i^{(2)}$, this means $\theta_i^{(1)} \cong \theta_i^{(2)}$ for all $i$ and therefore $\boldsymbol{\theta}_1 = \boldsymbol{\theta}_2$ (as elements of $SO(2)^m$). Now denote $\mathbf{c}^{\star} = A_{\text{tor}}(\boldsymbol{\theta}_1, \mathbf{c}) = A_{\text{tor}}(\boldsymbol{\theta}_2, \mathbf{c})$ and apply $A_{\text{tr}}(-\mathbf{r}_1, \cdot) = A_{\text{tr}}(-\mathbf{r}_2, \cdot)$ to both sides of Equation 12. We then have

$$A_{\text{rot}}(R_1, \mathbf{c}^{\star}) = A_{\text{rot}}(R_2, \mathbf{c}^{\star}) \tag{13}$$

which further leads to

$$\mathbf{c}^{\star} - \bar{\mathbf{c}}^{\star} = R_1^{-1} R_2 (\mathbf{c}^{\star} - \bar{\mathbf{c}}^{\star}) \tag{14}$$

In general, this does not imply that $R_1 = R_2$. However, $R_1 \neq R_2$ is possible only if $\mathbf{c}^{\star}$ is *degenerate*, in the sense that all points are collinear along the shared axis of rotation of $R_1, R_2$. However, in practice, conformers never consist of a collinear set of points, so we can safely assume $R_1 = R_2$. We now have $(\mathbf{r}_1, R_1, \boldsymbol{\theta}_1) = (\mathbf{r}_2, R_2, \boldsymbol{\theta}_2)$, or $g_1 = g_2$, contradicting our initial assumption. We thus conclude that $A(\cdot, \mathbf{c})$ is injective, completing the proof. □

## B  TRAINING AND INFERENCE

In this section we present the training and inference procedures of the diffusion generative model. First, however, there are a few subtleties of the generative approach to molecular docking that are worth mentioning. Unlike the standard generative modeling setting where the dataset consists of many samples drawn from the data distribution, each training example $(\mathbf{x}^{\star}, \mathbf{y})$ of protein structure $\mathbf{y}$ and ground-truth ligand pose $\mathbf{x}^{\star}$ is the *only sample* from the corresponding conditional distribution $p_{\mathbf{x}^{\star}}(\cdot \mid \mathbf{y})$ defined over $\mathcal{M}_{\mathbf{x}^{\star}}$. Thus, the innermost training loop iterates over distinct conditional distributions $p_{\mathbf{x}^{\star}}(\cdot \mid \mathbf{y})$, along with a single sample from that distribution, rather than over samples from a common data distribution $p_{\text{data}}(\mathbf{x})$.

As discussed in Section 4, during inference, $\mathbf{c}$ is the ligand structure generated with a method such as RDKit. However, during training we require $\mathcal{M}_{\mathbf{c}} = \mathcal{M}_{\mathbf{x}^{\star}}$ in order to define a bijection between $\mathbf{c} \in \mathcal{M}_{\mathbf{x}^{\star}}$ and $\mathbb{P}$. If we take $\mathbf{c} \in \mathcal{M}_{\mathbf{x}^{\star}}$, there will be a distribution shift between the manifolds $\mathcal{M}_{\mathbf{c}}$ considered at training time and those considered at inference time. To circumvent this issue, at training time we predict $\mathbf{c}$ with RDKit and replace $\mathbf{x}^{\star}$ with $\arg\min_{\mathbf{x}^{\dagger} \in \mathcal{M}_{\mathbf{c}}} \text{RMSD}(\mathbf{x}^{\star}, \mathbf{x}^{\dagger})$ using the conformer matching procedure described in Jing et al. [2022].

The above paragraph may be rephrased more intuitively as follows: during inference, the generative model docks a ligand structure generated by RDKit, keeping its non-torsional degrees of freedom (e.g., local structures) fixed. At training time, however, if we train the score model with the local structures of the ground truth pose, this will not correspond to the local structures seen at inference time. Thus, at training time, we replace the ground truth pose by generating a ligand structure with RDKit and aligning it to the ground truth pose while keeping the local structures fixed.

With these preliminaries, we now continue to the full procedures (Algorithms 1 and 2). The training and inference procedures of a score-based diffusion generative model on a Riemannian manifold consist of (1) sampling and regressing against the score of the diffusion kernel during training; and (2) sampling a geodesic random walk with the score as a drift term during inference [De Bortoli et al., 2022]. Because we have developed the diffusion process on $\mathbb{P}$ but continue to provide the score

model with elements in $\mathcal{M}_{\mathbf{c}} \subset \mathbb{R}^{3n}$, the full training and inference procedures involve repeatedly interconverting between the two spaces using the bijection given by the seed conformation $\mathbf{c}$.

---

**Algorithm 1:** Training procedure (single epoch)

---

**Input:** Training pairs $\{(\mathbf{x}^\star, \mathbf{y})\}$, RDKit predictions $\{\mathbf{c}\}$
**foreach** $\mathbf{c}, \mathbf{x}^\star, \mathbf{y}$ **do**

> Let $\mathbf{x}_0 \leftarrow \arg\min_{\mathbf{x}^\dagger \in \mathcal{M}_{\mathbf{c}}} \text{RMSD}(\mathbf{x}^\star, \mathbf{x}^\dagger)$;
> Compute $(\mathbf{r}_0, R_0, \boldsymbol{\theta}_0) \leftarrow A_{\mathbf{c}}^{-1}(\mathbf{x}_0)$;
> Sample $t \sim \text{Uni}([0, 1])$;
> Sample $\Delta\mathbf{r}, \Delta R, \Delta\boldsymbol{\theta}$ from diffusion kernels $p_t^{\text{tr}}(\cdot \mid 0), p_t^{\text{rot}}(\cdot \mid 0), p_t^{\text{tor}}(\cdot \mid 0)$;
> Set $\mathbf{r}_t \leftarrow \mathbf{r}_0 + \Delta\mathbf{r}$;
> Set $R_t \leftarrow (\Delta R)R_0$;
> Set $\boldsymbol{\theta}_t \leftarrow \boldsymbol{\theta}_0 + \Delta\boldsymbol{\theta} \mod 2\pi$;
> Compute $\mathbf{x}_t \leftarrow A((\mathbf{r}_t, R_t, \boldsymbol{\theta}_t), \mathbf{c})$;
> Predict scores $\alpha \in \mathbb{R}^3, \beta \in \mathbb{R}^3, \gamma \in \mathbb{R}^m = \mathbf{s}(\mathbf{x}_t, \mathbf{c}, \mathbf{y}, t)$ ;
> Take optimization step on loss
> $\mathcal{L} = ||\alpha - \nabla \log p_t^{\text{tr}}(\Delta\mathbf{r} \mid 0)||^2 + ||\beta - \nabla \log p_t^{\text{rot}}(\Delta R \mid 0)||^2 + ||\gamma - \nabla \log p_t^{\text{tor}}(\Delta\boldsymbol{\theta} \mid 0)||^2$

---

**Algorithm 2:** Inference procedure

---

**Input:** RDKit prediction $\mathbf{c}$, protein structure $\mathbf{y}$ (both centered at origin)
**Output:** Sampled ligand pose $\mathbf{x}_0$
Sample $\boldsymbol{\theta}_N \sim \text{Uni}(SO(2)^m), R_N \sim \text{Uni}(SO(3)), \mathbf{r}_N \sim \mathcal{N}(0, \sigma_{\text{tor}}^2(T))$;
Let $\mathbf{x}_N = A((\mathbf{r}_N, R_N, \boldsymbol{\theta}_N), \mathbf{c})$;
**for** $n \leftarrow N$ **to** 1 **do**

> Let $t = n/N$ and $\Delta\sigma_{\text{tr}}^2 = \sigma_{\text{tr}}^2(n/N) - \sigma_{\text{tr}}^2((n-1)/N)$ and similarly for $\Delta\sigma_{\text{rot}}^2, \Delta\sigma_{\text{tor}}^2$;
> Predict scores $\alpha \in \mathbb{R}^3, \beta \in \mathbb{R}^3, \gamma \in \mathbb{R}^m \leftarrow \mathbf{s}(\mathbf{x}_n, \mathbf{c}, \mathbf{y}, t)$;
> Sample $\mathbf{z}_{\text{tr}}, \mathbf{z}_{\text{rot}}, \mathbf{z}_{\text{tor}}$ from $\mathcal{N}(0, \Delta\sigma_{\text{tr}}^2), \mathcal{N}(0, \Delta\sigma_{\text{rot}}^2), \mathcal{N}(0, \Delta\sigma_{\text{tor}}^2)$ respectively;
> Set $\mathbf{r}_{n-1} \leftarrow \mathbf{r}_n + \Delta\sigma_{\text{tr}}^2\alpha + \mathbf{z}_{\text{tr}}$;
> Set $R_{n-1} \leftarrow \mathbf{R}(\Delta\sigma_{\text{rot}}^2\beta + \mathbf{z}_{\text{rot}})R_n$;
> Set $\boldsymbol{\theta}_{n-1} \leftarrow \boldsymbol{\theta}_n + (\Delta\sigma_{\text{tor}}^2\gamma + \mathbf{z}_{\text{tor}}) \mod 2\pi$;
> Compute $\mathbf{x}_{n-1} \leftarrow A((\mathbf{r}_{n-1}, R_{n-1}, \boldsymbol{\theta}_{n-1}), \mathbf{c})$;

Return $\mathbf{x}_0$;

---

However, as noted in the main text, the dependence of these procedures on the exact choice of $\mathbf{c}$ is potentially problematic, as it suggests that at inference time, the model distribution may be different depending on the orientation and torsion angles of $\mathbf{c}$. Simply removing the dependence of the score model on $\mathbf{c}$ is not sufficient since the update steps themselves still occur on $\mathbb{P}$ and require a choice of $\mathbf{c}$ to be mapped to $\mathcal{M}_{\mathbf{c}}$. However, notice that the update steps—in both training and inference—consist of (1) sampling the diffusion kernels at the origin; (2) applying these updates to the point on $\mathbb{P}$; and (3) transferring the point on $\mathbb{P}$ to $\mathcal{M}_{\mathbf{c}}$ via $A(\cdot, \mathbf{c})$. Might it instead be possible to apply the updates to 3D ligand poses $\mathbf{x} \in \mathcal{M}_{\mathbf{c}}$ *directly*?

It turns out that the notion of applying these steps to ligand poses "directly" corresponds to the formal notion of *group action*. The operations $A_{\text{tr}}, A_{\text{rot}}, A_{\text{tor}}$ that we have already defined are formally group actions if they satisfy $A_{(\cdot)}(g_1 g_2, \mathbf{x}) = A(g_1, A(g_2, \mathbf{x}))$. While true for $A_{\text{tr}}, A_{\text{rot}}$, this is not generally true for $A_{\text{tor}}$ if we take $SO(2)^m$ to be the direct product group; however, the approximation is increasingly good as the magnitude of the torsion angle updates decreases. If we then define $\mathbb{P}$ to be the direct product group of its constituent groups, $A$ is a group action of $\mathbb{P}$ on $\mathcal{M}_{\mathbf{c}}$, as the operations of $A_{\text{tr}}, A_{\text{rot}}, A_{\text{tor}}$ commute and are (under the approximation) individually group actions.

The implication of $A$ being a group action can be seen as follows. Let $\delta = g_b g_a^{-1}$ be the update which brings $g_a \in \mathbb{P}$ to $g_b \in \mathbb{P}$ via left multiplication, and let $\mathbf{x}_a, \mathbf{x}_b$ be the corresponding ligand poses $A(g_a, \mathbf{c}), A(g_b, \mathbf{c})$. Then

$$\mathbf{x}_b = A(g_b g_a^{-1} g_a, \mathbf{c}) = A(\delta, \mathbf{x}_a) \tag{15}$$

which means that the updates $\delta$ can be applied directly to $\mathbf{x}_a$ using the operation $A$. The training and inference procedures then become Algorithm 3 and 4 below. The initial conformer $\mathbf{c}$ is no longer used, except in the initial steps to define the manifold—to find the closest point to $\mathbf{x}^\star$ in training, and to sample $\mathbf{x}_N$ from the prior over $\mathcal{M}_\mathbf{c}$ in inference.

Conceptually speaking, this procedure corresponds to "forgetting" the location of the origin element on $\mathcal{M}_\mathbf{c}$, which is permissible because a change of the origin to some equivalent seed $\mathbf{c}' \in \mathcal{M}_\mathbf{c}$ merely translates—via right multiplication by $A_\mathbf{c}^{-1}(\mathbf{c}')$—the original and diffused data distributions on $\mathbb{P}$, but does not cause any changes on $\mathcal{M}_\mathbf{c}$ itself. The training and inference routines involve updates—formally left multiplications—to group elements, but as left multiplication on the group corresponds to group actions on $\mathcal{M}_\mathbf{c}$, the updates can act on $\mathcal{M}_\mathbf{c}$ directly, without referencing the origin $\mathbf{c}$.

We find that the approximation of $A$ as a group action works quite well in practice and use Algorithms 3 and 4 for all training and experiments discussed in the paper. Of course, disentangling the torsion updates from rotations in a way that makes $A_{\text{tor}}$ exactly a group action would justify the procedure further, and we regard this as a possible direction for future work.

---

**Algorithm 3:** Approximate training procedure (single epoch)

---

**Input:** Training pairs $\{(\mathbf{x}^\star, \mathbf{y})\}$, RDKit predictions $\{\mathbf{c}\}$
**foreach** $\mathbf{c}, \mathbf{x}^\star, \mathbf{y}$ **do**
  Let $\mathbf{x}_0 \leftarrow \arg\min_{\mathbf{x}^\dagger \in \mathcal{M}_\mathbf{c}} \text{RMSD}(\mathbf{x}^\star, \mathbf{x}^\dagger)$;
  Sample $t \sim \text{Uni}([0, 1])$;
  Sample $\Delta\mathbf{r}, \Delta R, \Delta\boldsymbol{\theta}$ from diffusion kernels $p_t^{\text{tr}}(\cdot \mid 0), p_t^{\text{rot}}(\cdot \mid 0), p_t^{\text{tor}}(\cdot \mid 0)$;
  Compute $\mathbf{x}_t \leftarrow A((\Delta\mathbf{r}, \Delta R, \Delta\boldsymbol{\theta}), \mathbf{x}_0)$;
  Predict scores $\alpha \in \mathbb{R}^3, \beta \in \mathbb{R}^3, \gamma \in \mathbb{R}^m = \mathbf{s}(\mathbf{x}_t, \mathbf{y}, t)$ ;
  Take optimization step on loss
  $\mathcal{L} = ||\alpha - \nabla \log p_t^{\text{tr}}(\Delta\mathbf{r} \mid 0)||^2 + ||\beta - \nabla \log p_t^{\text{rot}}(\Delta R \mid 0)||^2 + ||\gamma - \nabla \log p_t^{\text{tor}}(\Delta\boldsymbol{\theta} \mid 0)||^2$

---

**Algorithm 4:** Approximate inference procedure

---

**Input:** RDKit prediction $\mathbf{c}$, protein structure $\mathbf{y}$ (both centered at origin)
**Output:** Sampled ligand pose $\mathbf{x}_0$
Sample $\boldsymbol{\theta}_N \sim \text{Uni}(SO(2)^m)$, $R_N \sim \text{Uni}(SO(3))$, $\mathbf{r}_N \sim \mathcal{N}(0, \sigma_{\text{tor}}^2(T))$;
Let $\mathbf{x}_N = A((\mathbf{r}_N, R_N, \boldsymbol{\theta}_N), \mathbf{c})$;
**for** $n \leftarrow N$ **to** 1 **do**
  Let $t = n/N$ and $\Delta\sigma_{\text{tr}}^2 = \sigma_{\text{tr}}^2(n/N) - \sigma_{\text{tr}}^2((n-1)/N)$ and similarly for $\Delta\sigma_{\text{rot}}^2, \Delta\sigma_{\text{tor}}^2$;
  Predict scores $\alpha \in \mathbb{R}^3, \beta \in \mathbb{R}^3, \gamma \in \mathbb{R}^m \leftarrow \mathbf{s}(\mathbf{x}_n, \mathbf{y}, t)$;
  Sample $\mathbf{z}_{\text{tr}}, \mathbf{z}_{\text{rot}}, \mathbf{z}_{\text{tor}}$ from $\mathcal{N}(0, \Delta\sigma_{\text{tr}}^2), \mathcal{N}(0, \Delta\sigma_{\text{rot}}^2), \mathcal{N}(0, \Delta\sigma_{\text{tor}}^2)$ respectively;
  Set $\Delta\mathbf{r} \leftarrow \Delta\sigma_{\text{tr}}^2\alpha + \mathbf{z}_{\text{tr}}$;
  Set $\Delta R \leftarrow \mathbf{R}(\Delta\sigma_{\text{rot}}^2\beta + \mathbf{z}_{\text{rot}})$;
  Set $\Delta\boldsymbol{\theta} \leftarrow \Delta\sigma_{\text{tor}}^2\gamma + \mathbf{z}_{\text{tor}}$;
  Compute $\mathbf{x}_{n-1} \leftarrow A((\Delta\mathbf{r}, \Delta R, \Delta\boldsymbol{\theta}), \mathbf{x}_n)$;
Return $\mathbf{x}_0$;

---

## C  ARCHITECTURE DETAILS

As summarized in Section 4.5, we use convolutional networks based on tensor products of irreducible representations (irreps) of SO(3) [Thomas et al., 2018] as architecture for both the score and confidence models. In particular, these are implemented using the `e3nn` library [Geiger et al., 2020]. Below, $\otimes_w$ refers to the spherical tensor product of irreps with path weights $w$, and $\oplus$ refers to normal vector addition (with possibly padded inputs). Features have multiple channels for each irrep. Both the architectures can be decomposed into three main parts: embedding layer, interaction layers, and output layer. We outline each of them below.

## C.1 EMBEDDING LAYER

**Geometric heterogeneous graph.** Structures are represented as heterogeneous geometric graphs with nodes representing ligand (heavy) atoms, receptor residues (located in the position of the $\alpha$-carbon atom), and receptor (heavy) atoms (only for the confidence model). Because of the high number of nodes involved, it is necessary for the graph to be sparsely connected for runtime and memory constraints. Moreover, sparsity can act as a useful inductive bias for the model, however, it is critical for the model to find the right pose that nodes that might have a strong interaction in the final pose to be connected during the diffusion process. Therefore, to build the radius graph, we connect nodes using cutoffs that are dependent on the types of nodes they are connecting:

1. Ligand atoms-ligand atoms, receptor atoms-receptor atoms, and ligand atoms-receptor atoms interactions all use a cutoff of 5Å, standard practice for atomic interactions. For the ligand atoms-ligand atoms interactions we also preserve the covalent bonds as separate edges with some initial embedding representing the bond type (single, double, triple and aromatic). For receptor atoms-receptor atoms interactions, we limit at 8 the maximum number of neighbors of each atom. Note that the ligand atoms-receptor atoms only appear in the confidence model where the final structure is already set.

2. Receptor residues-receptor residues use a cutoff of 15 Å with 24 as the maximum number of neighbors for each residue.

3. Receptor residues-ligand atoms use a cutoff of $20 + 3 * \sigma_{tr}$ Å where $\sigma_{tr}$ represents the current standard deviation of the diffusion translational noise present in each dimension (zero for the confidence model). Intuitively this guarantees that with high probability, any of the ligands and receptors that will be interacting in the final pose the diffusion model converges to are connected in the message passing at every step.

4. Finally, receptor residues are connected to the receptor atoms that form the corresponding amino-acid.

**Node and edge featurization.** For the receptor residues, we use the residue type as a feature as well as a language model embedding obtained from ESM2 [Lin et al., 2022]. The ligand atoms have the following features: atomic number; chirality; degree; formal charge; implicit valence; the number of connected hydrogens; the number of radical electrons; hybridization type; whether or not it is in an aromatic ring; in how many rings it is; and finally, 6 features for whether or not it is in a ring of size 3, 4, 5, 6, 7, or 8. These are concatenated with sinusoidal embeddings of the diffusion time [Vaswani et al., 2017] and, in the case of edges, radial basis embeddings of edge length [Schütt et al., 2017]. These scalar features of each node and edge are then transformed with learnable two-layer MLPs (different for each node and edge type) into a set of scalar features that are used as initial representations by the interaction layers.

**Notation** Let $(\mathcal{V}, \mathcal{E})$ represent the heterogeneous graph, with $\mathcal{V} = (\mathcal{V}_\ell, \mathcal{V}_r)$ respectively ligand atoms and receptor residues (receptor atoms $\mathcal{V}_a$, present in the confidence model, are for simplicity not included here), and similarly $\mathcal{E} = (\mathcal{E}_{\ell\ell}, \mathcal{E}_{\ell r}, \mathcal{E}_{r\ell}, \mathcal{E}_{rr})$. Let $\mathbf{h}_a$ be the node embeddings (initially only scalar channels) of node $a$, $e_{ab}$ the edge embeddings of $(a, b)$, and $\mu(r_{ab})$ radial basis embeddings of the edge length. Let $\sigma_{tr}^2$, $\sigma_{rot}^2$, and $\sigma_{tor}^2$ represent the variance of the diffusion kernel in each of the three components: translational, rotational and torsional.

## C.2 INTERACTION LAYERS

At each layer, for every pair of nodes in the graph, we construct messages using tensor products of the current node features with the spherical harmonic representations of the edge vector. The weights of this tensor product are computed based on the edge embeddings and the *scalar* features—denoted $\mathbf{h}_a^0$—of the outgoing and incoming nodes. The messages are then aggregated at each node and used to update the current node features. For every node $a$ of type $t_a$:

$$\mathbf{h}_a \leftarrow \mathbf{h}_a \underset{t \in \{\ell, r\}}{\oplus} \mathrm{BN}^{(t_a, t)} \left( \frac{1}{|\mathcal{N}_a^{(t)}|} \sum_{b \in \mathcal{N}_a^{(t)}} Y(\hat{r}_{ab}) \otimes_{\psi_{ab}} \mathbf{h}_b \right) \tag{16}$$

$$\text{with } \psi_{ab} = \Psi^{(t_a, t)}(e_{ab}, \mathbf{h}_a^0, \mathbf{h}_b^0)$$

Here, $t$ indicates an arbitrary node type, $\mathcal{N}_a^{(t)} = \{b \mid (a,b) \in \mathcal{E}_{t_a t}\}$ the neighbors of $a$ of type $t$, $Y$ are the spherical harmonics up to $\ell = 2$, and BN the (equivariant) batch normalisation. The orders of the output are restricted to a maximum of $\ell = 1$. All learnable weights are contained in $\Psi$, a dictionary of MLPs, which uses different sets of weights for different edge types (as an ordered pair so four types for the score model and nine for the confidence) and different rotational orders. Convolutional layers simultaneously operate with different sets of weights for different connection types (so 9 sets of weights for the confidence model and 4 for the score at every layer) and generate scalar and vector representations for each node.

## C.3 OUTPUT LAYER

The ligand atom representations after the final interaction layer are used in the output layer to produce the required outputs. This is where the score and confidence architecture differ significantly. On one hand, the score model's output is in the tangent space $T_{\mathbf{r}}\mathbb{T}_3 \oplus T_R SO(3) \oplus T_{\boldsymbol{\theta}} SO(2)^m$. This corresponds to having two $SE(3)$-equivariant output vectors representing the translational and rotational score predictions and $m$ $SE(3)$-invariant output scalars representing the torsional score. For each of these, we design final tensor-product convolutions inspired by classical mechanics. On the other hand, the confidence model outputs a single $SE(3)$-invariant scalar representing the confidence score. Below we detail how each of these outputs is generated.

**Translational and rotational scores.** The translational and rotational score intuitively represent, respectively, the linear acceleration of the center of mass of the ligand and the angular acceleration of the rest of the molecule around the center. Considering the ligand as a rigid object and given a set of forces and masses at each ligand, a tensor product convolution between the atoms and the center of mass would be capable of computing the desired quantities. Therefore, for each of the two outputs, we perform a convolution of each of the ligand atoms with the (unweighted) center of mass $c$.

$$\mathbf{v} \leftarrow \frac{1}{|\mathcal{V}_\ell|} \sum_{a \in \mathcal{V}_\ell} Y(\hat{r}_{ca}) \otimes_{\psi_{ca}} \mathbf{h}_a \tag{17}$$

$$\text{with } \psi_{ca} = \Psi(\mu(r_{ca}), \mathbf{h}_a^0)$$

We restrict the output of $\mathbf{v}$ to a single odd and a single even vectors (for each of the two scores). Since we are using coarse-grained representations of the protein, the score will neither be even nor odd; therefore, we sum the even and odd vector representations of $\mathbf{v}$. Finally, the magnitude (but not direction) of these vectors is adjusted with an MLP taking as input the current magnitude and the sinusoidal embeddings of the diffusion time. Finally, we (revert the normalization) by multiplying the outputs by $1/\sigma_{tr}$ for the translational score and by the expected magnitude of a score in $SO(3)$ with diffusion parameter $\sigma_{rot}$ (precomputed numerically).

**Torsional score.** To predict the $m$ $SE(3)$-invariant scalar describing the torsional score, we use a pseudotorque layer similar to that of Jing et al. [2022]. This predicts a scalar score $\delta\tau$ for each rotatable bond from the per-node outputs of the atomic convolution layers. For rotatable bond $g = (g_0, g_1)$ and $b \in \mathcal{V}_\ell$, let $r_{gb}$ and $\hat{r}_{gb}$ be the magnitude and direction of the vector connecting the center of bond $g$ and $b$. We construct a convolutional filter $T_g$ for each bond $g$ from the tensor product of the spherical harmonics with a $\ell = 2$ representation of the bond axis $\hat{r}_g$:[9]

$$T_g(\hat{r}) := Y^2(\hat{r}_g) \otimes Y(\hat{r}) \tag{18}$$

$\otimes$ is the full (i.e., unweighted) tensor product as described in Geiger & Smidt [2022], and the second term contains the spherical harmonics up to $\ell = 2$ (as usual). This filter (which contains orders up to $\ell = 3$) is then used to convolve with the representations of every neighbor on a radius graph:

$$\mathcal{E}_\tau = \{(g, b) \mid g \text{ a rotatable bond}, b \in \mathcal{V}_\ell\}$$

$$e_{gb} = \Upsilon^{(\tau)}(\mu(r_{gb})) \quad \forall (g, b) \in \mathcal{E}_\tau$$

$$\mathbf{h}_g = \frac{1}{|\mathcal{N}_g|} \sum_{b \in \mathcal{N}_g} T_g(\hat{r}_{gb}) \otimes_{\gamma_{gb}} \mathbf{h}_b \tag{19}$$

$$\text{with } \gamma_{gb} = \Gamma(e_{gb}, \mathbf{h}_b^0, \mathbf{h}_{g_0}^0 + \mathbf{h}_{g_1}^0)$$

---

[9]Since the parity of the $\ell = 2$ spherical harmonic is even, this representation is indifferent to the choice of bond direction.

Here, $\mathcal{N}_g = \{b \mid (g, b) \in \mathcal{E}_\tau\}$ and $\Upsilon^{(\tau)}$ and $\Gamma$ are MLPs with learnable parameters. Since unlike Jing et al. [2022], we use coarse-grained representations the parity also here is neither even nor odd, the irreps in the output are restricted to arrays both even $\mathbf{h}'_g$ and odd $\mathbf{h}''_g$ scalars. Finally, we produce a single scalar prediction for each bond:

$$\delta\tau_g = \Pi(\mathbf{h}'_g + \mathbf{h}''_g) \tag{20}$$

where $\Pi$ is a two-layer MLP with `tanh` nonlinearity and no biases. This is also "denormalized" by multiplying by the expected magnitude of a score in $SO(2)$ with diffusion parameter $\sigma_{tor}$.

**Confidence output.** The single $SE(3)$-invariant scalar representing the confidence score output is instead obtained by concatenating the even and odd final scalar representation of each ligand atom, averaging these feature vectors among the different atoms, and finally applying a three layers MLP (with batch normalization).

## D  EXPERIMENTAL DETAILS

In general, all our code is available at `https://github.com/gcorso/DiffDock`. This includes running the baselines, runtime calculations, training and inference scripts for DIFFDOCK, the PDB files of DIFFDOCK's predictions for all 363 complexes of the test set, and visualization videos of the reverse diffusion.

### D.1  EXPERIMENTAL SETUP

**Data.** We use the molecular complexes in PDBBind [Liu et al., 2017] that were extracted from the Protein Data Bank (PDB) [Berman et al., 2003]. We employ the time-split of PDBBind proposed by Stärk et al. [2022] with 17k complexes from 2018 or earlier for training/validation and 363 test structures from 2019 with no ligand overlap with the training complexes. This is motivated by the further adoption of the same split [Lu et al., 2022] and the critical assessment of PDBBind splits by Volkov et al. [2022] who favor temporal splits over artificial splits based on molecular scaffolds or protein sequence/structure similarity. For completeness, we also report the results on protein sequence similarity splits in Appendix F. We download the PDBBind data as it is provided by EquiBind from `https://zenodo.org/record/6408497`. These files were preprocessed with Open Babel before adding any potentially missing hydrogens, correcting hydrogens, and correctly flipping histidines with the `reduce` library available at `https://github.com/rlabduke/reduce`.

**Metrics.** To evaluate the generated complexes, we compute the heavy-atom RMSD between the predicted and the crystal ligand atoms when the protein structures are aligned. To account for permutation symmetries in the ligand, we use the symmetry-corrected RMSD of sPyRMSD [Meli & Biggin, 2020]. For these RMSD values, we report the percentage of predictions that have an RMSD that is less than 2Å. We choose 2Å since much prior work considers poses with an RMSD less that 2Å as "good" or successful [Alhossary et al., 2015; Hassan et al., 2017; McNutt et al., 2021]. This is a chemically relevant metric, unlike the mean RMSD as detailed in Section 3 since for further downstream analyses such as determining function changes, a prediction is only useful below a certain RMSD error threshold. Less relevant metrics such as the mean RMSD are provided in Appendix F.

### D.2  APO-STRUCTURE DOCKING

Although large and comprehensive, the PDBBind benchmark only evaluates the capacity that various docking methods have to bind ligands to their corresponding receptor holo-structure. This is a much simpler and less realistic scenario than what is typically encountered in real applications where docking for new ligands is done against apo or holo-structures bound to a different ligand. In particular, since the development of accurate protein folding methods [Jumper et al., 2021], docking programs are often run on top of AI-generated protein structures. With this in mind, we develop a new benchmark where we combine the complex prediction of PDBBind with protein structures generated by ESMFold [Lin et al., 2022].

The structures were obtained running `esmfold_v1` on the sequences extracted from the PDB protein files of PDBBind. Protein complexes were passed to ESMFold by concatenating (with `':'`) the sequences of the proteins and removing all water molecules and other ligands. Predictions were

run on 48GB A6000 GPUs, but for 12/361 complexes the prediction ran out of memory even when reducing the `chunk_size` and were, therefore, discarded.

The main design choice when generating this benchmark relies on how to best align the PDBBind complex with the ESMFold structure to obtain the "ground-truth" docked prediction on the ESMFold structure. An unbiased global alignment of the two protein structures is not desirable because a difference in structure not affecting the pocket where the ligand binds would cause the two pockets to misalign; on the other hand, only aligning residues within a single arbitrary pocket cutoff has many undesirable cases where too many or too few residues are selected or not weighted properly.

Instead, we align receptors' residues with the Kabsch algorithm using exponential weighting, for every receptor $\mathbf{x}$ its weight is $w_{\mathbf{x}} = e^{-\lambda\, d_{\mathbf{x}}}$ where $\lambda$ is a smoothing factor and $d_{\mathbf{x}}$ is the minimum distance of $\mathbf{x}$ to a ligand atom in the original complex, this way residues closer to the ligand will have a higher weight in the alignment. For each complex, we individually select $\lambda \in [0, 1]$ so that it preserves distances as best as possible, in particular, we use the L-BFGS-B [Byrd et al., 1995] from `scipy` [Virtanen et al., 2020b] to minimize:

$$\lambda^* = \min_{\lambda} \sum_{\mathbf{x} \in \mathcal{X}} \sum_{\mathbf{y} \in \mathcal{Y}} \left( \frac{1}{\|\mathbf{x}_c - \mathbf{y}\|} - \frac{1}{\|\mathbf{x}_e(\lambda) - \mathbf{y}\|} \right)^2$$

where $\|\mathbf{x}_c - \mathbf{y}\|$ and $\|\mathbf{x}_e(\lambda) - \mathbf{y}\|$ correspond to the distances between protein residue $\mathbf{x}$ and ligand atom $\mathbf{y}$ respectively in the original crystal structure from PDBBind and in the complex structure obtained aligning the ESMFold structure with smoothing parameter $\lambda$. We use inverse distances to give more importance to residues closer to the ligand (in either structure) and avoid steric clashes. We only consider protein backbones because the side-chain predictions are often less reliable and their structure typically changes upon binding.

Thus we obtain protein structures on which we run the docking methods and the associated docked ligand positions that we use to evaluate them.

### D.3 Implementation details: hyperparameters, training, and runtime measurement

**Training Details.** We use Adam [Kingma & Ba, 2014] as optimizer for the diffusion and the confidence model. The diffusion model with which we run inference uses the exponential moving average of the weights during training, and we update the moving average after every optimization step with a decay factor of 0.999. The batch size is 16. We run inference with 20 denoising steps on 500 validation complexes every 5 epochs and use the set of weights with the highest percentage of RMSDs less than 2Å as the final diffusion model. We trained our final score model on four 48GB RTX A6000 GPUs for 850 epochs (around 18 days). The confidence model is trained on a single 48GB GPU. For inference, only a single GPU is required. Scaling up the model size seems to improve performance and future work could explore whether this trend continues further. For the confidence model uses the validation cross-entropy loss is used for early stopping and training only takes 75 epochs. Code to reproduce all results including running the baselines or to perform docking calculations for new complexes is available at `https://github.com/gcorso/DiffDock`.

**Hyperparameters.** For determining the hyperparameters of DIFFDOCK's score model, we trained smaller models (3.97 million parameters) that fit into 48GB of GPU RAM before scaling it up to the final model (20.24 million parameters) that was trained on four 48GB GPUs. The smaller models were only trained for 250 or 300 epochs, and we used the fraction of predictions with an RMSD below 2Å on the validation set to choose the hyperparameters. Table 2 shows the main hyperparameters we tested and the final parameters of the large model we use to obtain our results. We only did little tuning for the minimum and maximum noise levels of the three components of the diffusion. For the translation, the maximum standard deviation is 19Å. We also experimented with second-order features for the Tensor Field Network but did not find them to help. The complete set of hyperparameters next to the main ones we describe here can be found in our repository. From the start we have divided the inference schedule into 20 time steps, the effect of using more or fewer steps for inference is discussed in Appendix F.3. As we found that the large-scale diffusion models overfit the training data on low-levels of noise we stop the diffusion early after 18 steps. At the last diffusion step no noise is added.

The confidence model has 4.77 million parameters and the parameters we tried are in Table 3. We generate 28 different training poses for the confidence model (for which it predicts whether or not they have an RMSD below 2Å) with a small score model. The score model used to generate the training samples for the confidence model does not need to be the same one that the model will be applied to at inference time.

Table 2: The hyperparameter options we searched through for DIFFDOCK's score model. This was done with small models before scaling up to a large model. The parameters shown here that impact model size (bottom half of the table) are those of the large model. The final parameters for the large DIFFDOCK model are marked in **bold**.

| PARAMETER | SEARCH SPACE |
|---|---|
| USING ALL ATOMS FOR THE PROTEIN GRAPH | YES, **NO** |
| USING LANGUAGE MODEL EMBEDDINGS | **YES**, NO |
| USING LIGAND HYDROGENS | YES, **NO** |
| USING EXPONENTIAL MOVING AVERAGE | **YES**, NO |
| MAXIMUM NUMBER OF NEIGHBORS IN PROTEIN GRAPH | 10, 16, **24**, 30 |
| MAXIMUM NEIGHBOR DISTANCE IN PROTEIN GRAPH | 5, 10, **15**, 18, 20, 30 |
| DISTANCE EMBEDDING METHOD | **SINUSOIDAL**, GAUSSIAN |
| DROPOUT | 0, 0.05, **0.1**, 0.2 |
| LEARNING RATES | 0.01, 0.008, 0.003, **0.001**, 0.0008, 0.0001 |
| BATCH SIZE | 8, **16**, 24 |
| NON LINEARITIES | **ReLU** |
| CONVOLUTION LAYERS | 6 |
| NUMBER OF SCALAR FEATURES | 48 |
| NUMBER OF VECTOR FEATURES | 10 |

Table 3: The hyperparameter options we searched through for DIFFDOCK's confidence model. The final parameters are marked in **bold**.

| PARAMETER | SEARCH SPACE |
|---|---|
| USING ALL ATOMS FOR THE PROTEIN GRAPH | **YES**, NO |
| USING LANGUAGE MODEL EMBEDDINGS | **YES**, NO |
| USING LIGAND HYDROGENS | **NO** |
| USING EXPONENTIAL MOVING AVERAGE | **NO** |
| MAXIMUM NUMBER OF NEIGHBORS IN PROTEIN GRAPH | 10, 16, **24**, 30 |
| MAXIMUM NEIGHBOR DISTANCE IN PROTEIN GRAPH | 5, 10, **15**, 18, 20, 30 |
| DISTANCE EMBEDDING METHOD | **SINUSOIDAL** |
| DROPOUT | 0, 0.05, **0.1**, 0.2 |
| LEARNING RATES | 0.03, 0.003, **0.0003**, 0.00008 |
| BATCH SIZE | **16** |
| NON LINEARITIES | **ReLU** |
| CONVOLUTION LAYERS | 5 |
| NUMBER OF SCALAR FEATURES | 24 |
| NUMBER OF VECTOR FEATURES | 6 |

**Runtime.** Similar to all the baselines, the preprocessing times are not included in the reported runtimes. For DIFFDOCK the preprocessing time is negligible compared to the rest of the inference time where multiple reverse diffusion steps are performed. Preprocessing mainly consists of a forward pass of ESM2 to generate the protein language model embeddings, RDKit's conformer generation, and the conversion of the protein into a radius graph. We measured the inference time when running on an RTX A100 40GB GPU when generating 10 samples. The runtimes we report for generating 40 samples and ranking them are extrapolations where we multiply the runtime for 10 samples by 4. In practice, this only gives an upper bound on the runtime with 40 samples, and the actual runtime should be faster.

**Statistical significance.** To determine the statistical significance of the superior performance of our method we used the paired two-sample t-test implemented in `scipy` [Virtanen et al., 2020a].

### D.4 BASELINES: IMPLEMENTATION, USED SCRIPTS, AND RUNTIME DETAILS

Our scripts to run the baselines are available at `https://github.com/gcorso/DiffDock`. For obtaining the runtimes of the different methods, we always used 16 CPUs except for GLIDE as explained below. The runtimes do not include any preprocessing time for any of the methods. For instance, the time that it takes to run P2Rank is not included for TANKBind, and P2Rank + SMINA/GNINA since this receptor preparation only needs to be run once when docking many ligands to the same protein. In applications where different receptors are processed (such as reverse screening), the experienced runtimes for TANKBind and P2Rank + SMINA/GNINA will thus be higher.

We note that for all these baselines we have used the default hyperparameters unless specified differently below. Modifying some of these hyperparameters (for example the scoring method's exhaustiveness) will change the runtime and performance tradeoffs (e.g., if the searching routine is left running for longer then better poses are likely to be found), however, we leave these analyses to future work.

**SMINA** [Koes et al., 2013] improves Autodock Vina with a new scoring-function and user-friendliness. The default parameters were used with the exception of setting `--num_modes 10`. To define the search box, we use the automatic box creation option around the receptor with the default buffer of 4Å on all 6 sides.

**GNINA** [McNutt et al., 2021] builds on SMINA by additionally using a learned 3D CNN for scoring. The default parameters were used with the exception of setting `--num_modes 10`. To define the search box, we use the automatic box creation option around the receptor with the default buffer of 4Å on all 6 sides.

**QuickVina-W** [Hassan et al., 2017] extends the speed-optimized QuickVina 2 [Alhossary et al., 2015] for blind docking. We reuse the numbers from Stärk et al. [2022] which had used the default parameters except for increasing the exhaustiveness to 64. The files were preprocessed with the `prepare_ligand4.py` and `prepare_receptor4.py` scripts of the MGLTools library as it is recommended by the QuickVina-W authors.

**Autodock Vina** [Trott & Olson, 2010] is older docking software that does not perform as well as the other more recent search-based baselines, but it is a well-established tool. We reuse the numbers reported in TANKBind [Lu et al., 2022]

**GLIDE** [Halgren et al., 2004] is a strong heavily used commercial docking tool. These methods all use biophysics based scoring-functions. We reuse the numbers from Stärk et al. [2022] since we do not have a license. Running GLIDE involves running their command line tools for preprocessing the structures into the files required to run the docking algorithm. As explained by Stärk et al. [2022], the very high runtime of GLIDE with 1405 seconds per complex is partially explained by the fact that GLIDE only uses a single thread when processing a complex. This fact and the parallelization options of GLIDE are explained here `https://www.schrodinger.com/kb/1165`. With GLIDE, it is possible to start data-parallel processes that compute the docking results for a different complex in parallel. However, each process also requires a separate software license.

**EquiBind** [Stärk et al., 2022], we reuse the numbers reported in their paper and generate the predictions that we visualize with their code at `https://github.com/HannesStark/EquiBind`.

**TANKBind** [Lu et al., 2022], we use the code associated with the paper at `https://github.com/luwei0917/TankBind`. The runtimes do not include the runtime of P2Rank or any pre-processing steps. In Table 1 we report two runtimes (0.72/2.5 sec). The first is the runtime when making only the top-1 prediction and the second is for producing the top-5 predictions. Producing only the top-1 predictions is faster since TANKBind produces distance predictions that need to be converted to coordinates with a gradient descent algorithm and this step only needs to be run once for the top-1 prediction, while it needs to be run 5 times for producing 5 outputs. To obtain our runtimes we run the forward pass of TANKBind on GPU (0.28 seconds) with the default batch size of 5 that is used in their GitHub repository. To compute the time the distances-to-coordinates conversion step takes, we run the file `baseline_run_tankbind_parallel.sh` in our repository, which parallelizes the computation across 16 processes which we also run on an Intel Xeon Gold

6230 CPU. This way, we obtain 0.44 seconds runtime for the conversion step of the top-1 prediction (averaged over the 363 complexes of the testset).

**P2Rank** [Krivák & Hoksza, 2018], is a tool that predicts multiple binding pockets and ranks them. We use it for running TANKBind and P2Rank + SMINA/GNINA. We download the program from `https://github.com/rdk/p2rank` and run it with its default parameters.

**EquiBind + SMINA/GNINA** [Stärk et al., 2022], the bounding box in which GNINA/SMINA searches for binding poses is constructed around the prediction of EquiBind with the `--autobox_ligand` option of GNINA/SMINA. EquiBind is thus used to find the binding pocket and SMINA/GNINA to find the exact final binding pose. We use `--autobox_add 10` to add an additional 10Å on all 6 sides of the bounding box following [Stärk et al., 2022].

**EquiBind + SMINA$^{flex}$/GNINA$^{flex}$** The same as EquiBind + SMINA/GNINA but with the flexibility in terms of torsion angles activated for the sidechains that have an atom within 3.5Å of the output from EquiBind.

**P2Rank + SMINA/GNINA.** The bounding box in which GNINA/SMINA searches for binding poses is constructed around the pocket center that P2Rank predicts as the most likely binding pocket. P2Rank is thus used to find the binding pocket and SMINA/GNINA to find the exact final binding pose. The diameter of the search box is the diameter of a ligand conformer generated by RDKit with an additional 10Å on all 6 sides of the bounding box.

**P2Rank + SMINA$^{flex}$/GNINA$^{flex}$** The same as P2Rank + SMINA/GNINA but with the flexibility in terms of torsion angles activated for the sidechains that have an atom within 3.5Å + the radius of the ligand away from the pocket center of P2Rank. Additionally we consider at most 10 flexible sidechains with `--flex_max 10`.

**SMINA/GNINA + SMINA$^{flex}$/GNINA$^{flex}$** The same as EquiBind + SMINA/GNINA but the initial ligand is provided by SMINA/GNINA ran without sidechain flexibility. The flexibility is in terms of torsion angles activated for the sidechains that have an atom within 3.5Å of the output from SMINA/GNINA.

# E ADDITIONAL DISCUSSION

## E.1 ACCESS TO BOUND PROTEIN STRUCTURE

One limitation of DIFFDOCK is that it assumes access to the bound structure of the protein known as holo-structure. Although most of the literature in molecular docking makes this assumption, in practice, one often only has access to the unbound apo protein structure or the holo structure of the protein bound to a different ligand. Tackling the problem of binding to apo structures is challenging due to the limited amount of data of unbound structures that have an atom-to-atom correspondence to the holo structure with which they could be aligned. In DIFFDOCK the limitation of docking to holo structures may be less pronounced than for the search based docking methods that we compare against since DIFFDOCK's score model only uses the positions of the alpha carbon atoms (and not the side chain atoms). Thus DIFFDOCK would also work well for binding to apo structures when most of the conformational change during binding lies in the side chains and the backbone stays mostly rigid. However, in order to fully model binding to apo structures, one needs to additionally model protein flexibility, we leave the full treatment of this problem to future work.

## E.2 TORSIONAL VS EUCLIDEAN FLEXIBILITY

A ligand pose (and more generally speaking, a molecule conformation) consists of a position in 3D-space $\mathbb{R}^3$ for each of $n$ atoms of the molecule and can thus be considered as an element of $\mathbb{R}^{3n}$. Each such pose (or conformation), can alternatively be described in terms of its bond lengths, angles, and torsion angles (see Jing et al. [2022], Appendix A for a more formal discussion). Because of the nature of covalent interactions, bond lengths and angles (local structures) are highly energetically constrained, and any specific bond length or angle in a molecule will take on only a very narrow range of values, whether in isolation or bound to a protein receptor. Thus, ligand poses or conformations in which bond lengths or angles are *strained*—i.e., differ significantly from their standard values—can be easily judged to be energetically unfavorable. On the other hand, the en-

ergetic profile of varying torsion angles is significantly smoother, as they depend in large part on weaker, noncovalent interactions. A change of chemical environment, or an interaction with another molecule (such as a protein) can alter these profiles. Thus, we say that there is significant *flexibility* in the torsion angles, in contrast to the more rigid local structures.

Since bond lengths and angles are highly constrained, the set of chemically plausible ligand poses is a very small subset of all possible assignments of atoms to $\mathbb{R}^3$; that is, a very restricted subset of $\mathbb{R}^{3n}$. The set of ligand poses that satisfy all the bond length and angle constraints with equality appears as a lower-dimensional *manifold*. To a good approximation, all actual ligand poses lie on this manifold. Thus, we are interested in developing a generative model on this manifold, as any sample outside of it is chemically nonsensical.

Futher, the constrained values of standard bond lengths and angles are easily and quickly predicted using standard cheminformatics routines, such as those in RDKit. Indeed, far simpler rules of thumb—a fixed length for each type of bonding pair, and fixed angles determined solely by molecular geometry—are already quite accurate and widely taught. Consequently, the manifold of plausible poses is very easy to find—one merely has to generate *any* plausible conformation or ligand pose with RDKit. In previous work on conformer generation, Jing et al. [2022] verified that the manifolds corresponding to local structures generated by RDKit are, on average, less than 0.5 Å RMSD away from the true conformations.

The facts that (1) the space of plausible ligand poses is described by a manifold and (2) this manifold is easy to find, motivate the development of a diffusion generative model on the manifold rather than the full ambient space $\mathbb{R}^{3n}$. This significantly reduces the dimensionality of the state space, and thus of the function that the score network needs to approximate.

# F ADDITIONAL RESULTS

## F.1 PHYSICALLY PLAUSIBLE PREDICTIONS

Table 4: **Steric clashes.** Percentage of test complexes for which the predictions of the different methods exhibit steric clashes. Search-based methods never produced steric clashes.

| Method | Top-1 % steric clashes | Top-5 % steric clashes |
|---|---|---|
| EQUIBIND | 26 | - |
| TANKBIND | 6.6 | 3.6 |
| **DIFFDOCK (10)** | **2.8** | **0** |
| **DIFFDOCK (40)** | **2.2** | **2.2** |

Due to the averaging phenomenon of regression-based methods such as TANKBind and EquiBind, they make predictions at the mean of the distribution. If aleatoric uncertainty is present, such as in case of symmetric complexes, this leads to predicting the ligand to be at an un-physical state in the middle of the possible binding pockets as visualized in Figure 13. The Figure also illustrates how DIFFDOCK does not suffer from this issue and is able to accurately sample from the modes.

In the scenario when epistemic uncertainty about the correct ligand conformation is present, this often results in "squashed-up" predictions of the regression-based methods as visualized in Figure 4. If there is uncertainty about the correct conformer, the square error minimizing option is to put all atoms close to the mean.

These averaging phenomena in the presence of either aleatoric or epistemic uncertainty cause the regression-based methods to often generate steric clashes and self intersections. To investigate this quantitatively, we determine the fraction of test complexes for which the methods exhibit steric clashes. We define a ligand as exhibiting a steric clash if one of its heavy atoms is within 0.4Å of a heavy receptor atom. This cutoff is used by protein quality assessment tools and in previous literature [Ramachandran et al., 2011]. Table 4 shows that DIFFDOCK, as a generative model, produces fewer steric clashes than the regression-based baselines. We generally observe no unphysical

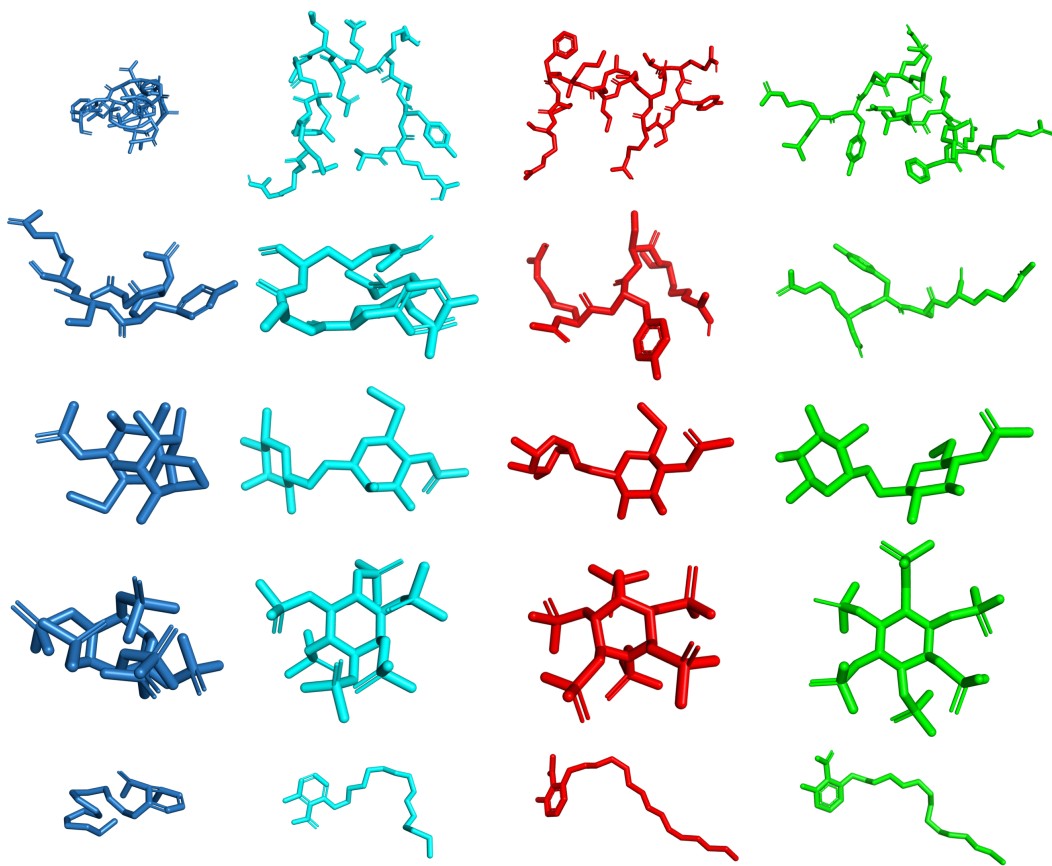

Figure 4: **Ligand self-intersections.** TANKBind (blue), EquiBind (cyan), DIFFDOCK (red), and crystal structure (green). Due to the averaging phenomenon that occurs when epistemic uncertainty is present, the regression-based deep learning models tend to produce ligands with atoms that are close together, leading to self-intersections. DIFFDOCK, as a generative model, does not suffer from this averaging phenomenon, and we never found a self-intersection in any of the investigated results of DIFFDOCK.

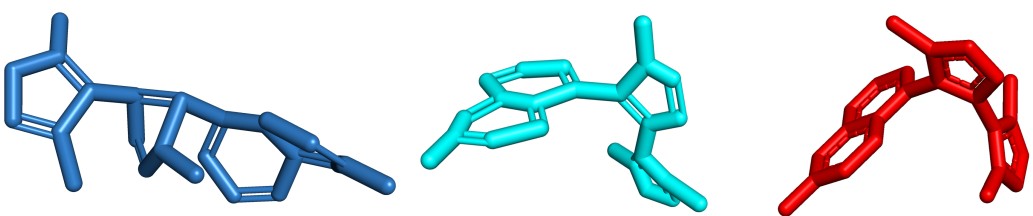

Figure 5: **Chemically plausible local structures.** TANKBind (blue), EquiBind (cyan), and DIFF-DOCK (red) structures for complex 6g2f. EquiBind (without their correction step) produces very unrealistic local structures and TANKBind, e.g., produces non-planar aromatic rings. DIFFDOCK's local structures are the realistic local structures of RDKit.

predictions from DIFFDOCK unlike the self intersections that, e.g., TANKBind produces (Figure 4) or its incorrect local structures (Figure 5). This is also visible in the randomly chosen examples of Figure 12 and can be examined in our repository, where we provide all predictions of DIFFDOCK for the test set.

### F.2 FURTHER RESULTS AND METRICS

In this section, we present further evaluation metrics on the results presented in Table 1. In particular, for both top-1 (Table 5) and top-5 (Table 6) we report: 25th, 50th and 75th percentiles, the proportion below 2Å and below 5Å of both ligand RMSD and centroid distance. Moreover, while Volkov et al. [2022] advocated against artificial protein set splits and for time-based splits, for completeness, in Table 7 and Figure 7, we report the performances of the different methods when evaluated exclusively on the portion of the test set where the UniProt IDs of the proteins are not contained in the data that is seen by DIFFDOCK in its training and validation.

To assess the impact of the molecule size on the performance of DIFFDOCK and GNINA, we provide scatter plots in Figure 8 showing that the correlation between RMSD and the number of rotatable bonds or the number of atoms in the molecule is similar for both methods. In Figure 9 we show that, based on Tanimoto similarity, the performance of DIFFDOCK does not depend on the test ligand's similarity to the ligands that have already been seen during training. The Spearman rank correlation coefficient between the RMSD and the Tanimoto similarity to the closest ligand in the training set is negligible with it being -0.031.

Further, in Table 8, we provide the performance of SMINA and GNINA in the apo-structure docking setting when the flexibility of sidechains in the pocket is turned on. These show that simply adding sidechain flexibility does not help these methods. Finally, in Figure 10, we plot the relationship between the quality of the ESMFold structure and the performance of the docking methods on apo and holo-structures. While DIFFDOCK retains a large part of its accuracy when the protein backbone is approximately correct, very small variations in the apo-structure backbone (and sidechains) causes search-based methods like GNINA to almost never find the right pose.

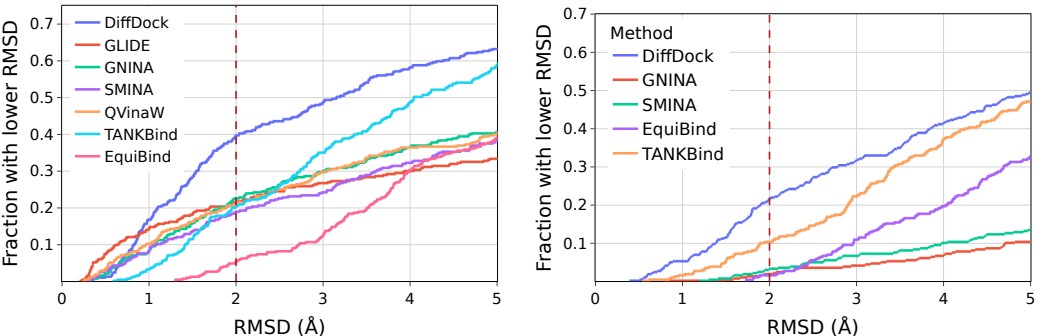

Figure 6: Cumulative density histogram of the methods' RMSD: left on holo crystal structures, right on apo ESMFold structures.

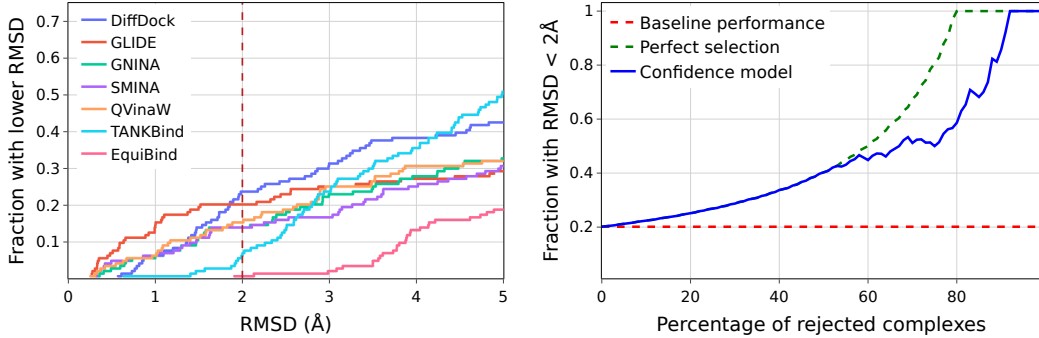

Figure 7: **PDBBind docking on unseen receptors. Left:** cumulative density histogram of the methods' RMSD. **Right:** Percentage of predictions with RMSD below 2Å when only making predictions for the portion of the dataset where DIFFDOCK is most confident.

Table 5: **Top-1 PDBBind docking.**

| | Ligand RMSD | | | | | Centroid Distance | | | | |
| | Percentiles ↓ | | | % below thresh. ↑ | | Percentiles ↓ | | | % below thresh. ↑ | |
| Methods | 25th | 50th | 75th | 5 Å | 2 Å | 25th | 50th | 75th | 5 Å | 2 Å |
|---|---|---|---|---|---|---|---|---|---|---|
| AUTODOCK VINA | 5.7 | 10.7 | 21.4 | 21.2 | 5.5 | 1.9 | 6.2 | 20.1 | 47.1 | 26.5 |
| QVINA-W | 2.5 | 7.7 | 23.7 | 40.2 | 20.9 | 0.9 | 3.7 | 22.9 | 54.6 | 41.0 |
| GNINA | 2.4 | 7.7 | 17.9 | 40.8 | 22.9 | 0.8 | 3.7 | 23.1 | 53.6 | 40.2 |
| SMINA | 3.1 | 7.1 | 17.9 | 38.0 | 18.7 | 1.0 | 2.6 | 16.1 | 59.8 | 41.6 |
| GLIDE (c.) | 2.6 | 9.3 | 28.1 | 33.6 | 21.8 | 0.8 | 5.6 | 26.9 | 48.7 | 36.1 |
| EQUIBIND | 3.8 | 6.2 | 10.3 | 39.1 | 5.5 | 1.3 | 2.6 | 7.4 | 67.5 | 40.0 |
| TANKBIND | 2.5 | 4.0 | 8.5 | 59.0 | 20.4 | 0.9 | 1.8 | 4.4 | 77.1 | 55.1 |
| P2RANK+SMINA | 2.9 | 6.9 | 16.0 | 43.0 | 20.4 | 0.8 | 2.6 | 14.8 | 60.1 | 44.1 |
| P2RANK+GNINA | 1.7 | 5.5 | 15.9 | 47.8 | 28.8 | 0.6 | 2.2 | 14.6 | 60.9 | 48.3 |
| EQUIBIND+SMINA | 2.4 | 6.5 | 11.2 | 43.6 | 23.2 | 0.7 | 2.1 | 7.3 | 69.3 | 49.2 |
| EQUIBIND+GNINA | 1.8 | 4.9 | 13 | 50.3 | 28.8 | 0.6 | 1.9 | 9.9 | 66.5 | 50.8 |
| **DIFFDOCK (10)** | 1.5 | 3.6 | **7.1** | 61.7 | 35.0 | **0.5** | **1.2** | 3.3 | **80.7** | 63.1 |
| **DIFFDOCK (40)** | **1.4** | **3.3** | 7.3 | **63.2** | **38.2** | **0.5** | **1.2** | **3.2** | 80.5 | **64.5** |

Table 6: **Top-5 PDBBind docking.**

| | Ligand RMSD | | | | | Centroid Distance | | | | |
| | Percentiles ↓ | | | % below thresh. ↑ | | Percentiles ↓ | | | % below thresh. ↑ | |
| Methods | 25th | 50th | 75th | 5 Å | 2 Å | 25th | 50th | 75th | 5 Å | 2 Å |
|---|---|---|---|---|---|---|---|---|---|---|
| GNINA | 1.6 | 4.5 | 11.8 | 52.8 | 29.3 | 0.6 | 2.0 | 8.2 | 66.8 | 49.7 |
| SMINA | 1.7 | 4.6 | 9.7 | 53.1 | 29.3 | 0.6 | 1.85 | 6.2 | 72.9 | 50.8 |
| TANKBIND | 2.1 | 3.4 | 6.1 | 67.5 | 24.5 | 0.8 | 1.4 | 2.9 | 86.8 | 62.0 |
| P2RANK+SMINA | 1.5 | 4.4 | 14.1 | 54.8 | 33.2 | 0.6 | 1.8 | 12.3 | 66.2 | 53.4 |
| P2RANK+GNINA | 1.4 | 3.4 | 12.5 | 60.3 | 38.3 | 0.5 | 1.4 | 9.2 | 69.3 | 57.3 |
| EQUIBIND+SMINA | 1.3 | 3.4 | 8.1 | 60.6 | 38.6 | 0.5 | 1.3 | 5.1 | 74.9 | 58.9 |
| EQUIBIND+GNINA | 1.4 | 3.1 | 9.1 | 61.7 | 39.1 | 0.5 | 1.1 | 5.3 | 73.7 | 60.1 |
| **DIFFDOCK (10)** | **1.2** | 2.7 | **4.9** | 75.1 | 40.7 | 0.5 | 1.0 | 2.2 | 87.0 | 72.3 |
| **DIFFDOCK (40)** | **1.2** | **2.4** | 5.0 | **75.5** | **44.7** | **0.4** | **0.9** | **1.9** | **88.0** | **76.7** |

Table 7: **PDBBind docking on unseen receptors.** Percentage of predictions for which the RMSD to the crystal structure is below 2Å and the median RMSD. "*" indicates the method run exclusively on CPU, "-" means not applicable; some cells are empty due to infrastructure constraints.

| | Top-1 RMSD | | Top-5 RMSD | | Average |
| Method | %<2 | Med. | %<2 | Med. | Runtime (s) |
|---|---|---|---|---|---|
| AUTODOCK VINA | 1.4 | 16.6 | | | 205* |
| QVINAW | 15.3 | 10.3 | | | 49* |
| GNINA | 14.0 | 13.6 | 23.0 | 7.0 | 127 |
| SMINA | 14.0 | 8.5 | 21.7 | 6.7 | 126* |
| GLIDE | 19.6 | 18.0 | | | 1405* |
| EQUIBIND | 0.7 | 9.1 | - | - | **0.04** |
| TANKBIND | 6.3 | **5.0** | 11.1 | 4.4 | 0.7/2.5 |
| **DIFFDOCK (10)** | 15.7 | 6.1 | 21.8 | 4.2 | 10 |
| **DIFFDOCK (40)** | **20.8** | 6.2 | **28.7** | **3.9** | 40 |

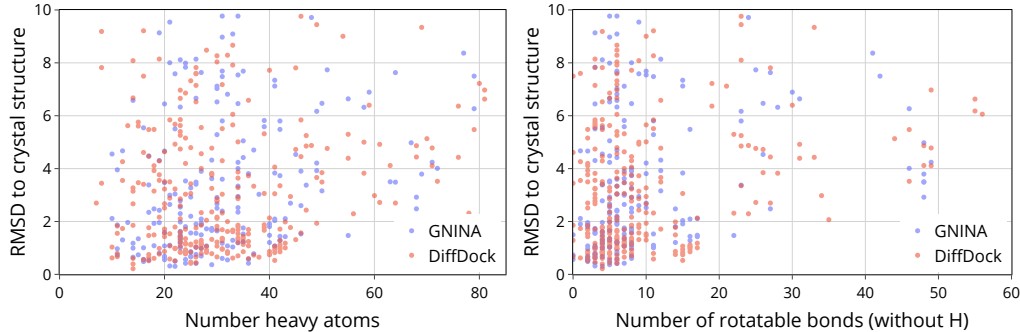

Figure 8: **Relationship between the number of atoms and RMSD. Left:** Scatter plot of the RMSD and the number of atoms. **Right:** Scatter plot of the RMSD and the number of rotatable bonds in the ligand.

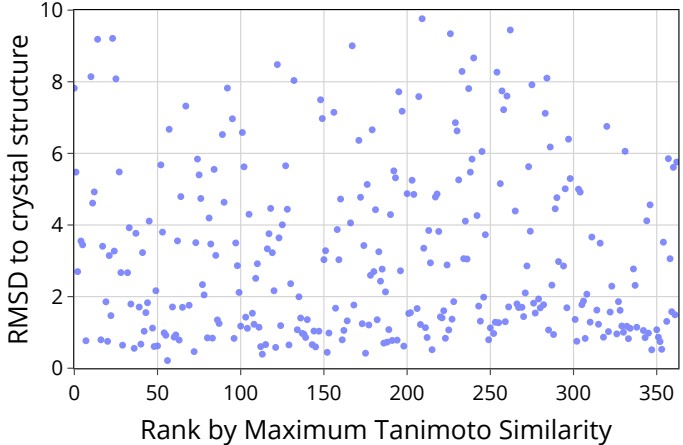

Figure 9: Scatterplot for the RMSD and the Tanimoto similarity of the test ligand to the closest ligand in the training data. The order by maximum Tanimoto similarity is such that the lowest similarity is at the left and the highest similarity is at the right on the x-axis.

Table 8: **PDBBind blind docking on apo proteins with sidechain flexibility in baselines.** All methods receive a small molecule and are tasked to find its binding location, orientation, and conformation. Shown is the percentage of predictions with RMSD $< 2$Å and the median RMSD (see Appendix D.3). In parenthesis, we specify the number of poses sampled from the generative model. * indicates that the method runs exclusively on CPU. No method received further training or tuning on ESMFold structures.

| | Apo ESMFold proteins | | | | |
| | Top-1 RMSD | | Top-5 RMSD | | Average |
| Method | %<2 | Med. | %<2 | Med. | Runtime (s) |
| --- | --- | --- | --- | --- | --- |
| P2RANK+SMINA$^{flex}$ | 5.7 | 9.6 | 13.4 | 6.3 | 292* |
| P2RANK+GNINA$^{flex}$ | 8.3 | 10.9 | 13.4 | 6.4 | 294 |
| EQUIBIND+SMINA$^{flex}$ | 4.3 | 7.3 | 11.7 | 5.8 | 1145* |
| EQUIBIND+GNINA$^{flex}$ | 6.6 | 9.8 | 14.6 | 6.1 | 1208 |
| SMINA+SMINA$^{flex}$ | 3.4 | 12.6 | 8.3 | 11.6 | 1145* |
| GNINA+GNINA$^{flex}$ | 1.7 | 22.1 | 5.1 | 20.0 | 1208 |
| **DIFFDOCK (10)** | **21.7** | **5.0** | **31.9** | **3.3** | 10 |
| **DIFFDOCK (40)** | 20.3 | 5.1 | 31.3 | **3.3** | 40 |

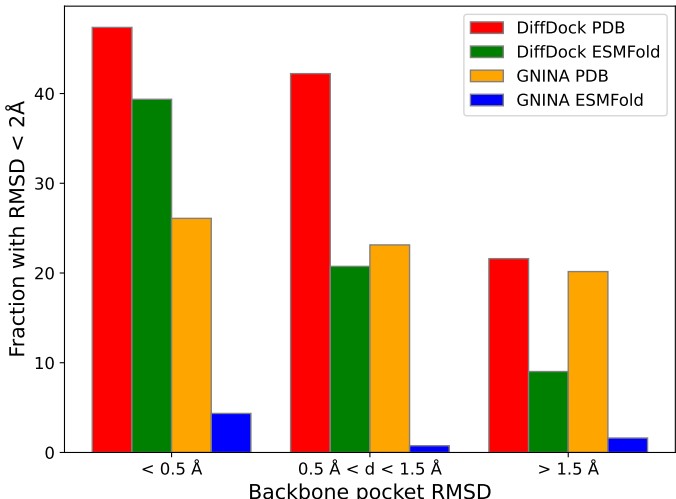

Figure 10: **Relationship between the ESMFold and docking accuracy.** The test set complexes on PDBBind are divided into three similar-sized groups based on the RMSD of the pocket residues (defined as ¡ 10 Å to a ligand atom) in the aligned ESMFold generated structure. The performance of DIFFDOCK and GNINA are shown when docking to both the ground truth crystal structure and the ESMFold structure. The baseline methods' performance on crystal structures is also negatively correlated with the ESMFold accuracy because the complexes where the methods do badly tend to be larger and with fewer examples in PDB(Bind).

### F.3    ABLATION STUDIES

Below we report the performance of our method over different hyperparameter settings. In particular, we highlight the different ways in which it is possible to control the tradeoff between runtime and accuracy in our method. These mainly are: (1) model size, (2) diffusion time, and (3) diffusion samples.

**Model size.** The final DIFFDOCK score model has 20.24 million parameters from its 6 convolution layers with 48 scalar and 10 vector features. In Table 9 we show the results for a smaller score model with 5 convolutions, 24 scalar, and 6 vector features resulting in 3.97 million parameters that can be trained on a single 48GB GPU. The confidence model used is the same for both score models. We find that scaling up the model size helped improve performance which we did as far as possible using four 48GB GPUs for training. Scaling the model size further is a promising avenue for future work.

**Protein embeddings.** As described in Section 4.5 and Appendix C, the architecture uses as initial features of protein residues the language model embeddings from ESM2 [Lin et al., 2022] in order for the model to more easily reason about the protein sequence. In Table 9 we show that while these provide some improvements they are not necessary to obtain state-of-the-art performance.

**Diffusion steps.**   Another hyperparameter determining the runtime of the method during inference is the number of steps we take during the reverse diffusion. Since these are applied sequentially DIFFDOCK's runtime scales approximately linearly with the number of diffusion steps. In the rest of the paper, we always use 20 steps, but in Figure 11 we show how the performance of the model varies with the number of steps. We note that the model reaches nearly the full performance even with just 10 steps, suggesting that the model can be sped up 2x with a small drop in accuracy.

**Diffusion samples.**   Given a score-based model and a number of steps for the diffusion model, it remains to be determined how many independent samples $N$ to query from the diffusion model and then feed to the confidence model. As expected the more samples the confidence model receives the more likely it is that it will find a pose that it is confident about and, therefore, the higher the performance. The runtime of DIFFDOCK on GPU scales sublinearly until the different samples fit in parallel in the model (depends on the protein size and the GPU memory) and approximately linearly

Table 9: **Model size and protein embeddings comparison.** All methods receive a small molecule and are tasked to find its binding location, orientation, and conformation. Shown is the percentage of predictions for which the RMSD to the crystal structure is below 2Å and the median RMSD.

| Method | Top-1 RMSD (Å) | | Top-5 RMSD (Å) | | Average |
|---|---|---|---|---|---|
| | %<2 | Med. | %<2 | Med. | Runtime (s) |
| **DIFFDOCK-SMALL-NOESM (10)** | 26.2 | 4.7 | 32.0 | 3.2 | 7 |
| **DIFFDOCK-SMALL-NOESM (40)** | 28.4 | 3.8 | 37.7 | 2.6 | 28 |
| **DIFFDOCK-SMALL (10)** | 26.0 | 4.3 | 33.3 | 3.2 | 7 |
| **DIFFDOCK-SMALL (40)** | 31.1 | 4.0 | 38.0 | 2.7 | 28 |
| **DIFFDOCK-NOESM (10)** | 33.9 | 3.8 | 39.4 | 2.8 | 10 |
| **DIFFDOCK-NOESM (40)** | 34.2 | 3.5 | 42.7 | 2.4 | 40 |
| **DIFFDOCK (10)** | 35.0 | 3.6 | 40.7 | 2.7 | 10 |
| **DIFFDOCK (40)** | **38.2** | **3.3** | **44.7** | **2.4** | 40 |

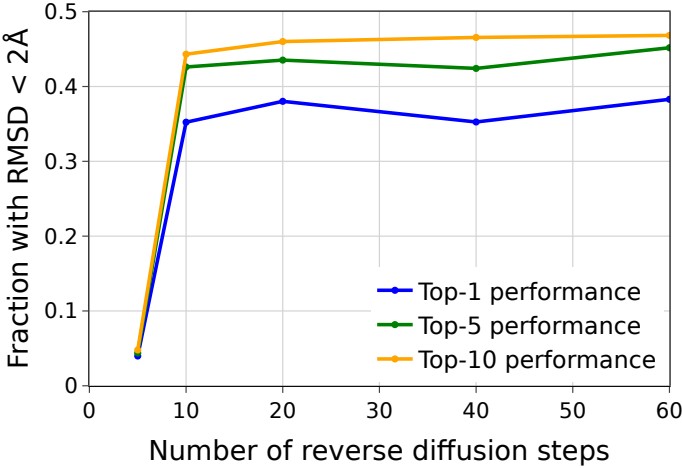

Figure 11: Ablation study on the number of reverse diffusion steps.

for larger sample sizes (however it can be easily parallelized across different GPUs). In Figure 3 we show how the success rate for the top-1, top-5, and top-10 prediction change as a function of $N$. For example, for the top-1 prediction, the proportion of the prediction with RMSD below 2Å varies between 22% of a random sample of the diffusion model ($N = 1$) to 38% when the confidence model is allowed to choose between 40 samples.

## F.4 VISUALIZATIONS

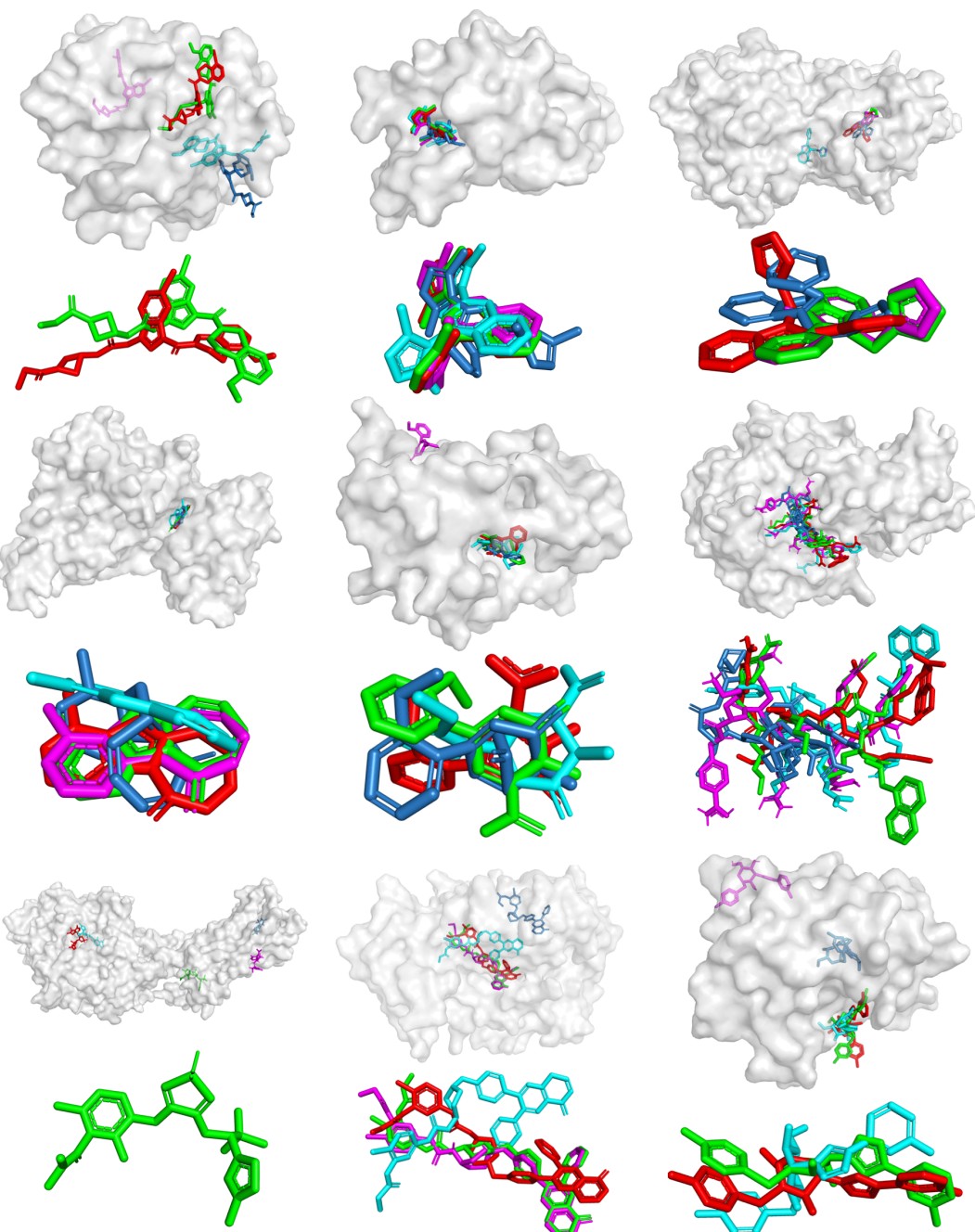

Figure 12: **Randomly picked examples.** The predictions of TANKBind (blue), EquiBind (cyan), GNINA (magenta), DIFFDOCK (red), and crystal structure (green). Shown are the predictions once with the protein and without it below. The complexes were chosen with a random number generator from the test set. TANKBind often produces self intersections (examples at the top-right; middle-middle; middle-right; bottom-right). DIFFDOCK and GNINA sometimes almost perfectly predict the bound structure (e.g., top-middle). The complexes in reading order are: 6p8y, 6mo8, 6pya, 6t6a, 6e30, 6hld, 6qzh, 6hhg, 6qln.

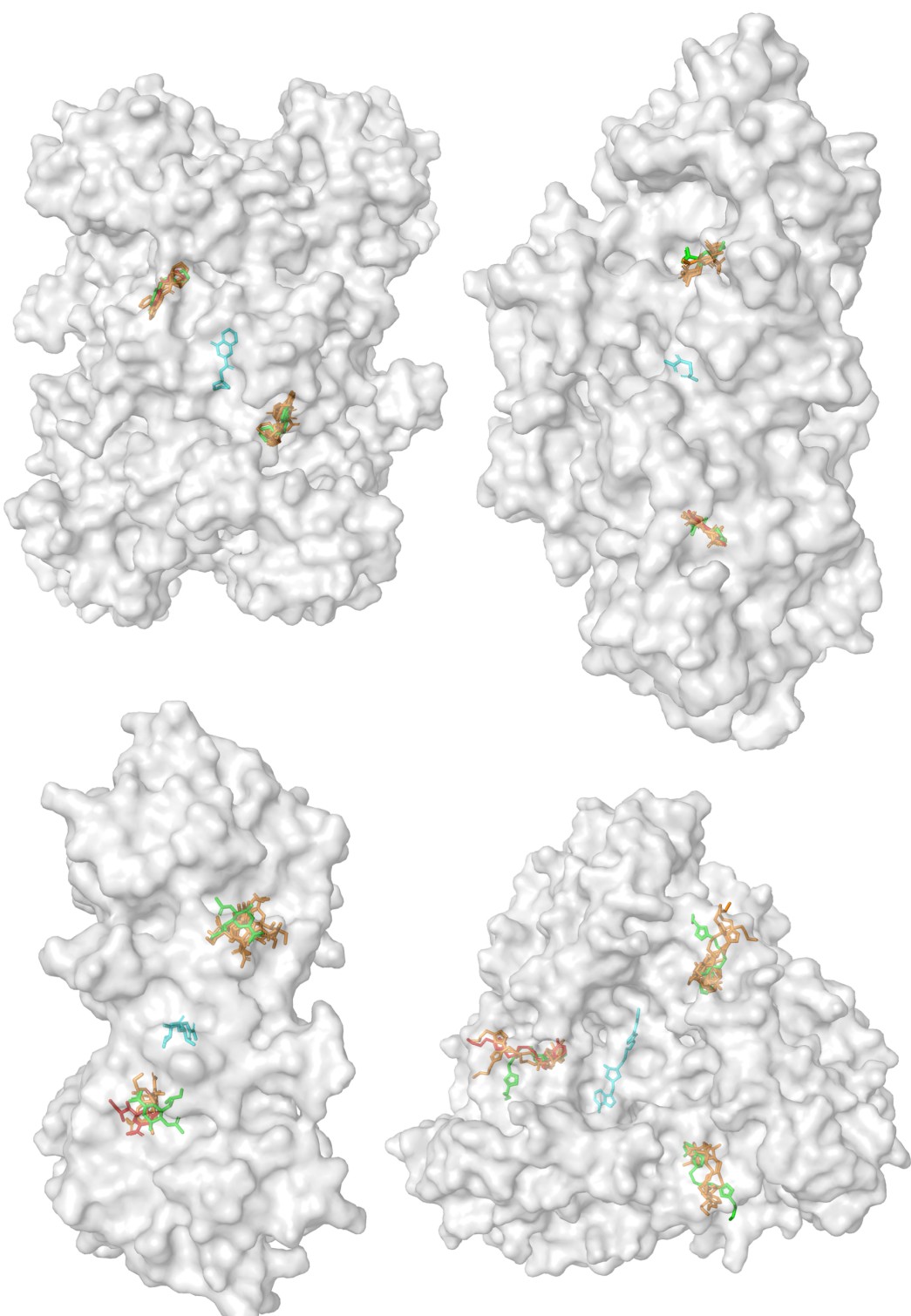

Figure 13: **Symmetric complexes and multiple modes.** EquiBind (cyan), DIFFDOCK highest confidence sample (red), all other DIFFDOCK samples (orange), and the crystal structure (green). We see that, since it is a generative model, DIFFDOCK is able to produce multiple correct modes and to sample around them. Meanwhile, as a regression-based model, EquiBind is only able to predict a structure at the mean of the modes. The complexes are unseen during training. The PDB IDs in reading order: 6agt, 6gdy, 6ckl, 6dz3.

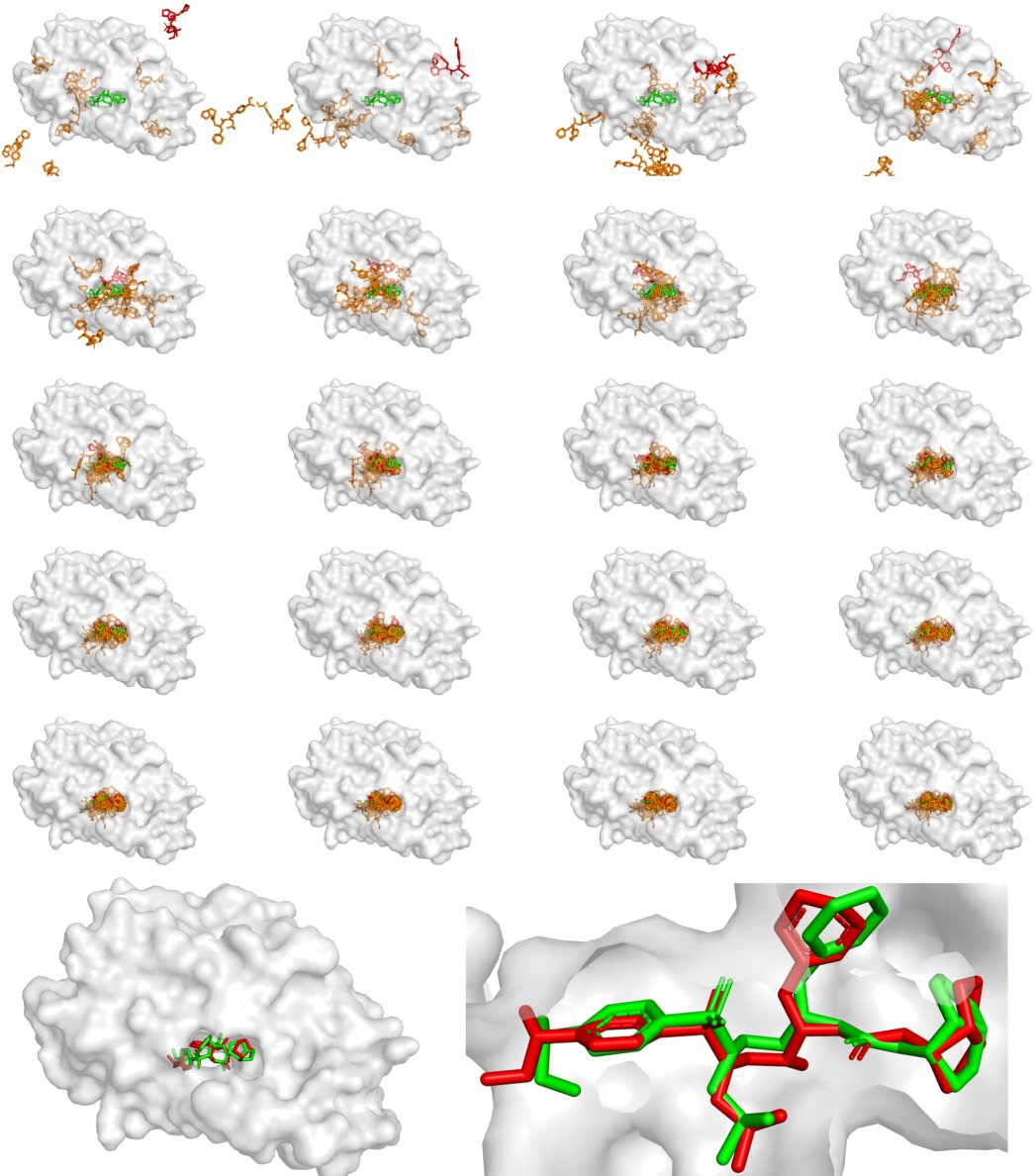

Figure 14: **Reverse Diffusion.** Reverse diffusion of a randomly picked complex from the test set. Shown are DIFFDOCK highest confidence sample (red), all other DIFFDOCK samples (orange), and the crystal structure (green). Shown are the 20 steps of the reverse diffusion process (in reading order) of DIFFDOCK for the complex 6oxx. Videos of the reverse diffusion are available at https://github.com/gcorso/DiffDock/visualizations/README.md.

