# OpenReview forum: "DiffDock: Diffusion Steps, Twists, and Turns for Molecular Docking"
_ICLR.cc/2023/Conference — ICLR 2023 poster_

### Official Review · Reviewer_Yi6P · 2022-10-23

**Confidence:** 4
**Correctness:** 2
**Technical Novelty And Significance:** 2
**Empirical Novelty And Significance:** 2
**Recommendation:** 3

**Clarity, Quality, Novelty And Reproducibility:**

### Clarity

The paper is well structured and clearly written, but the math is sloppy (see Minor Comments).

### Quality

a) This work does not embed itself into prior and similar works. It almost fully ignores a the decade-old field of molecular docking. Furthermore, it also ignores close works in machine learning (e.g. [1]).

b) There are several severe problems in the experimental part of this paper.

i) Only a single experiment on a dataset called PDBBind is performed, while there are many more benchmark datasets in the community (e.g. different variants of the DUD datasets).

ii)Inconsistencies with previous comparisons and focus on single metric that uses a threshold that is convenient for the author's method. In a previous method comparison ([5], Figure 4, comparable with Figure 3 of this paper) or another one ([7], Figure 2), the docking methods exhibit much better performance than in the presented work. Furthermore, the docking community has established multiple more metrics that are informative. One reason could be that the hyperparameters of the docking methods have not or insufficiently been adjusted to the author's training set. The authors should compare their method on previously established benchmarks in the community, ensure a fair comparison, and relate their results to prior results and show other metrics.

iii) A large number of methods has not been compared. There is a large battery of docking methods that have not been compared, VirtualFlow, Autodock, Glide, GOLD, ... [5]. The authors should compare their method against other docking methods.

iv) Complete absence of baselines. Method comparisons should contain a number of baselines such that the improvement over those can be judged. For example, a generic transformer or MPNN or even MLP could be compared. The authors should include baselines into the method comparison, e.g. generic transformers or SE3-equivariant tranformers, MLPs, generic MPNNs.

v) Except for the author's method in Table 1, all performance metrics in text, tables and figures are missing error bars. There is an absence of re-runs, error bars, confidence intervals and statistical test to determine whether the claimed improvement arises not just by chance. The others should perform multiple training runs or repetitions for each run; for docking methods this might mean that the hyperparameter should be adjusted to the training data. The authors should provide error bars or confidence intervals for all metrics and perform statistical tests.

c) The diffusion process and the model and its architecture contain many ad-hoc decisions. These are hardly justified or possible alternatives explained. For example, there are many possible representations on which the diffusion operates (e.g. [1]); there are many options for representing the protein (compare AlphaFold [6]) or the ligand. For example, an alpha-fold like representation as "residual gas" and a transformer-like architecture could also be used as model. The author's should justify their choices, perform ablations studies of the main architectural components and include other architectures as baselines.

d) It is unclear from which component the claimed performance gain arises: the formulation of the problem as diffusion process or from the neural network architecture. The authors should investigate from which of the many components of their work, the performance improvement mainly arises.

e) The proof of proposition 1 is either sloppy or wrong:
- It is unclear how equation (5) is derived from equation (4). The differential quotient $d(x(t))/dt = lim_{t\to0} (x(t)-x(0))/t$ in equation (5) seems different from taking the limit $t\to0$ of the difference $x(t)-x(0)$ in equation (4). It is further unclear, whether/under which conditions the $min_{R,p}$ at the RHS may compute with a potential $lim$-operation. Isn't it a $min_{R(t),p}$? Seeing intermediate steps might be helpful.
- Description of the proof between equation (5) and equation (6) "The derivative in the LHS of Equation 5 at t = 0 is": Is here really the LHS meant and not the RHS? Assuming the latter case: It would be helpful, if the intermediate steps would all be given. My best assumption for the derivative of the first term is: $d/dt (R(t)(B(t\theta,x(0))-\bar{x}_0)= (R'(t)(B(t\theta,x(0))-\bar{x}_0)+ (R(t)d/dt(B(t\theta,x(0))-\bar{x}_0)=(R'(t)(B(0,x(0))-\bar{x}_0)+ (R(t)d/dt(B(t\theta,x(0))-\bar{x}_0)=(R'(t)(x(0)-\bar{x}_0)+ (R(t)d/dt(B(t\theta,x(0))-\bar{x}_0)$ with the interpretation, that $x=x(0)$. But isn't then the inner derivative like $ d/dt(B(t\theta,x(0))=B'(t\theta,x(0)\theta$ missing?
- Is the step to take gradients wrt. to $v, \omega$ related to the min_{R,p} operation? Or why, does it make sense to take the gradient at this step?
- in general: there seem several name collisions in the proof, which make it hard to understand, e.g. x is used as a parameter of A, but also as a function x(t)

f) For proposition 2:
- Could a highly symmetric molecule part (e.g. with 3 identical atoms attached to a center atom, which is further) attached to a single bond, to which a second unsymmetric molecule part is attached, and a 120°-SO(2)-rotation together with some further SO(2) rotations in the unsymmetric part be a potential counter example to the proposition? (==> for the single bond there seem more parametrizations, which leave the molecule unchanged)

### Novelty

The main idea to use diffusion models to generate 3D coordinates of molecules and their atoms has been suggested at least by [1,7] and also [6]. The so-called "paradigm shift" that the authors claim, has already been done before. The remaining novelty about problem representation and adjustment to docking falls into the application field.

### Reproducibility

The dataset is publicly available, but the code has not been upload as supplementary material, which means that reproducibility is low. The authors should upload the dataset and the code as supplementary material

### Other, minor comments

Many incorrect or tenuous statements:

a) In 4.2 in the formula for overline x, the summation index is missing. In proposition 1, there is also an error in this formula, since over a constant x is summed.

b) Also in Proposition 1, a time dependent x(t) is introduced that is then never used again. The formulas below contain the non-time-dependent x again.

c) "DIFFDOCK significantly outperforms all previous methods": incorrect because no statistical test has been performed.

d) Figure 2 caption "when there is a global symmetry in the protein (aleatoric uncertainty)": Unclear, missing or incorrect description of "aleatoric uncertainty". Does the EquiDock model provide separate estimates for aleatoric and epistemic uncertainty? If it is aleatoric, it cannot be reduced by modeling (unavoidable uncertainty), however, DiffDock somehow decreases it.

e) "However, these approaches learn distributions over the full Euclidean space R3n with 3 coordinates per atom, making them ill-suited for molecular docking where the degrees of freedom are much more restricted." This claim is put forward without any experimental evidence and the mentioned methods are further ignored and not even compared.

### Questions:

- What does ${l,r}$ in the direct sum of formula (15) refer to (esp. r, is only a single l used)?
- Is the magnitude of the distance vector between nodes somewhere used in (15) or does the equation only use the direction?
- What does $\mu$ in formulas (16) and (18) refer to? Is it an MLP? Are they $\mu$'s related (shared)?
- Could the authors elaborate, why the coarse-grained representations don't allow for parity or non-parity?
- Why is the tangent space for torsions SE(3) invariant and not SE(3) equivariant?
- Can you provide a reference, which argues, that an RMSD < 2A is an important criterion? Figure 3 seems to show, that this is exactly the range, where DiffDock performs well. Lowering the threshold, it seems that other methods could gain additional advantages.

### References

[1] Xu, M., Yu, L., Song, Y., Shi, C., Ermon, S., & Tang, J. (2021, September). GeoDiff: A Geometric Diffusion Model for Molecular Conformation Generation. In *International Conference on Learning Representations*.

[2] Mysinger, M. M., Carchia, M., Irwin, J. J., & Shoichet, B. K. (2012). Directory of useful decoys, enhanced (DUD-E): better ligands and decoys for better benchmarking. *Journal of medicinal chemistry*, *55*(14), 6582-6594.

[3] Thomsen, R., & Christensen, M. H. (2006). MolDock: a new technique for high-accuracy molecular docking. *Journal of medicinal chemistry*, *49*(11), 3315-3321.

[4] Gorgulla, C., Boeszoermenyi, A., Wang, Z. F., Fischer, P. D., Coote, P. W., Padmanabha Das, K. M., ... & Arthanari, H. (2020). An open-source drug discovery platform enables ultra-large virtual screens. *Nature*, *580*(7805), 663-668.

[5] Çınaroğlu, S. S., & Timuçin, E. (2019). Comparative assessment of seven docking programs on a nonredundant metalloprotein subset of the PDBbind refined. *Journal of Chemical Information and Modeling*, *59*(9), 3846-3859.

[6] Jumper, J., Evans, R., Pritzel, A., Green, T., Figurnov, M., Ronneberger, O., ... & Hassabis, D. (2021). Highly accurate protein structure prediction with AlphaFold. *Nature*, *596*(7873), 583-589.

[7] Hoogeboom, E., Satorras, V. G., Vignac, C., & Welling, M. (2022, June). Equivariant diffusion for molecule generation in 3d. In *International Conference on Machine Learning* (pp. 8867-8887). PMLR.


**Strength And Weaknesses:**

Strengths:

* Clearly written and structured; hyperparameter selection clearly described
* The principle idea to use diffusion models to predict 3D coordinates of atoms of a molecule is good (however not new; see weaknesses)

Weaknesses

* the work is not novel, but the use of diffusions for molecule conformations has been suggested before (see ref [1], [7])
* the work is poorly embedded into related work and almost fully ignores decades of work in molecular docking
* the mathematical formulation is sloppy and potentially wrong (see below), with notation overloads and inconsistencies
* there is only a single experiment on one dataset which limits the conclusions
* the work is strongly focused on a specific problem in an application domain, with no idea put forward how this would be relevant for ML in general
* the method comparison neglects scientific standards to perform repetitions, provide error bars or confidence intervals and statistical tests. Almost all performance metrics are shown without error bars
* the paper contains many ad-hoc and un-justified decisions
* the paper contains many incorrect or tenuous claims


**Summary Of The Paper:**

The authors present a molecular docking method based on diffusion model. They suggest a diffusion process for this space and an architecture that learns on this process. The authors present results on a version of PDBbind.

**Summary Of The Review:**

The novelty of the proposed method is low because similar works using diffusion processes for molecules have been suggested before. The relevance for machine learning is very limited because many settings of the application domain have been used, therefore, it is unclear how to transfer this to similar ML problems. Technically, there are many severe problems, such as the limited experiments, problems in the method comparison, inconsistencies with previous method comparisons, missing error bars, and sloppy mathematical formalisms. Together with the many false and tenuous claims, the work defies some basic scientific practices.

However, with some improvements this work will be highly interesting for the molecular docking community and will be a valuable contribution there.

---

> ### Author Response · Authors · 2022-11-10
> **Response part 1 of 6**
>
> Thank you for your time and comments. We understand that you have concerns about some aspects of the paper. However, we respectfully suggest that many of these concerns have to do with misunderstandings about our claims and practices. We commence with a broader summary and then address your individual concerns and questions in detail.
>
> We respectfully disagree with the main claims in the review. We spent several paragraphs talking about traditional docking methods and they also form our main baselines. We are unclear how the approach could be contextualised better in this sense. Regarding comparisons to previous methods, we provide 10 different baselines (11 in the updated draft), including state-of-the-art approaches. These comparisons are made on PDBBind, a dataset that collects together a large number of structures and encompasses many smaller datasets. Moreover, we report several metrics in the paper, and focus on %RMSD below 2A since this is widely used in the literature. We fail to understand the point about reproducibility since we provide (anonymous) links to the dataset and code, and refer to them at least 5 times in the submission. We have further clarified  the proofs. We welcome many of your suggestions such as the addition of confidence intervals and statistical significance tests (now in place).
>
> ---
>
>
> > This work does not embed itself into prior and similar works. It almost fully ignores a the decade-old field of molecular docking…. Furthermore, it also ignores close works in machine learning (e.g. [1]).
>
> We have discussed at length and cited traditional docking methods in Sections 1, 2, and 3; our primary evaluations include four of the most common traditional docking methods; and we prominently mention that established docking paradigms were a key inspiration for our approach. Further, [1] albeit solving a different task is referenced both in the introduction and in the related work sections.
>
> ---
>
> > Only a single experiment on a dataset called PDBBind is performed, while there are many more benchmark datasets in the community (e.g. different variants of the DUD datasets).
>
> The amount of crystal structure data is limited, with those on the Protein Data Bank PDB comprising most of them. PDBBind is a curated subset of 19k protein-ligand complex structures that have associated binding affinities. Thus PDBBind's 19k complexes comprise most known data of this type and we aim to use as much of the known data as possible for training and evaluation. E.g. DUD-E [8], that you suggested, only contains 102 proteins of which only 37 are not already in PDBBind which contains 3774 proteins.
>
> We agree that different train-test splits are important and therefore adopted the two established test sets of previous and concurrent work [9, 10, 11].
>
> ---
>
> > Inconsistencies with previous comparisons and focus on single metric that uses a threshold that is convenient for the author's method.
>
> We provide multiple metrics for the accuracy of our predicted poses. This includes the median RMSD, RMSD < 2Å, the fraction of steric clashes/self-intersections, and the full distribution of RMSDs where the performance at all RMSD thresholds can be observed. Whenever we focus on a single metric we chose the RMSD < 2 Å since it is a well-established evaluation metric from the docking community. E.g. the reviewer mentions their citations [3] and [5] and both of these papers employ RMSD < 2 Å as the central metric. Many other prior works also do so [13,14,15] (see below). In the appendix, we also show that DiffDock outperforms all methods on the additional metrics used by EquiBind and TANKBind.
>
> Unlike your citation [5], our method does not predict binding affinity or ligand ranking. Our paper is concerned with predicting the 3D pose of a ligand when bound to a target protein as defined in the first paragraph of our introduction. For this task [5] also exclusively uses RMSD as the metric.

---

> > ### Author Response · Authors · 2022-11-10
> > **Response part 2 of 6**
> >
> > > In a previous method comparison ([5], Figure 4, comparable with Figure 3 of this paper) or another one ([7], Figure 2), the docking methods exhibit much better performance than in the presented work.
> >
> > The docking methods in [5] achieve lower RMSDs since the numbers correspond to a different task. [5] evaluates docking to a known binding pocket where the binding location is given as an additional input to the methods in the form of a restricted bounding box. This will naturally lead to lower RMSD since additional information is provided, dramatically reducing the search space. In contrast, we consider the harder blind docking scenario where the methods have to dock to the whole protein without any prior knowledge about the binding pocket. We acknowledge that three of our baselines were originally developed for pocket-level docking. For this reason, in order to simulate the operating modes of these baselines, our evaluations already included hybrid methods where we use two pocket prediction tools (P2Rank and EquiBind) before docking to the predicted pocket with the search-based baselines (SMINA and GNINA).
> >
> > The mentioned Figure 2 of [7] is an architecture overview and contains no performance comparisons. Moreover, the paper deals with molecular generation, a completely different task to docking.
> >
> > ---
> >
> > > A large number of methods has not been compared. There is a large battery of docking methods that have not been compared, VirtualFlow, Autodock, Glide, GOLD, ... [5]. The authors should compare their method against other docking methods.
> >
> > In the selection of baselines, we mostly follow previously published works [9, 10] while adding additional baselines. This includes the reviewer's suggestion of Glide. We now added the Autodock Vina as a baseline as well. We initially did not include Autodock Vina since it performs worse on the test set than e.g. our other baselines QvinaW and SMINA while also being slower.
> >
> > While it is always possible and helpful to include more baselines, we believe the selection established in previous work as sufficient considering that it includes SOTA open source methods, commercial methods, and ones specifically designed for blind docking.
> >
> > ---
> >
> > > iv) Complete absence of baselines … The authors should include baselines into the method comparison, e.g. generic transformers or SE3-equivariant tranformers, MLPs, generic MPNNs.
> >
> > As discussed above, we compare our performance with 10 different baselines including well-established methods. Regarding different choices of architecture, since our diffusion framework requires the prediction of SE3 equivariant vectors, we employed e3nn [12] which is a widely adopted and popular library. The reviewer mentioned SE3-Transformers [13] which is in fact a model built on e3nn's Tensor field network framework. The other baselines suggested in the reviewer’s comment would not satisfy the requirement of producing SE3-equivariant vector quantities. Please see the comments below for more discussion about specific architectural choices.
> >
> > ---
> >
> > > v) Except for the author's method in Table 1, all performance metrics in text, tables and figures are missing error bars… The authors should provide error bars or confidence intervals for all metrics and perform statistical tests.
> >
> > Thank you for the suggestion. We agree and recognize the value in supporting claims of improvements. We estimate the standard errors via bootstrapping in the estimates of each method and report them in tables 1 and 7 and perform paired two-sample t-tests.  As expected, these estimates confirm that the better performance of our method is indeed statistically significant with the highest p value being 0.0009. Please note that it is computationally infeasible to rerun the training procedures multiple times for almost all of the methods. For some this is not even possible since they are not open source or do not provide training code. For deterministic methods and very computationally expensive docking methods, it is not meaningful or feasible to run the model inference multiple times to estimate the variance in their performance. This is probably why it is not a common practice in docking papers to provide such estimates.

---

> > > ### Author Response · Authors · 2022-11-10
> > > **Response part 3 of 6**
> > >
> > > > c-d) The diffusion process and the model and its architecture contain many ad-hoc decisions. These are hardly justified or possible alternatives explained... The author's should justify their choices, perform ablations studies of the main architectural components and include other architectures as baselines… The authors should investigate from which of the many components of their work, the performance improvement mainly arises.
> > >
> > > We do our best to justify what we believed were the most significant decisions in building the model. The first paragraph of Section 4.1 is dedicated to the diffusion process. In the first paragraph of Section 4.4 we describe the motivation of working at the level of 3D point clouds instead of directly in the internal coordinates. In Section 4.5 and Appendix C we describe the motivation between the main differences between our model architecture and a traditional e3nn architecture. Indeed, there could be many other architectures and diffusion processes, we never claim our architecture or our diffusion process to be the best one could ever design, nor such a claim could ever be justified. Training a model like DiffDock, even in its reduced scale, requires significant computational resources and therefore we were not able to test all possible hyperparameters or architectures. In Appendix D.2 we describe the hyperparameters that were tested and in Appendix F.3 we report the ablation studies that we were able to perform (e.g. model size, diffusion steps, diffusion samples, use or not of protein LM embeddings). There could indeed be better architectures to learn the score model, their design is a non-trivial task worth future contributions.
> > >
> > > ---
> > >
> > > > e) The proof of proposition 1 is either sloppy or wrong:
> > >
> > > We have expanded and revised the proof in the revision, adding several intermediate steps. We hope that these revisions make it easier to follow and its correctness is now clear. We now address each of the specific points below:
> > >
> > > > It is unclear how equation (5) is derived from equation (4).... It is further unclear, whether/under which conditions the minR,p at the RHS may compute with a potential lim-operation.
> > >
> > > In the original version, (5) is derived from by (4) by noting that
> > > $$|y(t)-x| = \min_{R, p} | R(t)(B(t\theta, x)-\bar x) + \bar x + p(t) - x|$$
> > > and
> > > $$|\frac{y(t)-y(0)}{t}| = \min_{R, p} | \frac{R(t)(B(t \theta, x)-\bar x) + \bar x + p(t) - x}{t}|$$
> > > are equivalent minimization objectives for $t>0$. In the revision, we have expanded the derivation and these equations no longer appear in this exact form. Furthermore, no commutation of $\min_{R,p}$ and $\lim$ is involved.
> > >
> > > > Description of the proof between equation (5) and equation (6) "The derivative in the LHS of Equation 5 at t = 0 is": Is here really the LHS meant and not the RHS?… isn't then the inner derivative …missing?
> > >
> > > We have corrected that we mean RHS in the revision. However, the derivative given by the reviewer appears to match ours (equation 6) keeping in mind that $R(t) = I$ when evaluated at $t=0$. In particular, the term highlighted as missing was already present as our second term.
> > >
> > > > Is the step to take gradients wrt. to v,ω related to the min_{R,p} operation? Or why, does it make sense to take the gradient at this step?
> > >
> > > Yes, we set the gradients to zero in order to minimize the objective. We have further clarified the derivation in the revision with additional reasoning for each step.
> > >
> > > > in general: there seem several name collisions in the proof, which make it hard to understand, e.g. x is used as a parameter of A, but also as a function x(t)
> > >
> > > We have clarified the notation in the revision by using $y$ to denote the time-varying structure and $x$ to denote only the static initial structure.

---

> > > > ### Author Response · Authors · 2022-11-10
> > > > **Response part 4 of 6**
> > > >
> > > > > f) For proposition 2:
> > > > Could a highly symmetric molecule part (e.g. with 3 identical atoms attached to a center atom, which is further) attached to a single bond, to which a second unsymmetric molecule part is attached, and a 120°-SO(2)-rotation together with some further SO(2) rotations in the unsymmetric part be a potential counter example to the proposition? (==> for the single bond there seem more parametrizations, which leave the molecule unchanged)
> > > >
> > > > If we understand correctly, the reviewer has in mind an ethane molecule, or something like it. This is not a counterexample, which is explained as follows. Throughout our definitions of the ligand pose manifold, we take atoms to be distinguishable; indeed, the notion of a ligand pose as an element of R^{3n} (which is the starting point of our derivation) explicitly assumes that all atoms are assigned a unique index and are tracked independently. Thus, even if a torsion is applied which results in a chemically identical ligand pose, this is still a different point on the ligand pose manifold. One may say that the ligand pose manifold is chemically degenerate, in the sense that many points are chemically the same, but there nevertheless remains a bijection between the product space and the ligand pose manifold.
> > > >
> > > > Note that although the formalism treats all atoms as distinguishable, the learned model density is nonetheless guaranteed to be permutation equivariant with respect to the arbitrary ordering of identical atoms. This is because the score model is a GNN and has no notion of atomic orderings. Thus, if the ligand pose manifold is chemically degenerate, then the model density at chemically identical points is equal.
> > > >
> > > > ---
> > > >
> > > > > The main idea to use diffusion models to generate 3D coordinates of molecules and their atoms has been suggested at least by [1,7] and also [6]... The remaining novelty about problem representation and adjustment to docking falls into the application field.
> > > >
> > > > We do not suggest that the "idea to use diffusion models to generate 3D coordinates of molecules" is our contribution. Instead, what we claim as our contributions are (see Section 1): (1) novel generative approach to docking (2) novel diffusion process over ligand poses.
> > > >
> > > > The tasks of conformer generation and molecular docking are significantly different. Ignoring the difference between these two problems would run counter to decades of work in computational biology and their importance to the pharmaceutical and medical industry. The scale of the docking problem is also orders of magnitude higher than that of conformer generation. For example, DFT and related methods can be effectively used to obtain accurate conformers but are not even closely applicable to the scale of the molecular docking problem. Regarding [6] we note that AlphaFold is not a diffusion model and not even a generative model.
> > > >
> > > > From the point of view of diffusion methods, the differences between our work and those in [1,7] are quite significant. For example, [1,7] applies the diffusion model directly in the ambient Euclidean space of the data (3D coordinates) analogously to the original works in image generation. In contrast, our method defines a diffusion process on a non-Euclidean submanifold and learns it via a projection of actions to a product space. Finally, the idea of defining a diffusion model on a complex manifold by mapping transformations on the manifold to actions on a tractable (product) group is, to the best of our knowledge, novel in the field and is potentially applicable to many different domains.
> > > >
> > > > ---
> > > >
> > > > > the code has not been upload as supplementary material, which means that reproducibility is low. The authors should upload the dataset and the code as supplementary material
> > > >
> > > > All the code (and a link to the dataset) is available at https://anonymous.4open.science/r/DiffDock. We are a bit surprised by the comment as this link was referenced 5 times in our original submission including in the main text.
> > > >
> > > > ---
> > > >
> > > > > a) In 4.2 in the formula for overline x, the summation index is missing. In proposition 1, there is also an error in this formula, since over a constant x is summed.
> > > >
> > > > Thanks, we fixed the typos.
> > > >
> > > > ---
> > > >
> > > > > b) Also in Proposition 1, a time dependent x(t) is introduced that is then never used again. The formulas below contain the non-time-dependent x again.
> > > >
> > > > We have clarified the notation in the revision. See responses concerning Proposition 1 above.
> > > >
> > > > ---
> > > >
> > > > > c) "DIFFDOCK significantly outperforms all previous methods": incorrect because no statistical test has been performed.
> > > >
> > > > Paired two-sample t-tests have been added, confirming that the improvements in performance were indeed statistically significant (p<0.001 for all methods).

---

> > > > > ### Author Response · Authors · 2022-11-10
> > > > > **Response part 5 of 6**
> > > > >
> > > > > > d) … Unclear, missing or incorrect description of "aleatoric uncertainty". Does the EquiDock model provide separate estimates for aleatoric and epistemic uncertainty? If it is aleatoric, it cannot be reduced by modeling (unavoidable uncertainty), however, DiffDock somehow decreases it.
> > > > >
> > > > > Aleatoric uncertainty refers to natural uncertainty in the labels. For example, if the protein has two identical binding sites the one in which the ligand will be present in the crystal structure from PDBBind used as supervision will be arbitrary. We added a more precise definition and explanation of this concept. Neither EquiBind nor DiffDock provide separate estimates of epistemic and aleatoric uncertainty. Indeed, we are not claiming that DiffDock reduces aleatoric uncertainty which is impossible by definition. What we are claiming is that the treatment of uncertainty by the generative model is more suited to the task than that of the regression model. If there was aleatoric uncertainty between two identical binding poses (and no epistemic uncertainty), a generative model would sample each with 50% probability, while a regression model would always predict the mean of the two. From the point of view of the application, the former is the desired behavior.
> > > > >
> > > > > ---
> > > > >
> > > > > > e) "However, these approaches learn distributions over the full Euclidean space R3n with 3 coordinates per atom, making them ill-suited for molecular docking where the degrees of freedom are much more restricted." This claim is put forward without any experimental evidence and the mentioned methods are further ignored and not even compared.
> > > > >
> > > > > Thank you for raising this issue. We added pointers to Appendix E.2 where we now discuss this point in detail. That local structures (bond lengths, angles, and ring conformations) do not have significant flexibility and are relatively easy to compute is well-known and exploited in most computational chemistry approaches to conformer generation and docking over the past decades. For example, this is used by all the baselines, both search-based and machine learning based, that we compare to in our paper. For experimental evidence of this, we have added a reference to Appendix F of [Jing et al. 2022] where quantitative experiments are run.
> > > > >
> > > > > ---
> > > > >
> > > > > > What does l,r in the direct sum of formula (15) refer to (esp. r, is only a single l used)?
> > > > >
> > > > > As mentioned below the equation t \in {l, r} indicates the node type, i.e. whether the messages are being aggregated from the ligand atoms or the receptor residues (respectively l and r as specified in the Notation paragraph)
> > > > >
> > > > > ---
> > > > >
> > > > > > Is the magnitude of the distance vector between nodes somewhere used in (15) or does the equation only use the direction?
> > > > >
> > > > > Yes, the (radial basis function of the) magnitude of the vector is used as part of e_{ab}, this is specified in the paragraph above called "Node and edge featurization".
> > > > >
> > > > > ---
> > > > >
> > > > > > What does μ in formulas (16) and (18) refer to? Is it an MLP? Are they μ's related (shared)?
> > > > >
> > > > > We define $\mu$ 2 paragraphs above in Appendix C.1 in the paragraph called "Notation" as a radial basis embedding of edge lengths.
> > > > >
> > > > > ---
> > > > >
> > > > > > Could the authors elaborate, why the coarse-grained representations don't allow for parity or non-parity?
> > > > >
> > > > > In an atomic-level representation, the model density is parity invariant because the energy is parity invariant. However, in our coarse-grained representation, the points are the alpha carbons of L-amino acids (since D-amino acids generally do not occur in nature). When the parity of the point cloud is inverted, the new point cloud does not represent the inversion of the atomic-level structure, since the points are still assumed to represent L-amino acids. Thus, the energy does not remain invariant, but rather changes in a way that cannot be directly predicted from the operation of inversion alone. For example, inverting a coarse-grained representation of a RH-alpha helix (made with L-amino acids) would result in a LH-alpha helix (still made with L-amino acids), which do not have the same energy. Since the energy is no longer parity invariant, the learned scores cannot be parity invariant or equivariant.
> > > > >
> > > > > ---
> > > > >
> > > > > > Why is the tangent space for torsions SE(3) invariant and not SE(3) equivariant?
> > > > >
> > > > > Torsion angles are scalar quantities that (unlike position or orientation) are unchanged by elements of SE(3) acting on the ligand pose + protein complex. That is, a change of coordinates does not change the torsion angles of the ligand. The tangent space for torsions is the space of torsion updates; since torsions themselves are SE(3) invariant, their updates must also be. In contrast, position and orientation do vary with a change of basis; thus the updates applied to the position and orientation (i.e., the translation and rotation vectors written in the new basis) must change as well.

---

> > > > > > ### Author Response · Authors · 2022-11-10
> > > > > > **Response part 6 of 6**
> > > > > >
> > > > > > > Can you provide a reference, which argues, that an RMSD < 2A is an important criterion? Figure 3 seems to show, that this is exactly the range, where DiffDock performs well.
> > > > > >
> > > > > > References would be papers [3] and [5] which the reviewer cites where RMSD < 2 is used.
> > > > > > In appendix D.1 we explain that we choose this metric since much prior protein-ligand docking work [13,14,15] considers poses with an RMSD < 2A as “good” or successful and employ it as the main or only evaluation metric.
> > > > > > Our confidence model was trained specifically to distinguish <2 and >2 A RMSD. Thus, it is unsurprising that the model performs best on the threshold for which it was trained. As mentioned, we chose this threshold of 2 A RMSD because it is standard in the field. It is likely that a confidence model trained for any choice of threshold would lead to a performance curve with an elbow at that threshold.
> > > > > >
> > > > > > ## References
> > > > > >
> > > > > > [1] Xu, M., Yu, L., Song, Y., Shi, C., Ermon, S., & Tang, J. (2021, September). GeoDiff: A Geometric Diffusion Model for Molecular Conformation Generation. In International Conference on Learning Representations.
> > > > > >
> > > > > > [2] Mysinger, M. M., Carchia, M., Irwin, J. J., & Shoichet, B. K. (2012). Directory of useful decoys, enhanced (DUD-E): better ligands and decoys for better benchmarking. Journal of medicinal chemistry, 55(14), 6582-6594.
> > > > > >
> > > > > > [3] Thomsen, R., & Christensen, M. H. (2006). MolDock: a new technique for high-accuracy molecular docking. Journal of medicinal chemistry, 49(11), 3315-3321.
> > > > > >
> > > > > > [4] Gorgulla, C., Boeszoermenyi, A., Wang, Z. F., Fischer, P. D., Coote, P. W., Padmanabha Das, K. M., ... & Arthanari, H. (2020). An open-source drug discovery platform enables ultra-large virtual screens. Nature, 580(7805), 663-668.
> > > > > >
> > > > > > [5] Çınaroğlu, S. S., & Timuçin, E. (2019). Comparative assessment of seven docking programs on a nonredundant metalloprotein subset of the PDBbind refined. Journal of Chemical Information and Modeling, 59(9), 3846-3859.
> > > > > >
> > > > > > [6] Jumper, J., Evans, R., Pritzel, A., Green, T., Figurnov, M., Ronneberger, O., ... & Hassabis, D. (2021). Highly accurate protein structure prediction with AlphaFold. Nature, 596(7873), 583-589.
> > > > > >
> > > > > > [7] Hoogeboom, E., Satorras, V. G., Vignac, C., & Welling, M. (2022, June). Equivariant diffusion for molecule generation in 3d. In International Conference on Machine Learning (pp. 8867-8887). PMLR.
> > > > > >
> > > > > > [8] Mysinger, M. M., Carchia, M., Irwin, J. J., & Shoichet, B. K. (2012). Directory of useful decoys, enhanced (DUD-E): better ligands and decoys for better benchmarking. Journal of medicinal chemistry, 55(14), 6582-6594.
> > > > > >
> > > > > > [9] Stärk, H., Ganea, O., Pattanaik, L., Barzilay, R., & Jaakkola, T. (2022, June). Equibind: Geometric deep learning for drug binding structure prediction. In International Conference on Machine Learning (pp. 20503-20521). PMLR.
> > > > > >
> > > > > > [10] Lu, W., Wu, Q., Zhang, J., Rao, J., Li, C., & Zheng, S. (2022). TANKBind: Trigonometry-Aware Neural NetworKs for Drug-Protein Binding Structure Prediction. bioRxiv.
> > > > > >
> > > > > > [11] Zhang, Y., Cai, H., Shi, C., Zhong, B., & Tang, J. (2022). E3Bind: An End-to-End Equivariant Network for Protein-Ligand Docking. arXiv preprint arXiv:2210.06069.
> > > > > >
> > > > > > [12] Geiger, Mario, and Tess Smidt. "e3nn: Euclidean neural networks." arXiv preprint arXiv:2207.09453 (2022).
> > > > > >
> > > > > > [13] Fuchs, F., Worrall, D., Fischer, V., & Welling, M. (2020). Se (3)-transformers: 3d roto-translation equivariant attention networks. Advances in Neural Information Processing Systems, 33, 1970-1981.
> > > > > >
> > > > > > [14] Amr Alhossary, Stephanus Daniel Handoko, Yuguang Mu, and Chee-Keong Kwoh. Fast, accurate, and reliable molecular docking with QuickVina 2. Bioinformatics, 2015
> > > > > >
> > > > > > [15] Nafisa M. Hassan, Amr A. Alhossary, Yuguang Mu, and Chee-Keong Kwoh. Protein-ligand blind docking using quickvina-w with inter-process spatio-temporal integration.Scientific Reports, Nov 2017
> > > > > >
> > > > > > [16] Andrew T McNutt, Paul Francoeur, Rishal Aggarwal, Tomohide Masuda, Rocco Meli, Matthew Ragoza, Jocelyn Sunseri, and David Ryan Koes. Gnina 1.0: molecular docking with deep learning. Journal of cheminformatics, 2021

---

> ### Comment · Reviewer_Yi6P · 2022-11-22
> **Review after author rebuttal**
>
> I have read the authors comments and the updated version. While some minor problems could be clarified, my main concerns mostly remain and some are even aggravated.
>
> ### Main concerns
> * **Low novelty**: the paper proposed an application of diffusion to the problem of molecular docking. The main machine learning components already existed and this means that the novelty falls in an application field
> * Technical errors: **the proof of proposition 1 is incorrect**; see comments below
> * **Severe flaws concerning method comparison and assessment**:
>   * there is still only a single experiment on one dataset and in one setting;
>   * the methods are only compared in a single run;
>   * error bars on performance metrics are based on single runs, which means that differences between methods can just arise by chance; t-test is inappropriate due to the Gaussian assumption
>   * The hyperparameters of docking methods are still hardly adjusted on the training set, such that the comparison to docking methods is unfair
> * Many tenuous claims and conclusions (see, e.g., also below)
>
>
> ### Detailed comments and concerns
>
> > Thank you for the suggestion. We agree and recognize the value in supporting claims of improvements. We estimate the standard errors via bootstrapping in the estimates of each method and report them in tables 1 and 7 and perform paired two-sample t-tests. As expected, these estimates confirm that the better performance of our method is indeed statistically significant with the highest p value being 0.0009. Please note that it is computationally infeasible to rerun the training procedures multiple times for almost all of the methods. For some this is not even possible since they are not open source or do not provide training code. For deterministic methods and very computationally expensive docking methods, it is not meaningful or feasible to run the model inference multiple times to estimate the variance in their performance. This is probably why it is not a common practice in docking papers to provide such estimates.
>
> **The analysis is interesting. However the description of the application of the statistical test is rather rough and needs more details for reproducibility. What exactly is the null and the alternate hypothesis (i.e. which values exactly do you now compare)? Further, the usage of a t-test is questionable due to assumptions on the underlying distribution. More interesting would be to get results from a Wilcoxon signed rank test comparing results from Diffdock vs. all the other methods across ground truth test data.**
> **The conclusion that "better performance of our method" is inappropriate since you only calculate the variability of the estimator of the metrics on the dataset.**
>
>
>
> > We have expanded and revised the proof in the revision, adding several intermediate steps. We hope that these revisions make it easier to follow and its correctness is now clear.
>
>
> **\- First, it seems that for the new version of the proof, the authors apply Taylor series expansions for R, B and p.**
>
> **\- Especially, they seem to ignore higher-order terms, which may be considered a valid approximation, but needs to be stated to be an approximation at the respective steps. Furthermore, it seems that the formal formulation of the proposition is too wide due to the approximations. The authors have to note any limitations of the proposition directly at the proposition statement (e.g. valid only for small t). We acknowledge, that the authors at least tried to argue in the surrounding text, that they are interested in small t, but nevertheless the formulation of the proposition needs to be made more specific.**
>
> **\- Formula 5: Equality between second and third line; this seems not to be an equality, as the term R'B' dt^2 is thrown away. More correct may be the application of a triangular inequality, if the authors, want to pull dt and dt^2 out of the norms, i.e. line 2 $\\leq ||...||dt+||...||dt^2$. In case the authors consider it as an approximation, then it should be indicated as an approximation $\\approx$ and authors should follow our previous comment (concerning limitation of the proposition).**
>
> **\- For the angular momentum: It is not clear, how one can get from a potential (see comments on approximation from before) norm equality $||y'(0)||=||R'(0)(x-\\bar x) +B'(0,\\theta,x)+p'(0)||$ to an equality with the norm $y'(0)=R'(0)(x-\\bar x) +B'(0,\\theta,x)+p'(0)$, which is needed to derive the conclusion for the angular momentum. (equations (7), (10), (11))**
>
> **\- It might not be clear to readers (especially those without extensive physics background), what authors mean with velocity and angular momentum, if the concepts have not already been introduced in the proof, before using the terms, e.g., authors now simply write "We now show that minimizing the velocity results in zero linear and angular momentum." without referring to what is meant with the velocity at this stage.**

---

> > ### Comment · Reviewer_Yi6P · 2022-11-22
> > **Review after author rebuttal part 2**
> >
> >
> > >In the original version, (5) is derived from by (4) by noting that
> > > $|y(t)-x| = \\min\_{R, p} | R(t)(B(t\\theta, x)-\\bar x) + \\bar x + p(t) - x|$
> > > and
> > > $|\\frac{y(t)-y(0)}{t}| = \\min\_{R, p} | \\frac{R(t)(B(t \\theta, x)-\\bar x) + \\bar x + p(t) - x}{t}|$
> > > are equivalent minimization objectives for $t>0$. In the revision, we have expanded the derivation and these equations no longer appear in this exact form. Furthermore, no commutation of $\\min\_{R,p}$ and $\\lim$ is involved.
> >
> > **If t is fixed at the beginning, we agree that $\\min\_{R, p} | R(t)(B(t\\theta, x)-\\bar x) + \\bar x + p(t) - x|$ and $\\min\_{R, p} | \\frac{R(t)(B(t \\theta, x)-\\bar x) + \\bar x + p(t) - x}{t}|$ are equivalent minimization objectives for $t>0$.**
> >
> > **However, at some point for the proof, t is considered a variable again (therefore differentiation wrt. t). We believe that this happens exactly before taking the limit. Then it is not clear, why as given in the original version, the following (commutativity of the $lim$ and $min$) would have worked (considering that R=R(t)):**
> >
> > **$\\lim\_{t\\to0} \\min\_{R(t), p} | \\frac{R(t)(B(t \\theta, x)-\\bar x) + \\bar x + p(t) - x}{t}|= \\min\_{R(t), p} | \\lim\_{t\\to0} \\frac{R(t)(B(t \\theta, x)-\\bar x) + \\bar x + p(t) - x}{t}|$. Or how did the authors justify the differentiation within the $\\min$-term in the original version of the proof?**
> >
> >
> >
> > > We have corrected that we mean RHS in the revision. However, the derivative given by the reviewer appears to match ours (equation 6) keeping in mind that $R(t) = I$ when evaluated at $t=0$. In particular, the term highlighted as missing was already present as our second term.
> >
> > **No, it did not match the author's term. Specifically, as indicated there the inner derivative seemed missing. Our term: $d/dt(B(t\\theta,x(0))=B'(t\\theta,x(0)\\;\\boldsymbol{\\theta}$. The term from the authors: $B'(t\\theta,x(0)$. $\\theta$ could not be found in the original version of the proof.**
> >
> >
> > >If we understand correctly, the reviewer has in mind an ethane molecule, or something like it. This is not a counterexample, which is explained as follows. Throughout our definitions of the ligand pose manifold, we take atoms to be distinguishable; indeed, the notion of a ligand pose as an element of R^{3n} (which is the starting point of our derivation) explicitly assumes that all atoms are assigned a unique index and are tracked independently. Thus, even if a torsion is applied which results in a chemically identical ligand pose, this is still a different point on the ligand pose manifold. One may say that the ligand pose manifold is chemically degenerate, in the sense that many points are chemically the same, but there nevertheless remains a bijection between the product space and the ligand pose manifold.
> > >Note that although the formalism treats all atoms as distinguishable, the learned model density is nonetheless guaranteed to be permutation equivariant with respect to the arbitrary ordering of identical atoms. This is because the score model is a GNN and has no notion of atomic orderings. Thus, if the ligand pose manifold is chemically degenerate, then the model density at chemically identical points is equal.
> >
> >
> > **Thanks, this makes the proof more clear to our opinion.**
> > **Nevertheless, we wonder, whether a short discussion in the manuscript on, whether the ligand pose manifold may be chemically degenerated, might make sense?**
> > **additional comment concerning proposition 2 (not related to the question/answer): The authors should declare the limitation wrt. degenerate cases (collinearity of points along the shared axis of rotation) directly at the proposition in the main text.**
> >
> >
> >
> > > As mentioned below the equation t \\in {l, r} indicates the node type, i.e. whether the messages are being aggregated from the ligand atoms or the receptor residues (respectively l and r as specified in the Notation paragraph)
> >
> > **To our point of view, this should be clarified close to the formula again (it was not explicitly mentioned in the notation section, although one could possibly derive it from the definition of $\\mathcal{V_l}, \\mathcal{V_r}$, $\\mathcal{E}\_{ll}$, $\\mathcal{E}\_{lr}$, $\\mathcal{E}\_{rl}$, $\\mathcal{E}\_{rr}$ in the notation section). We also see a potential name conflict with the l indexing spherical harmonics. Anyway, thanks for the explanation.**
> >
> > > Yes, the (radial basis function of the) magnitude of the vector is used as part of e\_{ab}, this is specified in the paragraph above called "Node and edge featurization".
> >
> > **OK, thanks for clarifying. Due to the non-bold font of $e\_{ab}$, it was not immediately clear at a first glimpse, that the representation of an embedding vector is meant.**

---

> > > ### Comment · Reviewer_Yi6P · 2022-11-22
> > > **Review after rebuttal part 3**
> > >
> > >
> > >
> > > > We define $\\mu$ 2 paragraphs above in Appendix C.1 in the paragraph called "Notation" as a radial basis embedding of edge lengths.
> > >
> > > **OK, thanks nevertheless, we would hope, that the authors, could specify the definition of $\\mu$ more explicitly for reproducibility reasons (i.e. how exactly is the embedding computed: Is it a leared or a fixed embedding, etc.?)**
> > >
> > >
> > >
> > > > In an atomic-level representation, the model density is parity invariant because the energy is parity invariant. However, in our coarse-grained representation, the points are the alpha carbons of L-amino acids (since D-amino acids generally do not occur in nature). When the parity of the point cloud is inverted, the new point cloud does not represent the inversion of the atomic-level structure, since the points are still assumed to represent L-amino acids. Thus, the energy does not remain invariant, but rather changes in a way that cannot be directly predicted from the operation of inversion alone. For example, inverting a coarse-grained representation of a RH-alpha helix (made with L-amino acids) would result in a LH-alpha helix (still made with L-amino acids), which do not have the same energy. Since the energy is no longer parity invariant, the learned scores cannot be parity invariant or equivariant.
> > >
> > > **Ok, thanks for explaining that. We believe, the manuscript would get more accessible, if authors would add this explanation and relate it to the formulas of their model architecture. (maybe to the appendix, if there is not enough space in the main part)**
> > >
> > >
> > > > Torsion angles are scalar quantities that (unlike position or orientation) are unchanged by elements of SE(3) acting on the ligand pose + protein complex. That is, a change of coordinates does not change the torsion angles of the ligand. The tangent space for torsions is the space of torsion updates; since torsions themselves are SE(3) invariant, their updates must also be. In contrast, position and orientation do vary with a change of basis; thus the updates applied to the position and orientation (i.e., the translation and rotation vectors written in the new basis) must change as well.
> > >
> > > **Ok, thanks for clarifying what's meant here with an SE(3) invariant quantity.**

---

### Official Review · Reviewer_UKAX · 2022-10-25

**Confidence:** 5
**Correctness:** 3
**Technical Novelty And Significance:** 4
**Empirical Novelty And Significance:** 4
**Recommendation:** 8

**Clarity, Quality, Novelty And Reproducibility:**

The manuscript is clear and reads well. The theory is well described in detail. As mentioned above, more detail is required regarding data curation and protein-ligand preparation for all methods tested in the manuscript. Details matter when evaluating predictive methods in structure-based drug design, as the movement of a proton or alternative rotamer on a protein can completely change an interaction necessary for binding.

**Strength And Weaknesses:**

**Strengths**:

The proposed diffusion generative model formalism and application to the problem solving a global docking objective is novel and theoretically sound. The addition of the score-based generative model as a ranking mechanism takes the applicability of the diffusion probabilistic method to the next level in terms of posing, scoring, and creating a useful tool.

The comparison to search-based methods (as a quasi-null model) exemplified the significance of the results presented in the manuscript, and clearly DiffDock provides a significant improvement with the experimental protocol performed by the authors. The plots showing the fraction of RMSD vs. other factors further demonstrate the significant improvements compared to other methods and how well the distributions capture the data-generating process.

Weaknesses:
A large weakness of the work appears to be the lack of curation and preparation of the ligand and protein structures obtained from PDBBind. Structure-based drug design is only useful if the structures have been properly prepared by quantifying ionization/tautomeric states for ligands and H-bond optimization/protonation for proteins (garbage in~garbage out). When evaluating the code, it was clear that raw structures were input with minimal curation/preparation. If these raw structures were used as input to the other docking methods, then it is no surprise they performed poorly. The authors should describe their protein-ligand structure preparation protocol in detail.

It is not clear if PDBBind is the best dataset to use for evaluating the global docking problem, as all structures are holo-structures (co-crystal structures; i.e. structures in the ligand-bound conformation). A more relevant test set would be an apo (no ligand bound) structure of the same protein, similar to what is available in the PSCDB (http://idp1.force.cs.is.nagoya-u.ac.jp/pscdb). While this task would be more challenging, it would be more relevant for real-world problems encountered in drug discovery. It is well known in the docking community that predicting <2A˚ poses to holo-structures is much easier if the protein and ligand are prepared correctly, but recapitulating the crystallographic pose when docking to apo structures is much more challenging, even when integrating orthogonal biophysical data collected experimentally.

It appears that only Cα atoms are used for the node and edge featurization of the protein. This is a large limitation as structure-based drug design requires explicit representation of rotamers for amino acids as the orientation/protonation of these residues can dramatically alter what interactions can be made and ultimately whether a molecule will bind with sufficient potency to warrant detection in an in vitro assay.

Some of the molecules generated by this approach are not physically possible to produce and would be incredibly unstable on planet earth.

**Summary Of The Paper:**

The authors propose DiffDock, an approach to global molecular docking as a generative modeling problem that aims to map the non-Euclidean manifold of ligand poses to the degrees of freedom involved in docking sampling by developing a diffusion process on this space. They propose an objective formalism that aims to maximize the expected proportion of predictions with RMSD < some tolerance ϵ motivating the training of a generative model that learns a distribution over ligand poses conditioned on the protein structure as context. In addition, a confidence model is proposed which is trained on the poses sampled by the diffusion model to rank them based on its confidence that they are within an error tolerance. Therefore posing and scoring is achieved by this process. The paper evaluates PDBBind dataset with the task of blind docking evaluated by the RMSD compared to the reference co-crystal structure. DiffDock is then compared to SOTA search-based and deep-learning methods. When evaluating the top scoring pose being <2A˚ from the co-crystal ligand, DiffDock **significantly** outperforms all compared methods with also being 3 to 12 times faster than the best search-based method.

**Summary Of The Review:**

The authors propose DiffDock, a generative diffusion model tailored to the task of molecular docking. The approach fundamentally differs from regression-based frameworks and leverages a generative modeling approach to capture a diffusion process over the manifold describing ligand pose transformations and docking degrees of freedom. DiffDock demonstrates significant improvements over SOTA search-based and deep-learning approaches to global docking on the PDBBind dataset. It is not clear how exactly the protein-ligand structure data was curated and prepared from the PDBBind dataset or why the PDBBind database was chosen for the task of global docking. Still, the data was presented clearly, and the theory was well described. As outlined above, there are clear gaps that must be addressed before DiffDock will be useful for real-world applicability in drug discovery, and it is clear that there is a lack of domain expertise in computational chemistry and structure-based drug design. However, I believe the theoretical contributions and novelty of the manuscript warrant acceptance to ICML following revision.

---

> ### Author Response · Authors · 2022-11-10
> **Response to Reviewer UKAX**
>
> Thank you very much for your thorough review and for taking the time to read the paper in detail! We integrated all your helpful feedback into the paper, and it now discusses your points as outlined below.
>
> ---
>
>
> > The authors should describe their protein-ligand structure preparation protocol in detail.
>
> Thank you for raising this issue. We now clarify in the paper what preprocessing has been applied to the dataset which we downloaded from the preprocessed PDBBind data of the EquiBind paper. This includes preprocessing the protein with Open Babel which is the only step performed in this tutorial from the authors of GNINA and SMINA: https://colab.research.google.com/drive/1GXmk1v8C-c4UtyKFqIm9HnsrVYH0pI-c
> Additionally, the proteins are preprocessed with the "reduce"(https://github.com/rlabduke/reduce) library to add missing hydrogens, correct hydrogens, and correctly flip histidines.
>
> For GLIDE and QVinaW we reuse the numbers from the EquiBind paper where the authors additionally ran the recommended preprocessing scripts. For QVinaW they follow the recommendation of the QVinaW authors to preprocess the data with the prepare_ligand4.py and prepare_receptor4.py scripts of the MGLTools library. For GLIDE the ligand and receptor were converted to the GLIDE input files with the GLIDE command line interface. All these details are now in Appedix D.1 and D.3.
>
> ---
>
>
> > A more relevant test set would be an apo (no ligand bound) structure of the same protein
>
> We recognize the importance of predicting docking directly from apo-structures. However, the amount of data available with both apo and holo-structures is significantly limited (PSCDB contains only 839 complex pairs compared to nearly 19k of PDBBind), and, still, often the sequences in the holo and apo crystal structures are not exactly the same and this makes precise correspondence of the structures questionable.
>
> In order to get around this we have used the recently published OmegaFold to predict the protein structures of the sequences in the PDBBind test set from their sequence alone. We align the residues of the proteins with a weight that decreases exponentially with the distance of the residue to the closest ligand atom. We then run DiffDock on the OmegaFold protein and measure the RMSD to the ligand in the original crystal structure. Preliminary results are as follows (all report % RMSD below 2A):
>
> | Protein structure             | Generative model alone | Generative+confidence models |
> |----------------------|:------------------------:|:-----------------------:|
> | Crystal   | 26.0                   | 45.0                  |
> | OmegaFold | 25.7                   | 31.5                  |
>
> Both results use the same model trained on PDBBind holo structures and were obtained on a subset of PDBBind test set of 260 apo structures (the remaining are multimers or other proteins for which OmegaFold failed). The results highlight that the score model operating at the residue level is robust to the use of apo structures. The difference in the accuracy of the generative model alone is very marginal. On the other hand, as expected, the confidence model trained at atomic level on the crystal structures is sensitive to these changes. This suggests that the result shown above for OmegaFold structures can be significantly improved by training a new confidence model directly on the atomic structures from OmegaFold rather than the crystal. However, the overall performance of the model is already very promising and we expect it to improve further with a better-adapted confidence model.
>
> Note that this approach does not directly model the protein flexibility and its conformational changes. We are also actively working on this problem but it will require more technical advancements and may, therefore, be better suited for a separate future submission.
>
> ---
>
>
> > It appears that only Cα atoms are used for the node and edge featurization of the protein.
>
> We agree that interactions with side chain atoms are important during docking and now discuss the following in detail in our paper. While the score model operates on the Calpha atoms graphs, the confidence model uses an all-atom representation and is able to reason about the mentioned aspects. We also experimented with an all-atom representation for the generative score model but observed no improvement in accuracy while the computational cost increased. A potential reason for this is that the side chain conformation can largely be determined by the information present in the protein sequence and Calpha positions. E.g. AlphaFold has shown that ML methods are often able to accurately infer the sidechain conformation from only the sequence and our score model has all the sequence information and the Calpha atom positions available.
>
> Note that not modeling the side chains directly also makes our score model more robust to the initial structure not being the exact bound structure as shown in the experiments above.

---

### Official Review · Reviewer_5meX · 2022-10-25

**Confidence:** 4
**Correctness:** 4
**Technical Novelty And Significance:** 4
**Empirical Novelty And Significance:** 3
**Recommendation:** 10

**Clarity, Quality, Novelty And Reproducibility:**

**Clarity.** The paper is very clear, both in terms of the paper motivation and model presentation. The experimental results and their interpretation are comprehensible. The figures and the animation in the anonymously shared repository nicely present the idea of the paper.

**Quality.** All the experimental details are provided. The main table contains both classical molecular docking tools (4 of the most widely used docking softwares) and recent generative models (EquiBind and TANKBind). The standard deviation of the performance metrics is provided for DiffDock.

**Novelty.** The novelty of this research work is high. The field of neural docking is not well-explored yet, with only a few recent attempts such as EquiBind and TANKBind. DiffDock reformulates the neural docking task as a generative problem and uses the most recent advances in generative modeling, the diffusion models. It is noteworthy that DiffDock is implemented in a way that ensures that all the intermediate structures are valid (at least in terms of bond lengths and flat aromatic structures) because the diffusion is performed in a product space of the degrees of freedom. The theoretical part of the paper is strongly supported by the proofs of the key findings.

**Reproducibility.** The model description is detailed and contains the key formulas needed to reimplement the method. A pseudocode for the training and inference is provided in the supplementary materials. The set of final hyperparameters is also provided in the supplementary materials. The code is anonymously shared to ensure the full reproducibility.

**Strength And Weaknesses:**

Strengths:
- The authors indicate the problems with regression-based models and explain the motivation of using diffusion models in the context of pose prediction.
- The problem definition and the execution of the model are very good. The authors notice that molecular conformations lie in a submanifold where all the operations should be performed instead of moving the atom positions freely.
- The torsion updates are carefully defined so that they possess properties described in Propositions 1 and 2.
- A confidence model is trained as a classification model that predicts whether the RMSD of the generated pose is less than 2A.
- SE(3)-invariant models are used for the score model and the confidence model. This is important to distinguish two reflections of the same compound (enantiomers).
- The experimental results show the advantages of the proposed method in terms of binding pose prediction and inference runtime.
- Additional experiments as well as all the training details are provided in the supplementary materials.

Weaknesses:
- The only problem I see in the definition of the product space is that it still allows for the steric clashes (either with the protein or other ligand atoms during the torsional rotations). In Figure 5, you argue that self-intersections are not observed for your model. Do you have any intuition why these phenomena do not occur for DiffDock?
- If the confidence model is trained in a classification setup, then the output probabilities do not account for the uncertainty of this model. Do you think it could break the ranking for the unseen binding pockets and ligand poses? Could you elaborate on why you decided to use classification with the threshold of 2A?


**Summary Of The Paper:**

DiffDock is a diffusion model for molecular docking. The authors of the paper formulate the docking problem as a generative modeling task. In order to preserve the key conformational constraints of the molecules, compounds are mapped to a product space of the degrees of freedom, i.e. translation, rotation, and torsional angles. In such created space a diffusion procedure can be defined after ensuring that torsions are orthogonal to the roto-translations, and the mapping is bijective. The diffusion is defined in the product space with a stochastic differential equation and a score model. Furthermore, a confidence model is trained using RMSD values of docked compounds. The experiments show advantage of the proposed method over other classical and neural dockers in the blind docking task.

**Summary Of The Review:**

Based on the above comments, I recommend the acceptance of this paper.

---

> ### Author Response · Authors · 2022-11-10
> **Response to Reviewer 5meX**
>
> Thank you very much for your review. We appreciate your expert feedback and on-point concerns. We respond to all your questions below and clarified these aspects in the paper.
>
> ---
>
>
> > The definition of the product space is that it still allows for the steric clashes ... Do you have any intuition why these phenomena do not occur for DiffDock?
>
> True, the space of possible conformation does include steric clashes, and it also includes cases where the ligand is far from the protein. The reason why these are not observed lies in the accuracy with which the score is learned. Although there are parts of the product space that (implicitly) map to regions with such phenomena, the distribution over the product space that the diffusion model learns to converge to put very little or no probability mass on such spaces. Critical for this to happen is that the score model does not operate directly in the implicit space of the product space (like typical methods working with internal coordinates do) where recognizing steric clashes and multi-hop distances is hard, but it instead operates on the current pose in 3D space (where these phenomena are easy to detect) and predicts the implicit update as output.
>
> ---
>
>
> > If the confidence model is trained in a classification setup, then the output probabilities do not account for the uncertainty of this model. Do you think it could break the ranking for the unseen binding pockets and ligand poses?
>
> Since we take as score the logits of the confidence model these should, in theory, reflect the uncertainty of the model on whether the pose is or not within 2A. However, since the confidence model is trained on the same training data that the score model is trained on, we recognize that its capacity to generalize to unseen complexes could be limited. Therefore we added an analysis of the selective accuracy of the confidence model restricted to unseen receptors setting in Appendix F.2 Figure 7. We can observe that the confidence score’s selective accuracy remains high and close to the perfect selection that would be possible with an oracle that knows the RMSD to the ground truth crystal structure. This suggests that the confidence score generalises well to unseen receptors.
>
> ---
>
>
> > Could you elaborate on why you decided to use classification with the threshold of 2A?
>
> We used 2A because this is widely used as the cutoff defining a docking success in the literature. We also explored directly predicting the RMSD or doing classification with multiple bins, but this did not improve the performance of the model on the validation set across most metrics.

---

> > ### Comment · Reviewer_5meX · 2022-11-23
> > **Thank you for your response**
> >
> > Thank you for addressing my comments and answering my questions. The revised paper improved noticeably. I enjoyed reading the new discussion section in the appendix, including the note on method limitation. After reading other reviews, I still think this paper is outstanding. However, I must agree with Reviewer Yi6P that the proof of Proposition 1 could be further improved. I think that the current version of the proof is much easier to follow, without the big leap introducing the limit. Nonetheless, it took me a while to understand that higher-order derivatives are now ignored because of the infinitesimal $dt$, so the equation should still be considered in the limit (at least this is what I assume). This would explain the Taylor expansion and the missing $dt^2$ term. This assumption should be stated explicitly, or approximations should be used as suggested by Reviewer Yi6P for technical correctness.

---

> > > ### Author Response · Authors · 2022-11-26
> > > **Thank you for your response**
> > >
> > > Thank you for your response and for the additional feedback on the proof of Proposition 1. We changed the paper to now clearly explain why higher order terms can be dropped in the limit which will be in the camera ready version. We appreciate your constructive review and feedback!

---

### Official Review · Reviewer_pHbo · 2022-10-29

**Confidence:** 3
**Correctness:** 3
**Technical Novelty And Significance:** 2
**Empirical Novelty And Significance:** 2
**Recommendation:** 5

**Clarity, Quality, Novelty And Reproducibility:**

This paper takes the diffusion for ligand pose prediction.
I would to see more about the difference between current methods for molecule generation/conformation generation. What are the main unique advantages/differences compared to previous works in applying diffusion?
See above.

**Strength And Weaknesses:**

Strengths:
1. The authors think of the pose prediction task as a generation task and use the advanced diffusion model to perform the task definition. They also talked about the potential problem of the regression-based method. The regression-based method can not fit the overall multiple correct poses in a pocket, while the generative-based model can handle and provide different samples. A confidence model is used to keep the choice of the best ones of predictions.
2. The authors consider the rotation, translation, and torsion angle change as the key messages that are variant during the generation, which largely reduces the freedom space of the ligand pose generation.
3. The experiments are strong and effective compared to previous works, both in accuracy and running time.

Weaknesses:
1. Overall speaking, this paper is not very easy to read and follow (at least for me). The organization is good but not the details. The authors are experts in this domain (this paper is already put on arxiv and the authors are known), but the writing in this paper contains lots of parts that may not be friendly to the general readers. I can understand that there are lots of constraints or domain knowledge that the authors know, but for a paper, it should be easy for the readers to understand. Why I feel hard is that lots of items (words) and knowledge should be discovered by the ML readers. For a reviewer like me (I should say that I know basic knowledge about docking but am not so familiar, at least not familiar as authors), it is really hard for me to capture all of the detailed things that the authors are saying. In comparison, I read TankBand, which is much easier to follow and understand. I have also read the previous work Equiband. I strongly encourage the authors can use much simpler descriptions of the method. By the way, a simple and effective method is acknowledged in the ML committee I think.
2. For some details, there lacks of explanations at all. For example, the m+6 degrees, 6 refers to the rototranslations, what is the detail of this 6? Product space, group space, aleatoric uncertainty, epistemic uncertainty, authors look like to define lots of new items. I am not sure whether these are common knowledge or defined by the authors only. I do feel that these make the readers hard to follow. Even for the diffusion method, details are missing. It's hard to say that 4.3 contains the necessary information for diffusion generation. Lots of details are put in the appendix. This is limited by the page constraints of ICLR. But if this is the reason, a journal that is without page length constraints is better? In the current main text, the training and inference is also not clear, which are placed in the appendix. As for section 3, I am not sure whether this really impresses me, the authors put the whole section to criticize the regression method. What I understand best is that the regression method can not handle the multiple correct poses problem, and the prediction is only the mean pose of these poses. The variance is not enough compared to the generation method. In my view, all of these can be said about the difference between the general regression model and the diffusion model. VAE, flow, diffusion, these models are all good at variance prediction. Is that necessary to put the whole section here, and the cases in figure 2 are somehow cherry-picked?
3. In appendix, the training and inference procedure, the authors talk much about the problem of the different gaps caused by the initial conformation c. In my view, the description can be shortened since contains much about intuitively speaking, and conceptually speaking. Too much description also confused me instead of letting me understand better (only in my view when I read this part).
4. When I read the details of the implementation, I have more concerns. The protein model is initialized of ESM2, a strong protein pre-trained model, but the authors are not mentioned in the main text at all. This strongly pre-trained model may help a lot. Besides, the module ending is also different, the authors include a lot of knowledge features of the molecule, for example, the number of rings and the details of the rings. there are all different from previous works. I am not sure whether these are the main contribution to the accuracy or the diffusion model. I would like to see fair comparisons of the model and encoding ways.
5. For the small model and the large model, the small model is only used for the hyperparameter search? Even like this, it seems that the training time is also long, 18 days on 4 A6000 cards. And, why the inference is conducted on an A100 card then?

**Summary Of The Paper:**

This paper proposes a new way for molecule docking, which is to use diffusion model to form the docking pose prediction as a generation problem. In this work, the prediction is mainly based on the 3D translation group, rotation group and torsion angle change. The authors first critique the regression-based ligand propose prediction. Then define the ligand pose transformations, and the diffusion process, and then the training, model arch. The experiments on PDBBind dataset show that the method is strong with good verification results. The docking accuracy is highly improved and the running time is improved compared with some other methods.

Overall speaking, this paper is strong in terms of performance, but I have lots of concerns.

**Summary Of The Review:**

N/A

---

> ### Author Response · Authors · 2022-11-10
> **Response part 1**
>
> Thank you for your questions. We begin with a high-level summary since the main concerns were readability and understanding, and then answer all your questions in detail .
>
>
> The task our paper tackles is predicting the atom coordinates of a small molecule when it is bound to a protein. Any method predicting such poses needs to select promising ones out of a large set of alternatives. When the target set is uncertain (e.g., inherent symmetries or incomplete information provided to the method), regression methods tend to focus their top prediction on the mean of the possible locations (which is rarely useful). In contrast, generative models can sample all or most of the promising choices, one of which might be useful and/or correct. Thus we reframe the problem as a diffusion generative model over the ligand pose coordinates. Since the chemical structure of a ligand dictates its bond lengths and angles, docking the ligand amounts to finding its position, orientation (relative to the protein), and torsion angles (which are flexible). Learning a generative model over these degrees of freedom is nontrivial. We approach this problem by making use of a product space (a particular combination) of groups (which are common objects in mathematics) that correspond to translations, rotations, and torsion angles. This innovation makes the diffusion particularly efficient and is crucial for DiffDock’s accuracy and inference speed.
>
> ---
>
> > 1. … For a reviewer like me (I should say that I know basic knowledge about docking but am not so familiar, at least not familiar as authors), it is really hard for me to capture all of the detailed things that the authors are saying. …
>
> We are happy to revise the paper to make it more broadly applicable while still keeping the technical material precise. Could point to specific parts of the paper where additional background or explanation might be helpful?
>
> ---
>
> > 2. For example, the m+6 degrees, 6 refers to the rototranslations, what is the detail of this 6?
>
> A rototranslation in 3D is described by 6 degrees for freedom, 3 for the translation (a three-dimensional vector), and 3 for the rotation (e.g. an axis around which to rotate and an angle to rotate by). We have improved the explanation in section 4.1.
>
> ---
>
> > Product space, group space, aleatoric uncertainty, epistemic uncertainty, authors look like to define lots of new items. I am not sure whether these are common knowledge or defined by the authors only.
>
> The terms product space, groups, epistemic and aleatoric uncertainty are all foundational terms within the machine learning community.  We have added some additional explanations for these terms in the revision so as to reduce the use of technical jargon.
>
> ---
>
> > It's hard to say that 4.3 contains the necessary information for diffusion generation. Lots of details are put in the appendix… In the current main text, the training and inference is also not clear, which are placed in the appendix.
>
> We agree that incorporating geometric degrees of freedom (rototranslations and torsion angles) into a diffusion process is a complex, nuanced problem. We have done our best to describe the most salient features and key technical advances in the main text, providing the remaining full details in the appendix. We would be happy to make the exposition clearer if you can point to parts that are confusing.
>
> ---
>
> > The variance is not enough compared to the generation method. In my view, all of these can be said about the difference between the general regression model and the diffusion model. VAE, flow, diffusion, these models are all good at variance prediction.
>
> In section 3 we compare the general approach of using regression methods with that of using generative models. VAE, flow, and diffusion are all common examples of generative models.
>
> ---
>
> > In appendix, the training and inference procedure … description can be shortened since contains much about intuitively speaking, and conceptually speaking. Too much description also confused me instead of letting me understand better (only in my view when I read this part).
>
> We explain these issues thoroughly because they are nuanced and nontrivial. These details are in the appendix for interested readers, and for reproducibility, so as not to distract from the main narrative. We would be happy to clarify any specific confusion.
>
> ---
>
> > The protein model is initialized of ESM2, a strong protein pre-trained model, but the authors are not mentioned in the main text at all. This strongly pre-trained model may help a lot.
>
> We do mention the use of ESM2 in the main text in the architecture section: "Residue nodes receive as initial features language model embeddings trained on protein sequences [Lin et al., 2022]". We have added new ablation studies to Appendix F.3 where we show that even without using ESM2 embeddings the model outperforms all existing methods (34% RMSD below 2).

---

> > ### Author Response · Authors · 2022-11-10
> > **Response part 2**
> >
> > > the module ending is also different, the authors include a lot of knowledge features of the molecule, for example, the number of rings and the details of the rings. there are all different from previous works.
> >
> > The features are actually exactly the same as in EquiBind. They are a merger of the features in GeoMol [Ganea et al. 2021] and the Stanford Open Graph Benchmark library which are very commonly used. They were not modified or tuned in this project.
> >
> > ---
> >
> > > For the small model and the large model, the small model is only used for the hyperparameter search? Even like this, it seems that the training time is also long, 18 days on 4 A6000 cards. And, why the inference is conducted on an A100 card then?
> >
> > Yes, all the hyperparameters were tuned on the small model (see Appendix D.2). It is well-known that diffusion models are slow to train. The original model from [Song et al. 2021] takes 8.4 days on 4 comparable GPUs to train on the much simpler dataset CIFAR-10. We use A6000 GPUs for training because of their availability in our group. Inference was conducted on an A100 because the model does not need 48GB for inference and because they are widely used in many ML benchmarks. Running inference on A100 is approximately 25% faster than that on A6000.
> >
> > ---
> >
> > > I would to see more about the difference between current methods for molecule generation/conformation generation. What are the main unique advantages/differences compared to previous works in applying diffusion?
> >
> > To the best of our knowledge, there are no previous works applying diffusion for molecular docking. Conformer generation is about predicting atom coordinates only relative to the molecule itself while molecular docking is concerned with predicting them relative to a protein. The scale of the docking problem is orders of magnitude higher than that of conformer generation. For example, DFT and related methods can be effectively used to obtain accurate conformers but are not even close to being applicable for molecular docking.
> >
> > The idea of diffusing over a submanifold of the data by mapping a diffusion on a product space of orthogonal actions was critical to obtain an accurate and efficient docking method. This idea is novel across all applications of diffusion models in the literature. An advantage is that previous diffusion models for the different task of conformer generation such as GeoDiff [Xu et al. 2022] took 5000 inference steps compared to the 20 steps since we are diffusing in product space.
> >
> > We hope these answers can convince you of our paper’s merits and we would gladly clarify any further questions or comments.

---

> > > ### Comment · Reviewer_pHbo · 2022-11-22
> > > **Thanks for the rebuttal**
> > >
> > > First I would like to thank the authors for the kind and patient rebuttal. I notice authors have added some descriptions about some terms. Especially thanks for the appendix E and F.
> > > The ablation of ESM model is also interesting that the results are so good even without ESM (but on other hand, it is also upset that the pre-trained model didn't provide too much valuable knowledge).
> > >
> > > I would still recommend the authors when providing the algorithms, the corresponding equations can be mentioned for an easy matching. Besides, the detailed process description of the diffusion process, including formal equations one-by-one, and the process steps can better added in Appendix B. Indeed, I like these details (if readers are like me).

---

> > > > ### Author Response · Authors · 2022-11-26
> > > > **Thank you for your response**
> > > >
> > > > Thank you for your response. We are glad our answers helped clarify the paper. We will include your feedback in Appendix B citing the corresponding equations for the camera-ready revision.
> > > > We are glad you recognize the strong empirical performance in your review and we would be happy to respond to any further concerns that you have.

---

> ### Public Comment · ~Matej_Zečević1 · 2022-11-11
> **Public Comment [NOT a Reviewer]**
>
> **Disclaimer**: I'm not a reviewer for this paper and also not a scientist in this particular area of research. My background is in Causality for AI/ML, which overlaps in the AI/ML part at least. I stumbled upon this review and felt the urge to provide my perspective, that being said, the following text is intended to not affect the chances of the given paper's acceptance to the conference but is rather directly targeted towards the reviewer and the (S)AC. My motivation for this is simply to improve review quality and decision making by making use of what I believe is the main motivation behind the establishment of a system such as OpenReview. My comments aim at being polite and productive with the intention of improvement.
>
> **TL;DR**: I do appreciate the honesty of the reviewer and the fact that I get the impression that the reviewer put in the effort and time with good intentions for creating the review. However, as argued below, clearly the score is not justified by the review and the chosen confidence is too high.
>
> **Extended Notes:**
>
> * While I appreciate the fact that the reviewer is being open about the reviewing process not being double-blind anymore but actually single-blind since the reviewer knows who the authors are, this fact certainly suggest a conflict with the code of conduct / reviewing aspirations of ICLR 2023.
>   * How to improve maybe? This is arguably a problem on a larger scale, since we can expect there to be a number of cases where the reviewing process is de facto single-blind (for the same reasons), and the reviewers actually not being explicit about it. Kudos to the reviewer for speaking out IMHO.
> * IMHO the overall goal that papers should be maximally accessible is great as it fosters growing community with diverse opinions, in this point the reviewer and I share the same (philosophical) conviction. However, I believe using this argument (which by the way is of rather subjective nature since we do not have a formalization of what makes a paper "easily / clearly understandable") is not fair.
>   * How to improve maybe? Best would be to simply point to how a certain aspect of the paper can be made more accessible, but I do recognize the difficulty in the case where the reviewer is not an expert. In this case, I'd propose to provide an explicit disclaimer.
> * The key argument of the review seems to be (1), which is being repeated in major parts of (2) and (3). Inflating the section discussing weaknesses of the proposal with the very same argument is a No-Go IMHO. Furthermore, in (3) the appendix is being used for criticism. While the AI/ML community has reached a de facto standard on also considering the appendix of a submission, there is still the separation between main text and appendix (as otherwise the difference would only turn out to be in the formalities) and using the latter for critcism is IMHO unfair.
> * Proposing the authors to go for a journal purely because of formalities like page limit (since journals in AI/ML tend to be more relaxed regarding such restrictions) is IMHO disrespectful towards the authors who chose to submit to ICLR 2023 with intent since they found their work be relevant for the call for papers, therefore such a comment should not find its place in any review.
> * Furthermore, how come that the confidence of this review is being set to an arguably high value (indicating significant confidence for what is claimed), when the reviewer proclaims that he/she is not an expert? To quote, "basic knowledge of docking" only, or aspects such as aleatoric and epistemic uncertainty being undefined which are neither specific to GDL nor any unknown, advanced topics IMHO, are all indicators for a low confidence in the review. Confidence accounts are important as they help (S)AC to catgeorize the reviews w.r.t. to their level of expertise and familiarity with the subject of interest.
> * The strong point of the paper, namely "strong improvements in both accuracy and running time" as pointed to by the review, is arguably everything one can wish for from a new methodology in ML. Assuming for the moment that there are no issues with what is being presented in the sense that the method indeed does improve significantly in both terms as suggested and does not try to "sell" something, then both logically and historically speaking an acceptance of the proposal is warranted if not guaranteed.
>   * How to improve maybe? Be more particular, to the extent possible, of what actually makes these improvements be "strong", this also helps other readers to better understand where the key research problems in the given field are found (which was THE key point raised by the reviewer: accessibility).
>
> I really do hope that the above analysis with respective pointers for improvement can help improving the decision processes in our peer-reviewing more generally, maybe starting from this particular paper. I do hope that the reviewer and (S)AC can make good use of the points raised.
>
> Thank you.

---

### Decision · Program_Chairs · 2023-01-20

**Decision:**

Accept: poster

**Justification For Why Not Higher Score:**

There were some concerns that the level of technical innovation is not very high, as compared to prior work on manifold and torsional diffusion papers.

**Justification For Why Not Lower Score:**

It was felt that treating docking as a generative task is novel and interesting, and the experimental results are quite promising.

**Metareview: Summary, Strengths And Weaknesses:**

This paper proposes to study the docking problem as a generative task, in which diffusion models were used to generate molecular 3D shapes in order to bind to a target protein. There are some different opinions during review and discussions on this paper. An AC-reviewer video conference was hold to discuss this paper, and the disagreement was not completely resolved. Subsequently, there were intensive discussions between the AC and SAC after the video conference, and both the AC and SAC have read the paper in detail. Our conclusions are as below:

1. It was felt that treating docking as a generative task is novel and interesting, and the experimental results are quite promising. Although the level of technical innovation is not very high, as compared to prior work on manifold and torsional diffusion papers, treating docking as a generative task is novel and interesting with promising results.

2. There were some concerns on the correctness of the technical details of the paper. We checked these details and believe these issues have been addressed during rebuttals.

3. After thoughtful deliberations, the AC and SAC agreed on to recommend an accept.


**Note From Pc:**

if the above contains the word "oral" or "spotlight" please see: "oral" presentation means -> notable-top-5% and "spotlight" means -> notable-top-25%. As stated in our emails, we are disassociating presentation type from AC recommendations

**Summary Of Ac-Reviewer Meeting:**

A video conference was hold between the AC and three reviewers (Reviewer Yi6P was not able to make it, but emailed comments to the AC).

Reviewer 5meX was quite positive about his work, but admits that he/she did not read the technical details of the diffusion part. This reviewer likes the approach to model the docking as a generative approach, and appreciates the authors’ efforts in providing code.

Reviewer UKAX felt the overall approach is novel and liked the idea of using diffusion to solve docking problems. This reviewer appreciates the efforts in adding details on data curation and preparation during rebuttal. The improvements in experimental results are significant compared to prior methods.

Reviewer pHbo felt this is a practical work and the results are quite significant. This is also the first time that diffusion models are used in docking problems, and this reviewer was not sure if diffusion model is suitable for this task, and not sure why diffusion, rather than other generative models, was used. This reviewer also noticed the provided code might be slightly different with what described in the paper.